# A Sober Look at the Robustness of CLIPs to Spurious Features

**Qizhou Wang**[1][*]   **Yong Lin**[2][*]   **Yongqiang Chen**[3][*]
**Ludwig Schmidt**[4]    **Bo Han**[1][†]    **Tong Zhang**[5]

[1]TMLR Group, Department of Computer Science, Hong Kong Baptist University
[2]The Hong Kong University of Science and Technology
[3]The Chinese University of Hong Kong
[4]University of Washington
[5]University of Illinois Urbana-Champaign

**https://counteranimal.github.io/**

## Abstract

Large vision language models, such as CLIP, demonstrate impressive robustness to spurious features than single-modal models trained on ImageNet. However, existing test datasets are typically curated based on ImageNet-trained models, which aim to capture the spurious features inherited in ImageNet. Benchmarking CLIP models based on the ImageNet-oriented spurious features may not be sufficient to reflect the extent to which CLIP models are robust to spurious correlations within CLIP training data, e.g., LAION. To this end, we craft a new challenging dataset named `CounterAnimal` designed to reveal the reliance of CLIP models on realistic spurious features. Specifically, we split animal photos into groups according to the backgrounds, and then identify a pair of groups for each class where a CLIP model shows high-performance drops across the two groups. Our evaluations show that the spurious features captured by `CounterAnimal` are generically learned by CLIP models with different backbones and pre-train data, yet have limited influence for ImageNet models. We provide theoretical insights that the CLIP objective cannot offer additional robustness. Furthermore, we also re-evaluate strategies such as scaling up parameters and high-quality pre-trained data. We find that they still help mitigate the spurious features, providing a promising path for future developments.

## 1   Introduction

Large vision language models (LVLMs) have demonstrated huge success across a wide range of vision and multi-modal tasks, surpassing conventional ImageNet (-trained) models by a remarkably large margin [1]. LVLMs are typically trained with or based on Contrastive Language Image Pre-training (CLIP) [2] on an unprecedented scale of real-world vision and language data such as LAION [3], which are significantly larger than ImageNet. The huge success of CLIP has presented a paradigm shift for modern vision and vision-language models to conduct the pre-training from ImageNet benchmarks to web-scale multi-modal datasets [4].

A key signature of CLIP models is the impressive robustness against various ImageNet-oriented distribution shifts [2], which is shown to be prohibitive to ImageNet models [5]. The performance

---

[*]Equal contributions.
[†]Correspondence to Bo Han (bhanml@comp.hkbu.edu.hk).

38th Conference on Neural Information Processing Systems (NeurIPS 2024).

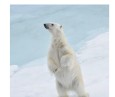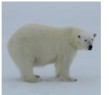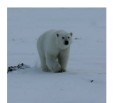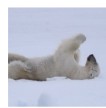 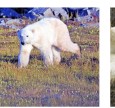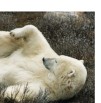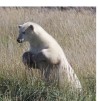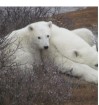

Photos of **ice bear** in **snow** background (**easy**, accu 97.62)  Photos of **ice bear** in **grass** background (**hard**, accu 70.91)

Figure 1: We showcase `CounterAnimal` examples from the class of `ice bear`, separated into `easy` and `hard` groups with different backgrounds (i.e., `snow` and `grass`). The zero-shot performance of `CLIP-LAION400M-ViT-B/32` drops from 97.62% (`easy`) to 70.91% (`hard`).

boosts over ImageNet models seem to suggest that CLIP resolves distribution shifts, thereby sparking a rich discussion about its rationale [6, 7, 8, 9, 10]. However, *the elephant in the room* is that adopted testsets (i.e., ImageNet variants) to evaluate the robustness of CLIPs are primarily designed for ImageNet-based models [5, 11]. These datasets may not correctly reflect the exact robustness of CLIP, given that CLIP models are trained on a large amount of data that may include, and possibly extend beyond those ImageNet variants during pre-training [10]. In this paper, we investigate the robustness of CLIP to distribution shifts caused by the presence of spurious features. These features are highly correlated with labels, but this correlation may break down under distributional shifts [12, 13, 14, 15, 16, 17, 18, 19, 20]. We raise a challenging research question in the following:

*Is there a benchmark that reflects the exact reliance on spurious features of CLIP?*

Sadly, most of the existing benchmarks [21, 22, 23, 24] are tailored primarily for ImageNet models, which are unsuitable for CLIP. To fill this gap, we introduce a new testset, named `CounterAnimal`, specifically designed for assessing the robustness of CLIP models against real-world spurious features. Figure 1 presents several examples of `CounterAnimal` where data are divided into two groups, a) the `easy` group: animals in commonly appeared backgrounds that the CLIP models make correct predictions, and b) the `hard` group: animals in less commonly yet still plausible backgrounds, where the CLIP models are likely to misclassify them. Intuitively, the `easy` part captures some real-world biases that the web-scale data may naturally inherit. Hence, by comparing the performances of the two groups, one can quantify to what extent the model relies on spurious features.

More specifically, the `CounterAnmial` dataset is curated based on raw photos collected from iNaturalist[3]. The construction pipeline consists of 4 steps. a) Data collection: querying iNaturalist with each animal class, where we select some of the animal names from the ImageNet-1K dataset [25]. b) Data curation: manually cleansing low-quality photos that potentially contain ambiguity and corruption. c) Background labeling: manually annotating photos with their respective backgrounds, selected from the label space of the candidate backgrounds. d) Spurious discovering: preserving classes and associated data based on the decrease in zero-shot performance.

Table 1: 1 vs. 1000 results of exemplary animal classes within the `CounterAnmial` dataset for `CLIP-LAION400M-ViT-B/32`. "bkg" denotes the background label, "accu" (%) denotes the zero-shot accuracy, and "drop" (%) denotes the drop in accuracy between `easy` and `hard` groups.

| object label | easy | | hard | | drop |
|---|---|---|---|---|---|
| | bkg | accu | bkg | accu | |
| ice bear | snow | 97.62 | grass | 70.91 | 26.71 |
| black swan | water | 93.63 | earth | 68.87 | 24.76 |
| flamingo | water | 79.70 | sky | 55.45 | 24.25 |
| vulture | sky | 87.76 | tree | 41.84 | 45.92 |
| dung beetle | earth | 56.92 | hand | 17.02 | 39.90 |

mance (i.e., evaluating based on pre-trained CLIP models without fine-tuning) when shifting the backgrounds. The resulting `CounterAnimal` dataset covers a total of 45 animal classes, and ends up with 7,174 `easy` photos and 5,926 `hard` photos, aligning with the standard size as an evaluation dataset, such as [26, 27]. Moreover, `CLIP-LAION400M-ViT-B/32` is used as the proxy CLIP model in spurious discovering (cf., Appendix C.5 for the model naming rules).

We evaluate the CLIP models on our `CounterAnmial` with various backbones, e.g., ViT [28], along with different pre-train datasets, e.g., LAION [3]. We also consider more advanced LVLMs like MiniGPT4 [29] and LLaVA [30]. We employ two evaluation setups crafted for different families of models (cf., Appendix C): a) **1 vs. 1000 setup**: using the full ImageNet-1K class names as the candidate label space and b) **1 vs. 20 setup**: using the top-20 most confusing classes regarding `CLIP-LAION400M-ViT-B/32` as the candidate label space. We provide some of results in Table 1 and Figure 2, highlighting the key observations in the following:

---

[3]`https://www.inaturalist.org/observations`

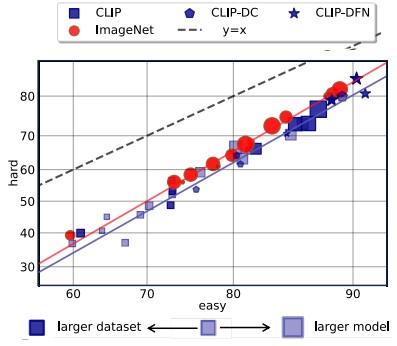

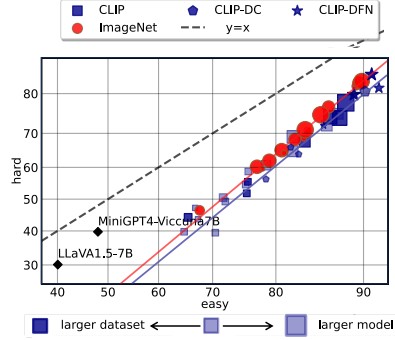

(a) **1 vs. 1000** (label space of ImageNet-1K)  (b) **1 vs. 20** (20 most confusing labels per class)

Figure 2: The `easy` vs. `hard` performance (%) for CLIP, ImageNet models, and more advanced LVLMs, i.e., MiniGPT4 and LLaVA. The marker size indicates the backbone scale and the color shade indicates pre-train data scale. We highlight the CLIP models pre-trained on high-quality datasets, i.e., DataComp (CLIP-DC) and Data Filtering Networks (CLIP-DFN). We linearly fit the trends for CLIP (CLIP, CLIP-DC, and CLIP-DFN) and ImageNet models to show their effective robustness. We also depict the perfect trend, i.e., $y = x$, where the models will not learn any bias.

`CounterAnimal` **captures general spurious correlations within CLIP.** As exemplified in Table 1, we observe a significant drop of `CLIP-LAION400M-ViT-B/32` in zero-shot accuracy from the `easy` to `hard` groups for each class. Furthermore, the observed biases in `CLIP-LAION400M-ViT-B/32` also generalize to other CLIP configurations, with non-trivial performance drop from the `easy` to `hard` groups across various backbones and pre-train datasets as in Figure 2. It implies that `CounterAnimal` characterizes some general spuriousness common in large-scale multi-modal datasets.

**ImageNet models are more robust to spurious correlations captured by** `CounterAnimal`. Figure 2 also illustrates the performance changes of ImageNet models (colored in red). Compared with CLIP models (colored in blue), we find that ImageNet models exhibit stronger robustness to spurious correlations captured by `CounterAnimal`. Our findings contrast with previous studies that assess the ImageNet variants [2], highlighting that CLIP models do not always generalize better than ImageNet models. It underscores the necessity of choosing appropriate benchmarks to comprehensively assess the robustness of different models and training schemes.

**Larger CLIP models are more robust.** Shown also in Figure 2, we use the sizes and the color shades of the markers to indicate the scales of backbones and the pre-train datasets, respectively. Overall, larger CLIP backbone models (i.e., larger markers) can improve the effective robustness, implying that scaling up backbones may enhance model performance against spurious features. In contrast, increasing the scale of the pre-train dataset (i.e., darker markers) does not yield the same improvement, implying that collecting more data alone cannot rectify much bias, which provides some new understanding in addition to the data-centric perspective [6, 10].

**CLIP models trained on high-quality data are more robust.** We categorize CLIP models into two distinct groups according to the pre-train data quality: a) CLIP-DC using DataComp [4] and CLIP-DFN employing Data Filtering Networks [31], as well as b) those pre-trained on datasets that lack stringent curation, labeled simply as CLIP. The results indicate that CLIP models pre-trained on high-quality datasets demonstrate enhanced robustness in general. It suggests that enhancing data quality remains a promising strategy for mitigating the spurious features.

**The CLIP objective may not offer additional robustness.** Complementary to our empirical observations, we also provide theoretical explanations for the reasons why CLIP learns spurious features. We further conduct confirmatory experiments that fine-tune CLIP models onto datasets with synthetic spurious features. The results align with our observations on `CounterAnimal` that the CLIP objective can not offer additional robustness over standard single-modal supervised training.

**Comparison with previous results.** Our work presents a new benchmark to effectively and systematically evaluate the robustness of CLIP models, which complements the literature in understanding the generalizability of CLIP models and LVLMs. More specifically, [32] reports that CLIP models

may wrongly align co-occurred objects with their texts. [33] reports similar failure modes for more sophisticated LVLMs such as MiniGPT4 or LLaVA. [34] finds that CLIP misaligned samples will further cause the hallucination of LVLMs. Complementary to these works, our study explicitly characterizes the spurious features captured by CLIP and explains the existence of the reported failure cases. Our study provides interesting empirical and theoretical counterexamples to the previous beliefs for the substantial improvements in robustness for CLIP models, especially for those results observed on ImageNet variants [7, 8, 9, 35]. Based on the newly collected `CounterAnimal` dataset, we suggest that distribution shifts remain an open problem for CLIP models. Also, we need to be cautious about the test setups when evaluating new models pre-trained on datasets that differ significantly in scales and distributions from traditional ImageNet models.

**Comparison with previous benchmarks.** There are many other datasets to study distribution shifts, e.g., ImageNet variants [21, 26, 36, 37, 38, 39, 40], `DomainBed` [41], and `Wilds` [42]. However, these datasets have biases when assessing the OOD robustness of CLIP models, as they may fail to represent the true OOD scenarios during CLIP training. Moreover, numerous recently released datasets, such as [22, 23, 43, 44, 45], have also explored distribution shifts. However, these studies primarily focus on synthetic distribution shifts, which may not fully represent real-world cases. In fact, it has been shown that previous OOD benchmarks are contained in CLIP training [10], making it hard to ablate ID/OOD cases for data in these benchmarks. Consequently, CLIP models have shown to be more robust than ImageNet models on these contaminated datasets [46].

## 2 Dataset and Evaluation Setups

To begin with, we describe the basic experimental setups, including the pipelines in constructing `CounterAnimal`, its key characteristics, as well as the adopted evaluation settings.

### 2.1 Construction of `CounterAnimal`

We introduce the curation pipeline of our new dataset `CounterAnimal`, tailored for CLIP to investigate spurious correlations. The pipeline consists of 4 steps as follows:

**Data Collection.** We query animal names listed in the ImageNet-1K dataset and collect raw data via the search interface of iNaturalist, a global biodiversity data-sharing platform. We retrieve the latest 300-800 photos per animal class, organizing them based on the queried labels.

**Data Curation.** The collected raw samples are susceptible to noise and ambiguities. Therefore, we manually cleanse the low-quality data that fall into any one of the following 4 situations: label noise, feature noise, obscurity, and clarity. Label noise refers to cases where photos do not belong to the queried classes; feature noise refers to cases where some pixels are disrupted or missing; obscurity occurs when photos belong to more than one object class; clarity issues refer to cases where animal objects are largely occluded by the backgrounds or other irrelevant objects. It also includes the cases where animal objects do not occupy the majority of the space in photos.

**Background labeling.** We consider a typical form of spurious features where the backgrounds of photos can be biased [47]. To identify such data for CLIP models, we manually label the backgrounds for the curated data. The considered class space of backgrounds is defined as follows: `ground`, `water`, `earth`, `sand`, `snow`, `grass`, `human`, `sky`, `road`, `rock`, `shrub`, `indoor`, `tree`, and `outdoor`. Note that the class space of backgrounds as above is not entirely orthogonal due to the inherent ambiguity: Some backgrounds may be ambiguous and some photos may contain more than one background. Nevertheless, we try our best to determine the assigned background labels for each animal class and exclude those photos challenging to be labeled.

**Spurious Discovery.** For each class, we quantify the impacts of spurious correlations to CLIP models by comparing the performances on the associated samples across different backgrounds. We take those classes as containing spurious features on which we observe a relatively obvious decrease in accuracy when changing backgrounds. In realization, we adopt the checkpoint of `CLIP-LAION400M-ViT-B/32` for evaluation, where the prompt for its text encoder is "A photo of <object label>.", and the space of <object label> is the ImageNet-1K class names, i.e., we follow an 1 vs. 1000 setup. Then, we consider the classes where the zero-shot accuracy varies by more than 5% when changing backgrounds as the cases where CLIP model has learned the spurious features. The data with the preserved classes and backgrounds are used to create our final

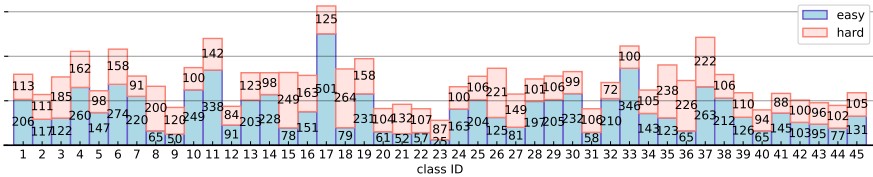

Figure 3: The data layout across various animal classes. The horizontal axis denotes the class IDs and the vertical axis denotes the number of photos for the `easy` and `hard` groups, respectively.

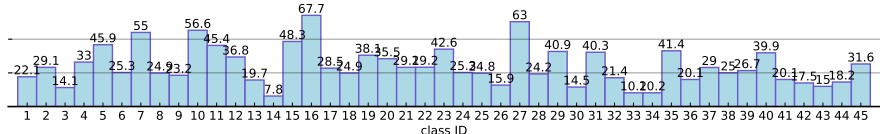

Figure 4: The 1 vs. 1000 performance drop (%) with `CLIP-LAION400M-ViT-B/32`. The horizontal axis denotes the class IDs and the vertical axis denotes the percentage points of decline.

`CounterAnimal` dataset. Photos with the highest CLIP accuracy are assigned to the `easy` group, and those with the lowest CLIP accuracy are assigned to the `hard` group. We further refine the collected data to remove any mistake that the labelers may made during data curation and background labeling.

Our objective in developing `CounterAnimal` is to reflect the spurious correlations learned by CLIP. Therefore, we need to employ the CLIP models for dataset curation and thus ensure the construction is effectively biased towards CLIP configurations [26]. In Appendix E, we further show that our data curation pipeline is general and reliable to characterize the spurious features within the considered models. Moreover, our experimental results later in Section 3 will corroborate that the spurious features captured by our `CounterAnimal` dataset are general across different CLIP setups and may not be so influential for ImageNet benchmarks. These findings will justify that our crafted testset satisfies our primary objective in characterizing the spuriousness for CLIP specifically.

## 2.2 Characteristics of `CounterAnimal`

We depict the data layout in Figure 3 and visualize the zero-shot gaps for each animal class in Figure 4, where we use `CLIP-LAION400M-ViT-B/32` as our referred model. Please refer to the detailed object/background names concerning the `easy` and `hard` groups in Appendix B. Recalling that, when CLIP models resort to the shortcut of data, the model performance will heavily correlate with the backgrounds presented in the `easy` group yet is compromised when coming to the `hard` group. Accordingly, Figure 4 implies a reliance for the CLIP models on the backgrounds.

## 2.3 Evaluation Setups

We evaluate a series of CLIP models on the `CounterAnimal` dataset for their zero-shot performance. For each class, we use the pre-defined prompt of "`A photo of <object label>.`" as in our data collection procedure and the similarity between image and text embeddings in classification. By default, we use the label space of the ImageNet-1K dataset and report the top-1 accuracy, i.e., the 1 vs. 1000 setup. Moreover, when involving more advanced LVLMs, we adopt the 1 vs. 20 setup where we employ the top-20 most confusing classes regarding `CLIP-LAION400M-ViT-B/32` as the candidate label space. For re-productivity, we adopt the pre-trained CLIP checkpoints from OpenCLIP [48] and ImageNet model checkpoints from the PyTorch repository. The model naming rules are in Appendix C.5 and the evaluation details are discussed in Appendix C.

## 3 Experimental Analysis

Our experiments center on the evaluation and the analysis of our `CounterAnimal` dataset. In Section 3.1, we examine the generality of the captured spurious correlations. In Section 3.2, we explore the potential facets that affect the robustness of CLIP models. In Section 3.3, we extend the evaluation to a broader family of models with different training paradigms.

## 3.1 Generality of the Spurious Correlations

In Section 2.1, we discover spurious correlations using `CLIP-LAION400M-ViT-B/32` and collect associated data to build the `CounterAnimal` dataset. A critical problem then arises: Is our dataset a general benchmark to examine spurious correlations of CLIP with other pre-train datasets and backbones? Hence, we need to examine whether the biases in the `CounterAnimal` dataset can hinder the robustness of other CLIP models, where we consider two situations: a) fixing pre-train datasets while varying backbones and b) varying pre-train datasets while fixing backbones.

**Varying Backbones.** We fix the pre-train dataset to be `LAION400M` and explore two other backbones within the ViT family [28], i.e., `ViT-B/16` and `ViT-L/14`. Their zero-shot results are depicted in Figure 5(a). There remains a drop above 17 percentage points for both the cases of `ViT-B/16` and `ViT-L/14`. It suggests that the `CounterAnimal` dataset captures some general spurious shifts that are at least present in the pre-train dataset of `LAION400M`.

**Varying Pre-train Datasets.** We fix the backbone to be `ViT-B/32` and consider other pre-train datasets. Here, we consider `LAION2B` and the closed-source dataset used by OpenAI. Their `easy` and `hard` results are in Figure 5(b). Here, the spurious features affect the zero-shot robustness of CLIP models trained on both `LAION2B` and by OpenAI, indicating that our `CounterAnimal` dataset possesses some realistic shifts that are contained in various CLIP setups. Therefore, we conclude that `CounterAnimal` captures some general spurious features learned by CLIP models.

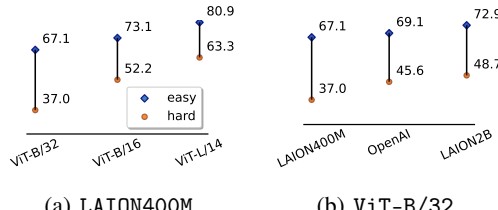

(a) `LAION400M`  (b) `ViT-B/32`

Figure 5: The 1 vs. 1000 results for varying CLIP setups beyond `CLIP-LAION400M-ViT-B/32`: a) fixing the pre-train dataset to be `LAION400M` and b) fixing the backbone to be `ViT-B/32`.

## 3.2 Scaling up May Relieve Spurious Correlations

We extend our evaluations to a wider range of CLIP models with different scales of parameters and pre-train data. The results are summarized in Table 2 and further depicted in Figure 2(a). Generally speaking, performance drops can be observed across all considered CLIP configurations, indicating that CLIP models in various scales still learn spurious features. More specifically, we investigate the influence of a) parameter scales and b) pre-train data scales in CLIP models on the sensitivity of spurious features. We exclude the backbone of `ViT-B/32` and the dataset of `LAION400M` to avoid biases in data collection.

**Scaling up Pre-train Data.** To test the impacts of enlarging scales of pre-train datasets, we consider two CLIP backbones, namely, `ViT-B/16` and `ViT-L/14`, along with a series of pre-train datasets of increasing sizes. The results are summarized in Figure 6. We observe that scaling up the data scale does not necessarily reduce the performance drop, suggesting that directly enlarging the scale of pre-train data alone cannot enhance robustness. One possible explanation is that larger datasets do not imply fewer biases, whereas the CLIP models will still inherit the spurious correlations therein.

Table 2: The 1 vs. 1000 results for CLIP checkpoints on the `CounterAnimal` dataset. The pre-train datasets with high-quality data are marked by *.

| backbone | pre-train dataset | easy | hard | drop |
|---|---|---|---|---|
| RN-101 | OpenAI | 64.27 | 45.15 | 19.12 |
| RN-50×4 | OpenAI | 70.02 | 49.07 | 20.95 |
| ViT-B/16 | LAION400M | 73.11 | 52.17 | 20.94 |
| ViT-B/16 | OpenAI | 73.08 | 56.56 | 16.52 |
| ViT-B/16 | DataComp1B* | 80.36 | 64.24 | 16.12 |
| ViT-B/16 | LAION2B | 73.18 | 53.18 | 20.00 |
| ViT-B/16 | DFN2B* | 85.03 | 70.61 | 14.42 |
| ViT-B/32 | LAION400M | 67.13 | 36.95 | 30.18 |
| ViT-B/32 | OpenAI | 69.13 | 45.62 | 23.51 |
| ViT-B/32 | DataComp1B* | 75.96 | 53.74 | 22.22 |
| ViT-B/32 | LAION2B | 72.94 | 48.74 | 24.20 |
| ViT-L/14 | LAION400M | 80.90 | 63.31 | 17.59 |
| ViT-L/14 | OpenAI | 85.38 | 70.28 | 15.10 |
| ViT-L/14 | DataComp1B* | 89.29 | 79.90 | 9.39 |
| ViT-L/14 | LAION2B | 82.23 | 66.27 | 15.96 |
| ViT-L/14 | DFN2B* | 90.77 | 80.55 | 10.22 |
| ViT-L/14-336 | OpenAI | 86.36 | 73.14 | 13.21 |
| ViT-H/14 | LAION2B | 85.74 | 73.13 | 12.61 |
| ViT-H/14 | DFN5B* | 88.55 | 79.13 | 9.42 |
| ViT-G/14 | LAION2B | 86.81 | 73.32 | 13.49 |
| ViT-bigG/14 | LAION2B | 87.57 | 76.96 | 10.61 |

**Scaling up CLIP Model Sizes.** We also explore the connection between model scales and spurious correlations. In Figure 7, we consider two pre-train datasets, namely, `LAION2B` and the close-soured

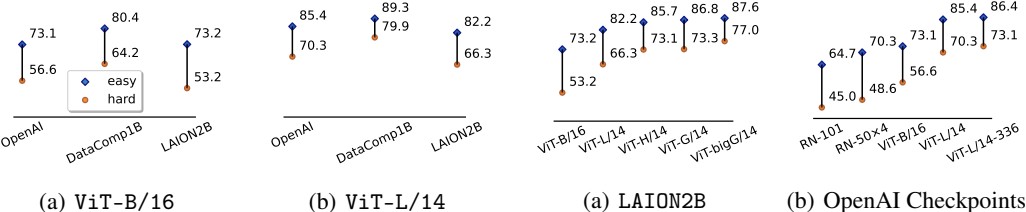

(a) `ViT-B/16`      (b) `ViT-L/14`      (a) `LAION2B`      (b) OpenAI Checkpoints

Figure 6: 1 vs. 1000 results for varying CLIP setups with different pre-train datasets.

Figure 7: 1 vs. 1000 results for varying CLIP setups with different backbones.

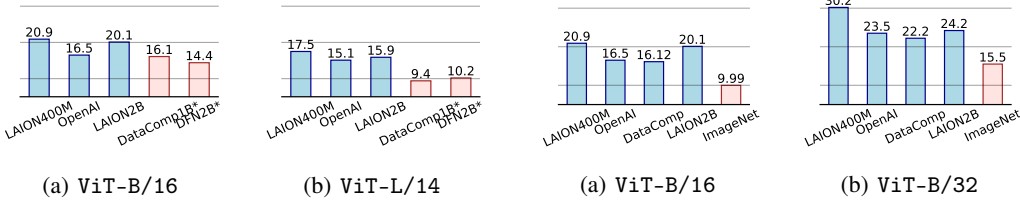

(a) `ViT-B/16`      (b) `ViT-L/14`      (a) `ViT-B/16`      (b) `ViT-B/32`

Figure 8: 1 vs. 1000 drops for varying CLIP setups with filtered and unfiltered pre-train data.

Figure 9: 1 vs. 1000 drops for varying training setups with CLIP and ImageNet supervision.

data from OpenAI, along with backbones of increasing scales. We observe a clear trend indicating that larger models exhibit better performance against spurious correlations. It may tell us that larger models possess stronger robustness, making them less prone to the shortcuts of spurious features.

**Data Quality Matters.** Moreover, we observe that the results obtained with DataComp- and DFN-trained CLIPs exhibit better performance and smaller drops across backbones, Figure 8 offers their comparisons. We notice that these datasets have been stringently filtered and thus possess high-quality data. It indicates that enhancing data quality is still a promising way to improve OOD generalization.

Our analysis focuses on absolute performance drop. In Appendix F, we strengthen our conclusions by incorporating the analysis based on effective robustness [5], where our findings still hold.

### 3.3 Evaluations for other Learning Paradigms

We extend our evaluations to broader families of models, including ImageNet-1K supervised models and more advanced LVLMs, such as MiniGPT4 and LLaVA.

**ImageNet Models**. We first extend our evaluations to include ImageNet models. The main results are summarized in Table 3. Moreover, Figure 9 further illustrates the accuracy drops of various CLIP models, in comparison to ImageNet models. Surprisingly, we find that ImageNet models are more robust to spurious features in `CounterAnimal`. This finding indicates that our `CounterAnimal` specifically characterizes the spurious features that are unique to CLIP configurations. Additionally, it indicates that spurious correlations in large-scale multi-modal data are distinct from that of the ImageNet scenarios which are widely used in conventional single-modal supervised learning. It further highlights the importance of our proposed dataset, which is especially suitable to study the spurious correlations for vision-language pre-training.

Table 3: The 1 vs. 1000 performance for ImageNet models `CounterAnimal`.

| backbone | easy | hard | drop |
|---|---|---|---|
| AlexNet | 59.56 | 39.24 | 20.31 |
| VGG-11 | 73.37 | 56.12 | 17.25 |
| VGG-13 | 75.33 | 58.43 | 16.90 |
| VGG-19 | 77.84 | 61.74 | 16.10 |
| RN-18 | 74.36 | 56.07 | 18.29 |
| RN-34 | 78.31 | 61.01 | 17.30 |
| RN-50 | 81.44 | 66.07 | 15.37 |
| RN-101 | 81.76 | 68.18 | 13.57 |
| ViT-B/16 | 84.97 | 74.98 | 9.99 |
| ViT-B/32 | 79.84 | 64.36 | 15.48 |
| ViT-L/16 | 83.74 | 72.69 | 11.05 |
| ViT-L/32 | 81.23 | 67.54 | 13.69 |
| ConvNext-S | 88.27 | 79.97 | 8.30 |
| ConvNext-B | 88.60 | 80.53 | 8.07 |
| ConvNext-L | 89.12 | 81.47 | 7.65 |

**Advanced LVLMs.** We further evaluate for more advanced LVLMs, which align CLIP visual encoders with advanced large language models like Vi-

Table 4: The 1 vs. 20 results of `CounterAnimal` for advanced LVLMs and several CLIP models. More results of CLIP models and ImageNet models can be found in Appendix F.

| LVLMs | easy | hard | drop |
|---|---|---|---|
| MiniGPT4-Viccuna7B | 47.99 | 39.73 | 8.26 |
| LLaVA1.5-7B | 40.06 | 30.09 | 9.97 |
| CLIP-LAION400M-ViT-L/14 | 80.90 | 63.31 | 17.59 |
| CLIP-OpenAI-ViT-L/14 | 85.38 | 70.28 | 15.10 |
| CLIP-DataComp1B-ViT-L/14 | 89.29 | 79.90 | 9.39 |
| CLIP-LAION2B-ViT-L/14 | 82.23 | 66.27 | 15.96 |
| CLIP-DFN2B-ViT-L/14 | 90.77 | 80.55 | 10.22 |

cuna [49]. To reduce inference costs, our evaluation follows the 1 vs. 20 setup. We summarize their results in Table 4, along with the 1 vs. 20 results for several CLIP models (cf., Appendix F for more results). We further depict the full results in Figure 2(b). As we can see, these advanced LVLMs have lower performance yet smaller drops, but the spurious features in `CounterAnimal` still impact them.

## 4 Understanding Why CLIPs Rely on Spurious Features

To better understand the observed phenomena in Section 3, we present a theoretical analysis of why the CLIP models rely on spurious features. We begin by establishing the setup for analyzing multi-modal contrastive learning following [9].

**Definition 1** (Multi-modal Dataset). *Consider $n$ image-text pairs $\{(\boldsymbol{x}_I^i, \boldsymbol{x}_T^i)\}_{i=1}^n$, both image $\boldsymbol{x}_I^i$ and text $\boldsymbol{x}_T^i$ are generated from the latent factor $\boldsymbol{z}_i$, where $\boldsymbol{z} = [z_{inv}, z_{spu}] \in \mathbb{R}^2$ is composed of an invariant feature $z_{inv} \sim \mathcal{N}(\mu_{inv}y, \sigma_{inv}^2)$ and a spurious feature $z_{spu} \sim \mathcal{N}(\mu_{spu}a, \sigma_{spu}^2)$ with $\Pr(a = y) = p_{spu}$ otherwise $a = -y$. $y$ is the label uniformly drawn from $\{-1, 1\}$. The training data $\mathcal{D}^{tr}$ is drawn with $\frac{1}{2} \leq p_{spu} \leq 1$ and OOD data $\mathcal{D}^*$ is drawn with a $p_{spu} = \frac{1}{2}$.*

We employ two linear encoders: $g_I : \mathbb{R}^{d_I} \to \mathbb{R}^h$ for the image modality and $g_T : \mathbb{R}^{d_T} \to \mathbb{R}^h$ for the text modality, implemented as $g_I(\boldsymbol{x}_I) = \boldsymbol{W}_I \boldsymbol{x}_I$ and $g_T(\boldsymbol{x}_T) = \boldsymbol{W}_T \boldsymbol{x}_T$ with $\boldsymbol{W}_I \in \mathbb{R}^{h \times d_I}$ and $\boldsymbol{W}_T \in \mathbb{R}^{h \times d_T}$. The encoders are trained through the linearized contrastive loss [9, 50] that mimics the CLIP dynamics:

$$\mathcal{L}_{\text{CLIP}} = \frac{1}{2n(n-1)} \sum_i \sum_{j \neq i} (s_{ij} - s_{ii}) + \frac{1}{2n(n-1)} \sum_i \sum_{j \neq i} (s_{ji} - s_{ii}) + \frac{\rho}{2} ||\boldsymbol{W}_I^T \boldsymbol{W}_T||_F^2, \quad (1)$$

where $s_{ij} = g_I(\boldsymbol{x}_I^i)^T g_T(\boldsymbol{x}_T^j)$ is the similarity with respect to the $i$-th image and $j$-th text representations. Once the CLIP $(g_I, g_T)$ has been trained, the performance will be measured in a zero-shot manner by matching the most similar caption with the corresponding object name filled in, such as "a photo of <object label>" [2]. Intuitively, once the model focuses more on invariant features, it will have a better zero-shot classification accuracy across different distributions. Nevertheless, in the following theorem, we justify that CLIP remains to learn to use spurious features, aligning with our experimental observations on the `CounterAnimal` dataset.

**Theorem 1.** *Given a multi-modal dataset (Def. 1) with suitable variance in the features $\sigma_{inv} = \Theta(1) > \sigma_{spu}$, and spurious features with a large spurious correlation $p_{spu} = 1 - o(1)$, an overparameterized CLIP model where $n = \omega(1), d_M = \Omega(n)$ and $d_T = \Omega(n)$, if the spurious features (e.g., backgrounds of the image) takes up a relatively large amount of the image $\mu_{spu} \geq \frac{\sigma_{inv}^2 + 2}{2} \geq \mu_{inv} = 1$, then with a high probability of at least $1 - O(\frac{1}{poly(n)}) = 1 - o(1)$, the CLIP model achieves a large error in zero-shot accuracy in the OOD test data where $a \neq y$:*

$$Err(g_I, g_T) \geq 1 - \Phi(\kappa_1) - o(1),$$

*and a small error in the OOD test data where $a = y$:*

$$Acc(g_I, g_T) \geq 1 - \Phi(\kappa_2) - o(1),$$

*where $\kappa_1 = \frac{\sigma_{inv}^2 + 2 - 2\mu_{spu}p_{spu}}{\sqrt{(1+\sigma_{inv}^2)^2 \sigma_{inv}^2 + (2\mu_{spu}p_{spu}-1)^2 \sigma_{spu}^2}}$, $\kappa_2 = \frac{-2\mu_{spu}p_{spu} - \sigma_{inv}^2}{\sqrt{(1+\sigma_{inv}^2)^2 \sigma_{inv}^2 + (2\mu_{spu}p_{spu}-1)^2 \sigma_{spu}^2}}$ and $\Phi$ denotes the CDF of a standard normal distribution.*

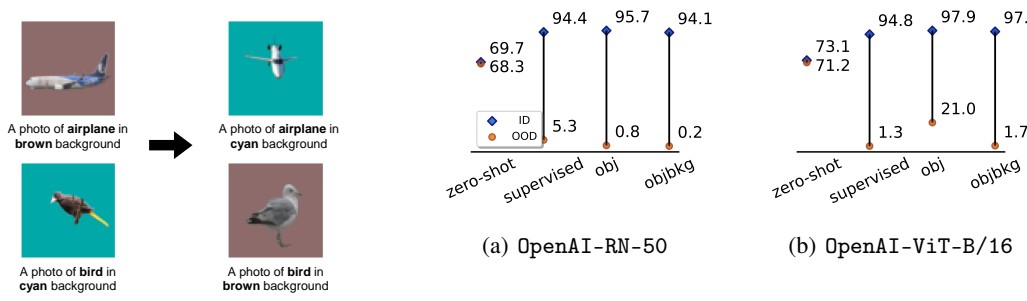

Figure 10: Illustration of `ColoredCOCO`.

(a) `OpenAI-RN-50`

(b) `OpenAI-ViT-B/16`

Figure 11: CLIP performance on `ColoredCOCO`. "supervised" refers to supervised trained models, while "obj" and "objbkg" refer to using different prompts to fine-tune CLIPs.

We leave more theoretical details as well as the proof to Appendix D due to space limit. Intuitively, Theorem 1 implies that once there exists a relatively strong correlation between the object captions and the parts of image backgrounds, CLIP will learn to align the backgrounds, i.e., spurious features, with object captions. Although our theory discusses a simplistic case of one invariant and one spurious feature, there could exist more features describing the objects and even more features describing the backgrounds. CLIP will fail to robustly align the visual features of objects to its captions, once there exists a spurious correlation between any of the background features with the object caption. Our theory is the first to provably demonstrate the drawbacks of CLIPs in OOD generalization, providing the foundation for future developments tackling the issue.

To verify our theory, we construct multi-modal datasets named `ColoredCOCO` following [51]. It contains 9 classes and the spurious correlation in the training part is $80\%$, i.e., each class has a correlation of $80\%$ to a specific biased color and $20\%$ uniformly correlates to 10 different randomly chosen colors, cf., Figure 10. The OOD datasets are built with classes randomly correlating to other 8 biased colors. We consider two prompts with different descriptiveness: a) obj: "a photo of <object label>" and b) objbkg: "a photo of <object label> in <color label> background", with either objects or both objects and backgrounds.

We tune the pre-trained CLIP models using the CLIP objective, which has been shown to be most robust to distribution shifts [52]. In addition, we also incorporate the baseline of full fine-tuning with a new MLP onto the image encoder using the ERM objective. As shown in Figure 11, fine-tuning with CLIP objective based on neither of the prompts provides any non-trivial robustness against the vanilla full fine-tuning. The results further verify our theory. Nevertheless, the degraded robustness of CLIP could also be caused by the weak language understanding capability of the BERT encoder in the CLIP. To this end, we also conduct additional experiments with a perfect language encoder setting. The results are given in Appendix D.4. Nevertheless, we find that CLIP still performs similarly to ERM and is prone to distribution shifts even with perfect captions.

## 5 Conclusion

In this paper, we highlight biases in previous evaluations for assessing the robustness of CLIP models, primarily relying on ImageNet variants. Such improper benchmarking would cause illusions that CLIP models seem to resolve spurious correlations, particularly in comparison with ImageNet models. It motivates us to craft the new testset, named `CounterAnimal`, which is specifically designed to probe the natural spurious correlations between animal and their backgrounds. The spuriousness captured by `CounterAnimal` is general across different CLIP setups and exerts relatively small impacts on the ImageNet benchmarks, thereby specifically capturing the spurious correlations within CLIP setups. Our experiments on `CounterAnimal` show that many conventional strategies, e.g., increasing backbone scales and improving pre-train data quality, remain effective in enhancing the robustness of CLIP models. Moreover, we present a theoretical analysis for the reasons of the CLIP objective to learn biases. Overall, we provide a platform for future developments of more advanced and robust CLIP and vision-language models, and we hope our presented experiments can offer a sober look at the robustness of CLIP models to spurious correlations.

## Acknowledgments

The authors would like to express their sincere gratitude to the anonymous reviewers and the area chairs for their thorough review and constructive feedback. Their insightful comments and valuable suggestions have significantly enhanced the quality and clarity of this manuscript. We deeply appreciate their time and effort in helping us improve our work.

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

# A Broader Impacts and Limitations

The current community often overestimates the robustness of CLIP models, largely due to the potentially misleading reliance on ImageNet variants for testing. To address this issue, we propose a new testset, named `CounterAnimal`, specifically tailored for CLIP models. Our findings indicate that CLIP models may not be as robust to distribution shifts as previously believed. Our dataset serves as a real-world benchmark, poised to be meaningful for the subsequent works to understand and enhance CLIP concerning their OOD robustness. For real-world applications, the understanding of spurious correlations for CLIP is also critical. We raise practical concerns when deploying CLIP models, which pertain to fairness and potential biases that may arise from inherent spurious correlations. We also present general strategies and theoretical analysis to understand the spurious correlations within CLIP models, which may motivate subsequent works to further enhance CLIP in real-world applications. However, although our dataset reaches the bar as a standard evaluation dataset, its research potential can be further benefited from expanding the scale of our dataset, diversifying the raw data sources beyond iNaturalist, broadening the semantic scope beyond animal classes, and studying other testbeds beyond the ImageNet benchmarks. In the future, we will extend our focus beyond animal subjects and include a wider array of high-quality data that are suitable for evaluating the robustness of CLIP and more advanced LVLMs.

# B Dataset Composition

We release our dataset `CounterAnimal` structured as follows:

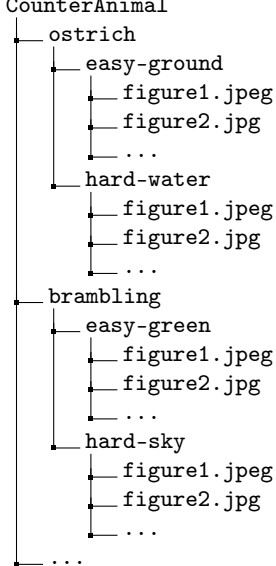

```
CounterAnimal
├── ostrich
│   ├── easy-ground
│   │   ├── figure1.jpeg
│   │   ├── figure2.jpg
│   │   └── ...
│   └── hard-water
│       ├── figure1.jpeg
│       ├── figure2.jpg
│       └── ...
├── brambling
│   ├── easy-green
│   │   ├── figure1.jpeg
│   │   ├── figure2.jpg
│   │   └── ...
│   └── hard-sky
│       ├── figure1.jpeg
│       ├── figure2.jpg
│       └── ...
└── ...
```

Overall, the `CounterAnimal` dataset is organized by the object names. The data therein are further separated into two parts, i.e., the `easy` and `hard` groups, where the background name is also provided for each sub-directory. By evaluating accuracy with respect to the `easy` and `hard` groups, one can quantify the impacts of the spurious correlations captured by `CounterAnimal`. We further summarize the ImageNet animal objects as well as the group names for the `easy` and `hard` groups in Table 5.

# C Experimental Configurations

In this section, we provide more details about our experimental configurations.

## C.1 Hardware Configurations

All experiments are realized by Pytorch 1.81 with CUDA 11.1, using machines equipped with GeForce RTX 3090 GPUs and AMD Threadripper 3960X Processors.

Table 5: The object names and the background names in the `CounterAnimal` dataset. The full names of labels are presented following the fashion of the ImageNet-1K dataset.

| ID | object label | easy | hard | ID | object label | easy | hard | ID | object label | easy | hard |
|---|---|---|---|---|---|---|---|---|---|---|---|
| 1 | ostrich, struthio camelus | ground | water | 2 | brambling, Fringilla montifringilla | grass | sky | 3 | bulbul | sky | grass |
| 4 | water ouzel, dipper | water | ground | 5 | vulture | sky | tree | 6 | bullfrog, rana catesbeiana | water | ground |
| 7 | loggerhead, loggerhead turtle, caretta caretta | water | ground | 8 | box turtle, box tortoise | grass | earth | 9 | common iguana, iguana iguana iguana | earth | shrub |
| 10 | whiptail, whiptail lizard | earth | human | 11 | agama | rock | tree | 12 | african crocodile, nile crocodile, crocodylus niloticus | earth | grass |
| 13 | hognose snake, puff adder, sand viper | earth | grass | 14 | king snake kingsnake | earth | grass | 15 | garter snake grass snake | grass | earth |
| 16 | water snake | water | ground | 17 | harvestman, daddy longlegs, Phalangium opilio | shrub | rock | 18 | scorpion | indoor | outdoor |
| 19 | tarantula | sand | grass | 20 | centipede | indoor | grass | 21 | black grouse | grass | tree |
| 22 | ptarmigan | snow | grass | 23 | prairie chicken, prairie grouse, prairie fowl | grass | snow | 24 | sulphur-crested cockatoo, Kakatoe galerita, cacatua galerita | tree | grass |
| 25 | black swan, cygnus atratus | water | ground | 26 | echidna, spiny anteater, anteater | grass | tree | 27 | black stork ciconia nigra | grass | sky |
| 28 | flamingo | water | sky | 29 | bittern | grass | tree | 30 | pelican | water | sky |
| 31 | sea lion | sand | water | 32 | african hunting dog, hyena dog, cape hunting dog, lycaon pictus | grass | tree | 33 | hyena, hyaena | grass | road |
| 34 | red fox, vulpes vulpes | grass | road | 35 | arctic fox, white fox, alopex lagopus | snow | grass | 36 | jaguar, panther, Panthera onca, Felis onca | water | tree |
| 37 | lion, king of beasts, panthera leo | grass | tree | 38 | cheetah, chetah, acinonyx jubatus | grass | tree | 39 | ice bear, polar bear, ursus maritimus, thalarctos maritimus | snow | grass |
| 40 | dung beetle | earth | human | 41 | cicada, cicala | tree | human | 42 | beaver | water | grass |
| 43 | bighorn, bighorn sheep, cimarron | grass | rock | 44 | mink | grass | water | 45 | otter | water | tree |

## C.2 Candidate Label Space

We consider two different label spaces of candidate labels: a) using the full ImageNet-1K class names and b) using the top-20 most confusing classes for more computing-intensive models like MiniGPT4. It leads to the following two evaluation setups, i.e., the 1 vs. 1000 setup and the 1 vs. 20 setup.

**1 vs. 1000 Setup.** As a default option, we use the full label space of the ImageNet-1K dataset, which is suitable given that the object labels for `CounterAnimal` all belong to that of the ImageNet-1K dataset. Furthermore, this choice also reflects a more realistic situation in the open world, where we have a vast number of candidate labels and the failure cases of LVLMs are common.

**1 vs. 20 Setup.** To suit more advanced LVLMs of which the inference costs are much higher than CLIP models, we constrain the sizes of candidate label space for each class. Specifically, based on `CLIP-LAION400M-ViT-B/32`, we select the top-20 most confusing labels, which is calculated by the average cosine similarity for both the `easy` and `hard` groups.

## C.3 Evaluation Metrics

Now, we discuss the evaluation metrics. Typically, they are applied to the `easy` and `hard` groups separately when we evaluate the robustness of various models.

**Class-wise Accuracy.** We are interested in the effects of spurious features for each class. Therefore, we calculate the prediction accuracy specifically for photos within each class. It can be referred to as the class-wise accuracy, which is given by

$$\texttt{ACC}(\texttt{label}) = \frac{1}{|\mathcal{I}_{\texttt{label}}|} \sum_{i \in \mathcal{I}_{\texttt{label}}} \mathbf{1}\{\hat{y}_i = \texttt{label}\},$$

where $\mathcal{I}_{\texttt{label}}$ is the indices of photos belonging to `label` and $\hat{y}_i$ is the predicted label for the $i$-th image. The class-wise accuracy reflects the class-level model reliability against spurious correlations.

**Average Accuracy.** Upon the class-wise accuracy, we can calculate the average performance of models, namely,

$$\texttt{ACC} = \frac{1}{|\mathcal{L}|} \sum_{\texttt{label} \in \mathcal{L}} \texttt{ACC}(\texttt{label}).$$

Compared to the conventional average accuracy, i.e., $\frac{1}{|\mathcal{I}|} \sum_{i \in \mathcal{I}} \mathbf{1}\{\hat{y}_i = \texttt{gt}\}$ with $\mathcal{I}$ the image indices and `gt` the true labels, our definition of the average accuracy further offsets the impact of class

Table 6: Adopted versions of CLIP checkpoints employed in our main experiments.

| backbone | pre-train dataset | checkpoint |
|---|---|---|
| ViT-B/16 | LAION400M | E31 |
| ViT-B/16 | LAION2B | S34B B88K |
| ViT-B/16 | DataComp1B | XL S13B B90K |
| ViT-B/32 | LAION400M | E31 |
| ViT-B/32 | LAION2B | S34B B79K |
| ViT-B/32 | DataComp1B | XL S13B B90K |
| ViT-L/14 | LAION400M | E31 |
| ViT-L/14 | LAION2B | S32B B82K |
| ViT-L/14 | DataComp1B | XL S32B B82K |
| ViT-H/14 | LAION2B | S32B B79K |
| ViT-G/14 | LAION2B | S34B B88K |
| ViT-bigG/14 | LAION2B | S34B B160K |
| ConvNext-B | LAION400M | S13B B51K |
| ConvNext-BW | LAION2B | S13B B82K |

imbalance. We default to using the average accuracy, and present the results without balancing in Tables 17-18 for CLIP and ImageNet models.

**Accuracy Drop.** To quantify the spurious correlations that make CLIP models fail, we measure the performance drop when moving from the easy and hard groups. At the class level, the accuracy drop is defined as the class-wise accuracy of easy minuses that of hard. At the dataset level, it is the average value for the class-level accuracy drop.

### C.4   Evaluation Details of MiniGPT4 and LLaVA

To evaluate LVLMs with a backend of language models, we follow the common practice that constructs questions to prompt LVLMs [29, 30]. Specifically, we construct the question as:

```
What is the main object in the image?
```

and then calculate the language modeling loss with respect to the answer:

```
A <object name>
```

for each ImageNet class name. Meanwhile, we also try another question prompt that is widely used in training MiniGPT4 and LLaVA [30, 53]:

```
Describe this image in detail.
```

while the performance will generically decrease. In addition, when we switch to the object-centric evaluation protocol as [33]:

```
Is there a <object name> in the image?
```

or

```
Is this image a photo of <object name>?
```

and evaluate the answer with Yes for each class, we observe a severe performance decrease as LVLMs easily hallucinate the objects. If we strictly follow the evaluation metrics of [33] by simply fetching the answers instead of comparing the losses, there exist lots of hallucinated objects by LVLMs in our dataset.

### C.5   CLIP Naming Rules

For the CLIP checkpoints, we adopt the naming rule of "CLIP-<dataset>-<backbone>", where <dataset> is the name of pre-train datasets and <backbone> is the specific name of backbone models. For example, CLIP-LAION400M-ViT-B/32 indicates the ViT-B/32 model CLIP-trained on LAION400M. Different training setups are considered in OpenCLIP, and the versions of the adopted checkpoints are summarized in Table 6. Moreover, in Table 15, we consider the results of checkpoints beyond the adopted ones.

# D   Theoretical Understanding of CLIP's Robustness to Spurious Features

We provide a more detailed setup and analysis in complementary to Section 4.

## D.1   Detailed Theoretical Setup

We begin by introducing more details about the data generation process following the literature [9, 50, 54].

**Definition 2** (Multi-modal Dataset). *Consider $n$ image-text pairs $\{(\boldsymbol{x}_I^i, \boldsymbol{x}_T^i)\}_{i=1}^n$, both image $\boldsymbol{x}_I^i \in \mathbb{R}^{d_I}$ and text $\boldsymbol{x}_T^i \in \mathbb{R}^{d_T}$ are generated from the underlying latent factor $\boldsymbol{z}_i \in \mathbb{R}^l$. The samples are generated as follows:*

- *$\boldsymbol{z} = [z_{inv}, z_{spu}] \in \mathbb{R}^2$ is composed of a invariant feature $z_{inv} \sim \mathcal{N}(\mu_{inv}y, \sigma_{inv}^2)$ and a spurious feature $z_{spu} \sim \mathcal{N}(\mu_{spu}a, \sigma_{spu}^2)$ with $\Pr(a = y) = p_{spu}$ otherwise $a = -y$, $y$ is the label uniformly drawn from $\{-1, 1\}$, $\mathcal{D}^{tr}$ is drawn with $1/2 \le p_{spu} \le 1$ while the OOD test data $\mathcal{D}^*$ is drawn uniformly with $p_{spu} = 1/2$.*

- *Given $\boldsymbol{z}$, the $\boldsymbol{x}$ at modality $M$ is generated via $\boldsymbol{x}_M = \boldsymbol{D}_M \boldsymbol{\mu}_M(\boldsymbol{z}) + \xi_M$, with $\boldsymbol{D}_M \in \mathbb{R}^{d_M \times l}$ and $\xi_M \sim \mathcal{N}(0, \sigma_\xi^2/d_m \boldsymbol{I}_{d_m})$. The matrix $\boldsymbol{D}_M \in \mathbb{R}^{d_m \times l}$ with $d_m > l$ is a matrix with orthonormal columns which can be considered as a dictionary matrix.*

With the definition, we can write every $\boldsymbol{z}^i = \begin{bmatrix} y^i + \eta_{1,i} \\ \mu_{spu}p_{spu} + \eta_{2,i} \end{bmatrix}$ where $\eta_{1,i}, \eta_{2,i}$ are two Gaussian variables in the definition.

**CLIP Training.** We employ two linear encoders $g_I : \mathbb{R}^{d_I} \to \mathbb{R}^h$ for the image modality and $g_T : \mathbb{R}^{d_T} \to \mathbb{R}^h$ for the text modality, implemented as $g^I(\boldsymbol{x}_I) = \boldsymbol{W}_I \boldsymbol{x}_I$ and $g_T(\boldsymbol{x}_T) = \boldsymbol{W}_T \boldsymbol{x}_T$ with $\boldsymbol{W}_I \in \mathbb{R}^{h \times d_I}$ and $\boldsymbol{W}_T \in \mathbb{R}^{h \times d_T}$, respectively. The encoders are trained through the linearized contrastive loss [9, 50] that mimics CLIP training dynamics:

$$
\begin{aligned}
\mathcal{L}_{\text{CLIP}} = &\frac{1}{2n(n-1)} \sum_i \sum_{j \ne i} (s_{ij} - s_{ii}) \\
&+ \frac{1}{2n(n-1)} \sum_i \sum_{j \ne i} (s_{ji} - s_{ii}) + \frac{\rho}{2} ||\boldsymbol{W}_I^T \boldsymbol{W}_T||_F^2,
\end{aligned}
\tag{2}
$$

where $s_{ij} = g_I(\boldsymbol{x}_I^i)^T g_T(\boldsymbol{x}_T^j)$ is the similarity with respect to the $i$-th image and $j$-th text representations, and $||\boldsymbol{W}_I^T \boldsymbol{W}_T||_F^2$ is the a regularization term with $\rho > 0$.

**Zero-shot Inference.** Once the CLIP model $(g_I, g_T)$ is trained, the performance will be measured in a zero-shot manner by matching the most similar caption such as 'a photo of {object name}' across different `object name` as class names. Meanwhile, one could also leverage several prompts and leverage the average text embeddings across the available prompts to facilitate the evaluation [2]. The prompt with respect to $y$ could be modeled as $\boldsymbol{p}_y = \boldsymbol{D}_T \mathbb{E}[\boldsymbol{z}^t|y]$, where $\boldsymbol{D}_T$ is the prompt transformation matrix. Then, the zero-shot accuracy of CLIP could be formalized as follows:

$$
\text{Acc}(g_I, g_T) = \mathbb{E}_{(\boldsymbol{x}, y)}[\boldsymbol{1}(\arg\max_{\hat{y}} g_I(\boldsymbol{x}_I)^T g_T(\boldsymbol{p}_{\hat{y}}), y)],
\tag{3}
$$

while the error is $\text{Err}(g_I, g_T) = 1 - \text{Acc}(g_I, g_T)$. Intuitively, once the model extracts more of the invariant features, it will have a better zero-shot classification accuracy across different distributions.

## D.2   Proof for Theorem 1

**Theorem 2** (Restatement of Theorem 1). *Given a multi-modal dataset (Def. 2) with suitable variance in the features $\sigma_{inv} = \Theta(1) > \sigma_{spu}$, and spurious features with a large spurious correlation $p_{spu} = 1 - o(1)$, an overparameterized CLIP model where $n = \omega(1), d_M = \Omega(n)$ and $d_T = \Omega(n)$, if the spurious features (e.g., backgrounds of the image) takes up a relatively large amount of the image $\mu_{spu} \ge \frac{\sigma_{inv}^2 + 2}{2} \ge \mu_{inv} = 1$, then with a high probability of at least $1 - O(\frac{1}{poly(n)}) = 1 - o(1)$, the CLIP model achieves a large error in zero-shot accuracy in the OOD test data where $a \ne y$:*

$$
Err(g_I, g_T) \ge 1 - \Phi(\kappa_1) - o(1),
$$

*and a small error in the OOD test data where $a = y$:*

$$Acc(g_I, g_T) \geq 1 - \Phi(\kappa_2) - o(1),$$

*where $\kappa_1 = \frac{\sigma_{inv}^2 + 2 - 2\mu_{spu}p_{spu}}{\sqrt{(1+\sigma_{inv}^2)^2\sigma_{inv}^2 + (2\mu_{spu}p_{spu}-1)^2\sigma_{spu}^2}}$, $\kappa_2 = \frac{-2\mu_{spu}p_{spu} - \sigma_{inv}^2}{\sqrt{(1+\sigma_{inv}^2)^2\sigma_{inv}^2 + (2\mu_{spu}p_{spu}-1)^2\sigma_{spu}^2}}$ and $\Phi$ denotes the CDF of a standard normal distribution.*

*Proof.* We will introduce some useful lemmas to help with our proof.

**Lemma 1** ([9]). *The minimizer of linearized CLIP loss $\boldsymbol{W}_I^{*T}\boldsymbol{W}_T^*$ satisfies the following with a probability of at least $1 - O(\frac{1}{poly(n)})$ such that,*

$$||\boldsymbol{W}_I^{*T}\boldsymbol{W}_T^* - \frac{1}{\rho}\boldsymbol{D}_I \begin{bmatrix} 1+\sigma_{inv}^2 & 2\mu_{spu}p_{spu} - 1 \\ 2\mu_{spu}p_{spu} - 1 & 1 + \sigma_{spu}^2 \end{bmatrix} \boldsymbol{D}_T^T||_2 \leq \frac{1}{\rho}O(\sqrt{\epsilon_0}),$$

*where $\epsilon_0 = O(\sqrt{\frac{\log n}{n}})$.*

Intuitively, the lemma indicates the importance of the training distribution, that the minimizer of CLIP will converge to the data characteristics of the latent features of the training distribution.

Then, consider the case where the model is inferred onto a test sample with $y = 1, a = -1$. Then, with the aforementioned lemma, we have

$$|\boldsymbol{x}_I^T\boldsymbol{W}_I^*\boldsymbol{W}_T^*\boldsymbol{x}_T^{\hat{y}} - \frac{1}{\rho}\boldsymbol{x}_I^T\boldsymbol{D}_I \begin{bmatrix} 1+\sigma_{inv}^2 & 2\mu_{spu}p_{spu} - 1 \\ 2\mu_{spu}p_{spu} - 1 & 1 + \sigma_{spu}^2 \end{bmatrix} \boldsymbol{D}_T^T\boldsymbol{x}_T^{\hat{y}}||_2 \leq ||\boldsymbol{x}_I||||\boldsymbol{x}_T^{\hat{y}}||\frac{1}{\rho}O(\sqrt{\epsilon_0})$$

$$\leq \frac{1}{\rho}O(\sqrt{\epsilon_0}\log n). \tag{4}$$

Then, notice that

$$\frac{1}{\rho}\boldsymbol{x}_I^T\boldsymbol{D}_I \begin{bmatrix} 1+\sigma_{inv}^2 & 2\mu_{spu}p_{spu} - 1 \\ 2\mu_{spu}p_{spu} - 1 & 1 + \sigma_{spu}^2 \end{bmatrix} \boldsymbol{D}_T^T\boldsymbol{x}_T^{\hat{y}} = \hat{y}((1+\eta_1)(1+\sigma_{inv}^2) + (-1+\eta_2)(2\mu_{spu}p_{spu}-1)). \tag{5}$$

When CLIP makes an incorrect prediction, we have

$$\boldsymbol{x}_I^T\boldsymbol{W}_I^*\boldsymbol{W}_T^*\boldsymbol{x}_T^{\hat{y}=1} < \boldsymbol{x}_I^T\boldsymbol{W}_I^*\boldsymbol{W}_T^*\boldsymbol{x}_T^{\hat{y}=-1}.$$

Then, we have

$$\frac{1}{\rho}\boldsymbol{x}_I^T\boldsymbol{D}_I \begin{bmatrix} 1+\sigma_{inv}^2 & 2\mu_{spu}p_{spu} - 1 \\ 2\mu_{spu}p_{spu} - 1 & 1 + \sigma_{spu}^2 \end{bmatrix} \boldsymbol{D}_T^T\boldsymbol{x}_T^{\hat{y}=1} - \frac{1}{\rho}O(\sqrt{\epsilon_0}\log n) <$$
$$\frac{1}{\rho}\boldsymbol{x}_I^T\boldsymbol{D}_I \begin{bmatrix} 1+\sigma_{inv}^2 & 2\mu_{spu}p_{spu} - 1 \\ 2\mu_{spu}p_{spu} - 1 & 1 + \sigma_{spu}^2 \end{bmatrix} \boldsymbol{D}_T^T\boldsymbol{x}_T^{\hat{y}=-1} - \frac{1}{\rho}O(\sqrt{\epsilon_0}\log n), \tag{6}$$

with Eq. 5 plugged in, denote $\epsilon_1 = O(\sqrt{\epsilon_0}\log n)$, we further have

$$-2\left[(1+\eta_1)(1+\sigma_{inv}^2) + (-1+\eta_2)(2\mu_{spu}p_{spu} - 1) - \epsilon_1\right] > 0. \tag{7}$$

Since $\eta_1(1+\sigma_{inv}^2) + \eta_2(2\mu_{spu}p_{spu} - 1)$ is a Gaussian variable follows the distribution of

$$\eta_1(1+\sigma_{inv}^2) + \eta_2(2\mu_{spu}p_{spu} - 1) \sim \mathcal{N}(0, (1+\sigma_{inv}^2)^2\sigma_{inv}^2 + (2\mu_{spu}p_{spu} - 1)^2\sigma_{spu}^2),$$

then, we have

$$\Pr(-2\left[(1+\eta_1)(1+\sigma_{inv}^2) + (-1+\eta_2)(2\mu_{spu}p_{spu} - 1) - \epsilon_1\right] > 0)$$

$$= \Pr_{v\sim\mathcal{N}(0,1)}(v > \frac{\sigma_{inv}^2 + 2 - 2\mu_{spu}p_{spu} + \epsilon_1}{\sqrt{(1+\sigma_{inv}^2)^2\sigma_{inv}^2 + (2\mu_{spu}p_{spu} - 1)^2\sigma_{spu}^2}})$$

$$= 1 - \Phi(\frac{\sigma_{inv}^2 + 2 - 2\mu_{spu}p_{spu} + \epsilon_1}{\sqrt{(1+\sigma_{inv}^2)^2\sigma_{inv}^2 + (2\mu_{spu}p_{spu} - 1)^2\sigma_{spu}^2}}), \tag{8}$$

Table 7: Comparison between CLIPs and standard supervised learning on `ColoredCOO`

| backbone | pre-train dataset | approach | in-distribution | out-of-distribution | drop |
|---|---|---|---|---|---|
| RN-50 | OpenAI | zero shot | 69.67 | 68.33 | 1.34 |
| RN-50 | OpenAI | obj | 95.67 | 0.78 | 94.89 |
| RN-50 | OpenAI | objbkg | 94.11 | 0.22 | 93.89 |
| RN-50 | OpenAI | supervised | 94.44 | 5.33 | 89.11 |
| ViT-B/16 | OpenAI | zero shot | 73.11 | 71.22 | 1.89 |
| ViT-B/16 | OpenAI | obj | 97.89 | 21 | 76.89 |
| ViT-B/16 | OpenAI | objbkg | 97.11 | 1.67 | 95.44 |
| ViT-B/16 | OpenAI | supervised | 94.78 | 1.33 | 93.45 |

where $\Phi$ is the CDF of the standard Gaussian distribution. Then, it suffices to know that the $\mathrm{Err}_{y=1,a=-1}$ is lower bounded by $\Phi\left(\frac{\sigma_{inv}^2+2-2\mu_{spu}p_{spu}+\epsilon_1}{\sqrt{(1+\sigma_{inv}^2)^2\sigma_{inv}^2+(2\mu_{spu}p_{spu}-1)^2\sigma_{spu}^2}}\right)$, which also applies to the case $y=-1, a=1$.

Similarly, given the case $y=a$, as the model fits the spurious feature, we could derive the lower bound for its Acc by leveraging the spurious features as $\Phi\left(\frac{-2\mu_{spu}p_{spu}-\sigma_{inv}^2}{\sqrt{(1+\sigma_{inv}^2)^2\sigma_{inv}^2+(2\mu_{spu}p_{spu}-1)^2\sigma_{spu}^2}}\right)$.  $\square$

### D.3   More Details on `ColoredCOO` Experiments

To further validate our theoretical results, we construct the `ColoredCOO` dataset following [51]. More specifically, `ColoredCOO` is constructed as follows:

- The dataset contains 9 classes of COCO objects. The spurious correlation in the trainset is $80\%$ such that each class has a correlation of $80\%$ to a specific biased color and $20\%$ uniformly correlates to 10 sufficiently different randomly chosen colors.
- The OOD testsets are constructed with classes randomly correlating to 8 biased colors different from the one correlated in the training set.

Then, we further generate two prompts for each sample:

1. obj: `a photo of <object label>`;
2. objbkg: `a photo of <object label> in <color label> background`

We tune the pre-trained CLIP models using the CLIP objective based on the OpenCLIP library. We consider the learning rate of $\{1e-3, 1e-4, 1e-5\}$, with a weight decay of $\{1e-1, 1e-3, 1e-5\}$, and a warmup of $\{0, 1000\}$ steps. We select the model according to the best in-distribution test performance. The detailed results are given in Table 7. As we can see, the CLIPs finetuned using either the CLIP objective or the standard supervised training both exhibit high sensitivity to the spurious features.

### D.4   More Details on `MultiColoredMNIST` Experiments

One possible explanation for the failure of CLIP objective in `ColoredCOCO` is that, the language encoder of the CLIP models may not understand the captions well. Therefore, we further construct a new setup called `MultiColoredMNIST`, where each image contains only the digit information from MNIST dataset and the color information. Therefore, we can directly derive the one hot encoding for all of the useful factors in the dataset.

**Data.** We consider a multi-class classification setting with a number of classes no less than 2. The objects are the

- Training data: Fix two class (0/1) and color (r/g), they are spurious correlated by a correlation $p_{spu}$. The invariant feature's correlation with labels is $p_{inv}$.
- Test data (Rand): All classes and the colors are randomly correlated, given $k$ class, $p_{spu} = 1/k$.
- Test data (Rev): All classes and the colors are reversely correlated, $p_{spu}$ is 10% 0/1 classes and $1/k$ for others.

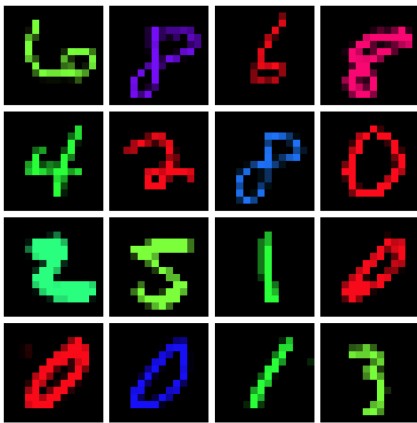

Figure 12: Examples of `MultiColoredMNIST` dataset.

In Figure 12, we give some examples for the `MultiColoredMnist` dataset.

**Experimental setting.** We compare the standard supervised training and CLIP. To avoid noises or information loss in encoding language modality, we consider the perfect language supervision for a single model. Given a batch of image and caption representations $\{(\boldsymbol{h}^{x_i}, \boldsymbol{h}^{c_i})\}_i^B$, for a image-caption pair, the CLIP objective aims to

$$\max(\boldsymbol{M}_x \boldsymbol{h}^{x_i} \cdot \boldsymbol{M}_c \boldsymbol{h}^{c_i}) - (\boldsymbol{M}_x \boldsymbol{h}^{x_j} \cdot \boldsymbol{M}_c \boldsymbol{h}^{c_j}), \forall i \neq j, \tag{9}$$

where $\boldsymbol{M}_x \in \mathbb{R}^{d \times h_x}$ and $\boldsymbol{M}_c^{d \times h_c}$ are the learnable projection layers for image and caption representations. Assuming the perfect language encoding as the one-hot encoding for all possible object and background appearance $\boldsymbol{h}^{c_i} \in [0, 1]^{|\mathcal{O}| + |\mathcal{B}|}$, and $\boldsymbol{M}_c$ can simply be an identity matrix, then Eq. 9 can be considered as a classification task for objects and backgrounds respectively:

$$\max \mathrm{CE}(\boldsymbol{M}_c^T \boldsymbol{M}_x \boldsymbol{h}^{x_i}, \boldsymbol{h}^{c_i}), \tag{10}$$

where the labels are simply the one-hot encodings of the objects and the backgrounds, and the classifier is $\boldsymbol{M}_c^T \boldsymbol{M}_x$. For the `MultiColoredMNIST` task where there is only one object and background (i.e., color), to implement Eq. 10, we only need to construct an additional classification head for the background. Given the aforementioned setup, we conduct experiments comparing CLIP-based contrastive learning to the standard supervised learning. The results are given in Table 8. As we can see, both contrastive learning and supervised learning perform similarly across different numbers of classes and bias degrees.

Table 8: Comparison of standard supervised learning and contrastive learning on MultiColoredMNIST dataset.

| # classes | # samples | $p_{inv}$ | $p_{spu}$ | train method | class 0/1 (Rand) | class 0/1 (Rev.) | rest class |
|---|---|---|---|---|---|---|---|
| 2 | 10,610 | 0.9 | 0.75 | Contrastive | 87.42±0.79 | 81.87±1.86 | n/a |
| 2 | 10,610 | 0.9 | 0.75 | Supervised | 86.44±0.90 | 80.22±1.73 | n/a |
| 2 | 10,610 | 0.9 | 0.9 | Contrastive | 71.56±1.79 | 50.08±3.97 | n/a |
| 2 | 10,610 | 0.9 | 0.9 | Supervised | 71.62±1.58 | 50.13±3.24 | n/a |
| 2 | 10,610 | 0.75 | 0.75 | Contrastive | 65.06±2.21 | 43.18±3.78 | n/a |
| 2 | 10,610 | 0.75 | 0.75 | Supervised | 65.01±1.68 | 43.76±3.44 | n/a |
| 2 | 10,610 | 0.75 | 0.9 | Contrastive | 53.73±1.08 | 16.42±1.74 | n/a |
| 2 | 10,610 | 0.75 | 0.9 | Supervised | 53.89±0.96 | 17.14±1.88 | n/a |
| 3 | 15,578 | 0.9 | 0.75 | Contrastive | 85.86±0.70 | 81.88±0.52 | 88.33±1.48 |
| 3 | 15,578 | 0.9 | 0.75 | Supervised | 85.03±1.25 | 79.20±1.91 | 88.03±1.10 |
| 3 | 15,578 | 0.9 | 0.9 | Contrastive | 69.05±2.26 | 45.55±4.52 | 88.60±1.20 |
| 3 | 15,578 | 0.9 | 0.9 | Supervised | 68.29±1.37 | 44.74±3.50 | 88.43±0.89 |
| 3 | 15,578 | 0.75 | 0.75 | Contrastive | 61.57±2.86 | 37.76±2.81 | 68.84±3.53 |
| 3 | 15,578 | 0.75 | 0.75 | Supervised | 59.51±2.28 | 36.66±2.06 | 68.75±2.58 |
| 3 | 15,578 | 0.75 | 0.9 | Contrastive | 42.47±2.48 | 7.08±1.10 | 71.07±3.01 |
| 3 | 15,578 | 0.75 | 0.9 | Supervised | 41.60±1.67 | 8.18±0.95 | 71.89±1.55 |
| 5 | 25,538 | 0.9 | 0.75 | Contrastive | 86.06±0.56 | 82.41±0.77 | 88.30±0.39 |
| 5 | 25,538 | 0.9 | 0.75 | Supervised | 85.60±0.74 | 80.99±0.99 | 87.76±0.57 |
| 5 | 25,538 | 0.9 | 0.9 | Contrastive | 71.78±0.77 | 44.66±4.02 | 88.15±0.42 |
| 5 | 25,538 | 0.9 | 0.9 | Supervised | 70.73±1.41 | 43.47±4.01 | 87.80±0.59 |
| 5 | 25,538 | 0.75 | 0.75 | Contrastive | 61.15±1.10 | 33.97±3.70 | 71.88±0.79 |
| 5 | 25,538 | 0.75 | 0.75 | Supervised | 57.69±1.29 | 33.66±3.18 | 68.75±0.91 |
| 5 | 25,538 | 0.75 | 0.9 | Contrastive | 35.37±1.70 | 4.60±0.45 | 72.47±0.58 |
| 5 | 25,538 | 0.75 | 0.9 | Supervised | 34.82±1.97 | 5.44±0.70 | 69.38±0.59 |
| 6 | 30,044 | 0.9 | 0.75 | Contrastive | 85.76±0.74 | 81.87±1.41 | 86.58±0.54 |
| 6 | 30,044 | 0.9 | 0.75 | Supervised | 85.84±0.81 | 81.81±1.27 | 86.29±0.47 |
| 6 | 30,044 | 0.9 | 0.9 | Contrastive | 70.99±2.39 | 40.07±10.53 | 86.57±0.49 |
| 6 | 30,044 | 0.9 | 0.9 | Supervised | 70.97±2.45 | 40.63±9.81 | 86.25±0.52 |
| 6 | 30,044 | 0.75 | 0.75 | Contrastive | 62.05±1.18 | 32.70±4.50 | 70.76±0.40 |
| 6 | 30,044 | 0.75 | 0.75 | Supervised | 59.49±1.26 | 33.94±3.69 | 67.91±0.81 |
| 6 | 30,044 | 0.75 | 0.9 | Contrastive | 38.96±2.55 | 4.71±0.56 | 70.65±0.40 |
| 6 | 30,044 | 0.75 | 0.9 | Supervised | 35.85±2.27 | 4.87±0.71 | 68.36±0.91 |
| 8 | 40,170 | 0.9 | 0.75 | Contrastive | 84.81±0.86 | 80.54±1.27 | 86.43±0.40 |
| 8 | 40,170 | 0.9 | 0.75 | Supervised | 85.49±0.67 | 81.47±1.08 | 86.78±0.39 |
| 8 | 40,170 | 0.9 | 0.9 | Contrastive | 71.75±1.65 | 39.85±8.81 | 86.34±0.36 |
| 8 | 40,170 | 0.9 | 0.9 | Supervised | 72.82±1.37 | 41.36±7.19 | 86.78±0.39 |
| 8 | 40,170 | 0.75 | 0.75 | Contrastive | 63.73±1.96 | 31.46±7.20 | 71.08±0.57 |
| 8 | 40,170 | 0.75 | 0.75 | Supervised | 62.22±2.00 | 33.12±6.54 | 70.58±0.63 |
| 8 | 40,170 | 0.75 | 0.9 | Contrastive | 43.91±2.36 | 5.11±0.68 | 70.76±0.60 |
| 8 | 40,170 | 0.75 | 0.9 | Supervised | 40.39±2.82 | 5.28±0.92 | 70.43±0.64 |
| 10 | 50,000 | 0.9 | 0.75 | Contrastive | 84.52±0.77 | 80.42±1.70 | 85.19±0.27 |
| 10 | 50,000 | 0.9 | 0.75 | Supervised | 85.10±0.67 | 81.83±0.97 | 86.11±0.15 |
| 10 | 50,000 | 0.9 | 0.9 | Contrastive | 73.79±1.43 | 48.02±5.50 | 85.18±0.34 |
| 10 | 50,000 | 0.9 | 0.9 | Supervised | 74.97±1.69 | 52.09±5.72 | 85.96±0.24 |
| 10 | 50,000 | 0.75 | 0.75 | Contrastive | 65.31±1.43 | 32.31±6.73 | 69.67±0.53 |
| 10 | 50,000 | 0.75 | 0.75 | Supervised | 66.00±1.52 | 36.35±5.59 | 70.27±0.30 |
| 10 | 50,000 | 0.75 | 0.9 | Contrastive | 48.03±1.56 | 5.53±1.25 | 69.13±0.47 |
| 10 | 50,000 | 0.75 | 0.9 | Supervised | 46.83±1.33 | 5.72±1.35 | 69.92±0.37 |

# E    Ablation Studies

In this section, we present ablation studies to further validate the feasibility of our data curation process.

**Biasing to ImageNet Setups.** We follow the same curation procedure while using ImageNet models (i.e., ResNet50-ImageNet) to construct `easy` and `hard` splits, where we name the corresponding dataset as `CounterAnimal-I`. We present the results of CLIP and ImageNet models on `CounterAnimal-I` in Tables 9-10, respectively. The effective robustness is further shown in Figure 13. Contrary to the observations within original `CounterAnimal`, CLIP models can demonstrate better robustness against spurious features within `CounterAnimal-I`. It aligns with our expectation since different training data (e.g., LAION for CLIPs, and ImageNet for ImageNet models) follow different distributions and naturally contain different spurious features. It also demonstrates the generality of our data curation method to reveal the spurious features for different kinds of the models. We also list the background names for `easy` and `hard` splits with respect to some of the selected classes in Table 11. As observed, using different models to split data will capture very different spurious features. It highlights the necessity to curate an OOD testset for CLIP models, as CLIP models learn different spurious features than ImageNet models.

**Correctness vs. Frequency.** We further explain why `easy` and `hard` examples can characterize spurious features within CLIP setups. In general, spurious features can be caused by biases inherent in the data distribution concerning backgrounds. For example, for the animal class of `ice bear`, the background of `ice` is more common than other backgrounds, such as `grass`, thus causing spurious correlations learned by CLIP models. Therefore, we investigate in terms of the background frequency, employing the searching tool of `Have I Been Trained` [4] that can retrieve images from LAION5B closely matching a given class name. We examine 10 animal classes as our case studies. For each class, we collect the top 100 most relevant images and tally the occurrences of the backgrounds of our consideration. It is important to note that our counting process excludes cartoon images, irrelevant photos, corrupted photos, and those featuring multiple distinct animal subjects or ambiguous backgrounds. The results are summarized in Table 12. As we can see, in general, the spurious features captured align with our conjecture that hard examples contain uncommon backgrounds in the CLIP training data, e.g., LAION5B, further justifying the feasibility of our `CounterAnimal` in assessing the robustness of CLIP models in real-world situations.

---

[4]`https://haveibeentrained.com`

Table 9: The 1 vs. 1000 results for CLIP checkpoints on the `CounterAnimal-I` dataset.

| backbone | pre-train dataset | easy | hard | drop |
|---|---|---|---|---|
| RN-50 | OpenAI | 60.90 | 42.56 | 18.34 |
| RN-101 | OpenAI | 61.22 | 40.25 | 20.97 |
| RN-50×4 | OpenAI | 64.40 | 47.85 | 16.55 |
| RN-50×16 | OpenAI | 72.00 | 57.65 | 14.35 |
| RN-50×64 | OpenAI | 81.41 | 68.36 | 13.05 |
| ViT-B/16 | LAION400M | 73.71 | 53.22 | 20.49 |
| ViT-B/16 | OpenAI | 73.46 | 56.56 | 17.10 |
| ViT-B/16 | DataComp1B* | 79.33 | 63.10 | 16.23 |
| ViT-B/16 | LAION2B | 68.66 | 52.13 | 16.53 |
| ViT-B/16 | DFN2B* | 83.39 | 68.75 | 14.64 |
| ViT-B/32 | LAION400M | 57.32 | 37.61 | 19.71 |
| ViT-B/32 | OpenAI | 66.95 | 47.12 | 19.84 |
| ViT-B/32 | DataComp1B* | 73.59 | 53.99 | 19.60 |
| ViT-B/32 | LAION2B | 67.37 | 47.64 | 19.73 |
| ViT-B/32-256 | DataComp1B* | 78.18 | 60.80 | 17.39 |
| ViT-L/14 | LAION400M | 77.96 | 60.85 | 17.11 |
| ViT-L/14 | OpenAI | 81.67 | 67.55 | 14.12 |
| ViT-L/14 | DataComp1B* | 88.87 | 77.06 | 11.82 |
| ViT-L/14 | LAION2B | 78.89 | 63.14 | 15.75 |
| ViT-L/14 | DFN2B* | 88.72 | 77.51 | 11.21 |
| ViT-L/14-336 | OpenAI | 84.09 | 71.62 | 12.47 |
| ViT-H/14 | LAION2B | 83.77 | 71.04 | 12.72 |
| ViT-H/14 | DFN5B* | 89.32 | 79.65 | 9.68 |
| ViT-H/14-384 | DFN5B* | 92.55 | 83.19 | 9.36 |
| ViT-G/14 | LAION2B | 84.46 | 68.16 | 16.31 |
| ViT-bigG/14 | LAION2B | 86.39 | 74.03 | 12.36 |
| ConvNext-B | LAION400M | 52.06 | 38.85 | 14.22 |
| ConvNext-BW | LAION2B | 57.19 | 38.74 | 18.45 |

Table 10: The 1 vs. 1000 performance for ImageNet models on the `CounterAnimal-I` dataset.

| backbone | easy | hard | drop |
|---|---|---|---|
| AlexNet | 59.27 | 31.87 | 27.40 |
| VGG-11 | 73.82 | 46.66 | 27.15 |
| VGG-13 | 74.32 | 48.08 | 26.24 |
| VGG-19 | 77.07 | 52.77 | 24.30 |
| RN-18 | 73.08 | 49.19 | 23.89 |
| RN-34 | 77.52 | 52.74 | 24.78 |
| RN-50 | 80.71 | 52.97 | 27.74 |
| RN-101 | 82.35 | 59.46 | 22.90 |
| ViT-B/16 | 85.06 | 66.64 | 18.42 |
| ViT-B/32 | 78.27 | 55.42 | 22.85 |
| ViT-L/16 | 83.65 | 64.03 | 19.63 |
| ViT-L/32 | 80.28 | 59.17 | 21.11 |
| ConvNext-S | 88.87 | 73.86 | 15.01 |
| ConvNext-B | 88.92 | 74.48 | 14.44 |
| ConvNext-L | 89.88 | 77.21 | 12.67 |

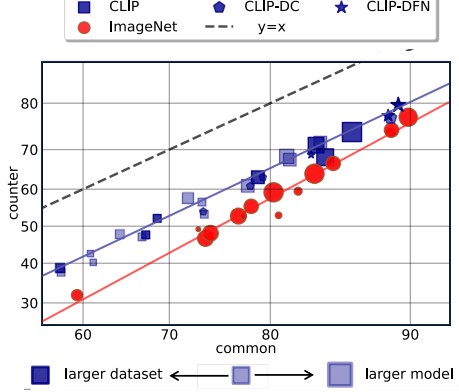

Figure 13: The `easy` verus `hard` performance (%) for CLIP and ImageNet models on `CounterAnimal-I`. The 1 vs. 1000 setup is considered.

Table 11: Selected animal object and background names in `CounterAnimal` and `CounterAnimal-I`. We bold the background names differently between `CounterAnimal` and `CounterAnimal-I`.

| object label | CounterAnimal | | CounterAnimal-I | |
|---|---|---|---|---|
| | easy | hard | easy | hard |
| Ostrich | ground | **water** | ground | **rock** |
| Brambling | grass | **sky** | grass | **water** |
| Bulbul | sky | **tree** | sky | **grass** |
| Vluture | sky | tree | sky | tree |
| Box turtle | grass | **earth** | grass | **water** |
| Common iguana | earth | shrub | earth | shrub |
| Whiptail | **earth** | **human** | **water** | **shurb** |
| Agama | rock | **tree** | rock | **grass** |
| Crocodile | earth | **grass** | earth | **tree** |

Table 12: The number of photos counted with respect to `easy` and `hard` backgrounds, based on the searching tool of `Have I Been Trained`.

| object label | easy | | hard | |
|---|---|---|---|---|
| | name | number | name | number |
| ostrich | ground | 30 | water | 0 |
| brambling | grass | 9 | sky | 17 |
| bulbul | sky | 5 | grass | 3 |
| water ouzel | water | 31 | ground | 4 |
| bullfrog | water | 28 | ground | 19 |
| vulture | grass | 9 | sky | 1 |
| box turtle | grass | 5 | earth | 3 |
| loggerhead | water | 8 | grass | 0 |
| whiptail | earth | 58 | human | 2 |
| agama | rock | 50 | tree | 8 |
| african crocodile | earth | 15 | grass | 8 |
| hognose snake | earth | 34 | grass | 14 |
| king snake | earth | 24 | grass | 21 |
| garter snake | grass | 36 | earth | 28 |
| water snake | water | 34 | ground | 29 |
| harvestman | shrub | 40 | rock | 27 |
| scorpion | indoor | 2 | outdoor | 4 |
| tarantula | sand | 41 | grass | 6 |
| centipede | indoor | 1 | grass | 4 |
| black grouse | grass | 41 | tree | 3 |
| ptarmigan | snow | 13 | grass | 15 |
| prairie chicken | grass | 61 | snow | 1 |
| sulphur-crested cockatoo | tree | 51 | grass | 14 |
| black swan | water | 13 | ground | 0 |
| echidna | grass | 9 | tree | 0 |
| black stork | grass | 35 | sky | 20 |
| flamingo | water | 1 | sky | 0 |
| bittern | grass | 28 | tree | 9 |
| pelican | water | 19 | sky | 4 |
| sea lion | sand | 22 | water | 19 |
| african hunting dog | grass | 78 | tree | 3 |
| hyena | grass | 36 | road | 8 |
| red fox | grass | 24 | road | 4 |
| arctic fox | snow | 23 | grass | 26 |
| jaguar | water | 0 | tree | 3 |
| lion | grass | 4 | tree | 2 |
| cheetah | grass | 26 | tree | 2 |
| ice bear | snow | 17 | grass | 1 |
| dung beetle | earth | 52 | human | 0 |
| cicada | tree | 13 | human | 0 |
| beaver | water | 6 | grass | 7 |
| bighorn | grass | 20 | rock | 3 |
| mink | grass | 1 | water | 1 |
| otter | water | 14 | tree | 3 |

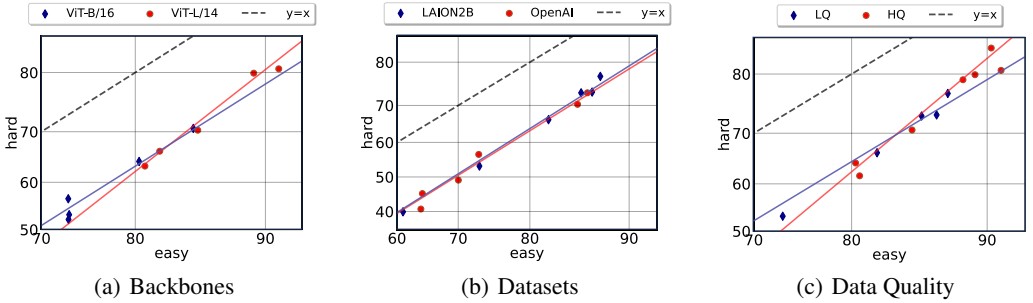

Figure 14: Comparison for the effective robustness with respect to a) different backbones, b) different pre-train datasets, as well as c) high-quality (HQ) and low-quality (LQ) pre-train datasets.

# F   More Results

In this section, we present more experimental results to support our claims.

**Effective Robustness.** In Section 3, we mainly examine the absolute robustness to assess and compare the OOD performance across various CLIP setups, which are well known to be sensitive to the original value scales. Therefore, in Figure 14, we apply the measures of effective robustness [5] to further substantiate our conclusions. Overall, our previous conclusions are upheld, demonstrating that the benefits derived from increasing model scales and enhancing data quality notably outweigh those obtained by merely expanding dataset sizes.

**Top-5 Results for CLIP models.** We present 1 vs. 1000 results for more CLIP checkpoints on the `CounterAnimal` dataset in Table 13, which is an extension of Table 2. Moreover, we present more results for the evaluations on `CounterAnimal`, supplementing our analysis of CLIP models under spurious correlations. To begin with, we report the top-5 scores under the 1 vs. 1000 setup, where we check if the target label is one of the top-5 model predictions. The results are summarized in Table 14. Comparing with the top-1 results in Table 13, we find that there is still a large performance gap between the `easy` and `hard` groups, indicating that the label confusion is quite diverse and not limited to the top two classes.

**Other Versions of Pre-train Datasets.** OpenCLIP provides other CLIP checkpoints beyond our adopted ones. Table 15 summarizes the results of CLIP models similar to Table 2 while using different versions of checkpoints. As we can see, the performance for both `easy` and `hard` is very stable across varying versions, except for `DataComp1B`. The reason is that their various checkpoints use subsets of `DataComp1B`, where `XL` indicates the fully `DataComp1B`, `L` indicates a 140M subset, `M` indicates a 14M subset, and `S` indicates a 1.4M subset.

**Results of OpenAI Prompts.** We further consider the prompt setups following OpenAI CLIP [2], using average text embeddings over 80 predefined prompts as the final text embeddings. The results are summarized in Table 16. As we can see, the average performance for both the `easy` and `hard` groups generally improves 1 to 3 percentage points over the results of our simpler prompt. However, our main conclusion remains unchanged: the ImageNet models generally exhibit better performance and smaller drops. Another interesting finding is that when evaluating with `CLIP-LAION400M-ViT-B/32` (the CLIP checkpoint employed in our data collection), the performance drop with OpenAI prompts is not as high as that of our simple prompt used in Table 2. It indicates that our curation procedure mainly overfit the adopted prompt instead of the particular CLIP checkpoint.

**Average Performance without Balancing.** We by default adopt the balanced average accuracy to offset the impacts of class imbalance. In Tables 17-18, we further summarize the results without class balance, following $\frac{1}{|\mathcal{I}|} \sum_{i \in \mathcal{I}} \mathbf{1}\{\hat{y}_i = \texttt{gt}\}$. As we can see, the performance drop remains obvious, and similar conclusions can be drawn as the balanced results: a) Backbone scales are more important for spurious robustness than pre-train dataset scales, and b) ImageNet models are more reliable when facing spurious features in `CounterAnimal`.

**1 vs. 20 Results for CLIP and ImageNet Models.** We adopt the 1 vs. 20 setup for the evaluations of more advanced LVLMs in Table 4. For a fair comparison, we further summarize the 1 vs. 20

Table 13: The 1 vs. 1000 results for CLIP checkpoints on `CounterAnimal`. The pre-train datasets with high-quality data are marked by *.

| backbone | pre-train dataset | easy | hard | drop |
|---|---|---|---|---|
| RN-50 | OpenAI | 64.02 | 40.70 | 23.32 |
| RN-101 | OpenAI | 64.27 | 45.15 | 19.12 |
| RN-50×4 | OpenAI | 70.02 | 49.07 | 20.95 |
| RN-50×16 | OpenAI | 76.43 | 59.13 | 17.30 |
| RN-50×64 | OpenAI | 80.25 | 66.77 | 13.48 |
| ViT-B/16 | LAION400M | 73.11 | 52.17 | 20.94 |
| ViT-B/16 | OpenAI | 73.08 | 56.56 | 16.52 |
| ViT-B/16 | DataComp1B* | 80.36 | 64.24 | 16.12 |
| ViT-B/16 | LAION2B | 73.18 | 53.18 | 20.00 |
| ViT-B/16 | DFN2B* | 85.03 | 70.61 | 14.42 |
| ViT-B/32 | LAION400M | 67.13 | 36.95 | 30.18 |
| ViT-B/32 | OpenAI | 69.13 | 45.62 | 23.51 |
| ViT-B/32 | DataComp1B* | 75.96 | 53.74 | 22.22 |
| ViT-B/32 | LAION2B | 72.94 | 48.74 | 24.20 |
| ViT-B/32-256 | DataComp1B* | 80.72 | 61.65 | 19.07 |
| ViT-L/14 | LAION400M | 80.90 | 63.31 | 17.59 |
| ViT-L/14 | OpenAI | 85.38 | 70.28 | 15.10 |
| ViT-L/14 | DataComp1B* | 89.29 | 79.90 | 9.39 |
| ViT-L/14 | LAION2B | 82.23 | 66.27 | 15.96 |
| ViT-L/14 | DFN2B* | 90.77 | 80.55 | 10.22 |
| ViT-L/14-336 | OpenAI | 86.36 | 73.14 | 13.21 |
| ViT-H/14 | LAION2B | 85.74 | 73.13 | 12.61 |
| ViT-H/14 | DFN5B* | 88.55 | 79.13 | 9.42 |
| ViT-H/14-384 | DFN5B* | 90.23 | 83.67 | 6.56 |
| ViT-G/14 | LAION2B | 86.81 | 73.32 | 13.49 |
| ViT-bigG/14 | LAION2B | 87.57 | 76.96 | 10.61 |
| ConvNext-B | LAION400M | 59.85 | 36.77 | 23.08 |
| ConvNext-BW | LAION2B | 61.03 | 39.91 | 21.12 |

results for CLIP models in Table 19 and for ImageNet models in Table 20. As we can see, there does not exist a significant change in performance drop compared to 1 vs. 1000 results, indicating that mistakes made by CLIP models are relatively concentrated. As in Figure 2, we also depict the easy versus hard performance for various learning setups with their names, following the 1 vs. 1000 setup in Figure 15 and 1 vs. 20 setups in Figure 16.

**Class-wise Results.** In Tables 21-22, we summarize the detailed results of the class-wise accuracy for the main results in Figure 5. We further depict the drop in accuracy in Figure 17. Generally speaking, the spurious features found in `CLIP-LAION400M-ViT-B/32` can also fail other CLIP setups, and the general trends of decline are preserved class-wise. However, there are some cases where the drop in accuracy between easy and hard is negative, e.g., for data in class ID 33 and 42. It means that for these cases, our collection pipeline may have a large overfit to the adopted CLIP setup, i.e., `CLIP-LAION400M-ViT-B/32`.

Table 14: The 1 vs. 1000 results with top-5 performance scores for CLIP checkpoints on `CounterAnimal`. The pre-train datasets with high-quality data are marked by *.

| backbone | pre-train dataset | easy | hard | drop |
|---|---|---|---|---|
| RN-50 | OpenAI | 91.02 | 77.15 | 13.87 |
| RN-101 | OpenAI | 89.04 | 79.98 | 9.06 |
| RN-50×4 | OpenAI | 91.21 | 83.65 | 7.55 |
| RN-50×16 | OpenAI | 92.72 | 87.65 | 7.55 |
| RN-50×64 | OpenAI | 95.22 | 92.35 | 2.87 |
| ViT-B/16 | LAION400M | 92.54 | 84.03 | 8.51 |
| ViT-B/16 | OpenAI | 94.74 | 88.21 | 6.53 |
| ViT-B/16 | DataComp1B* | 95.04 | 90.89 | 4.15 |
| ViT-B/16 | LAION2B | 91.04 | 84.64 | 6.40 |
| ViT-B/16 | DFN2B* | 95.45 | 91.98 | 3.48 |
| ViT-B/32 | LAION400M | 87.54 | 71.48 | 16.06 |
| ViT-B/32 | OpenAI | 91.28 | 81.29 | 9.99 |
| ViT-B/32 | DataComp1B* | 92.60 | 85.88 | 6.72 |
| ViT-B/32 | LAION2B | 90.73 | 81.47 | 9.25 |
| ViT-B/32-256 | DataComp1B* | 94.26 | 88.33 | 5.93 |
| ViT-L/14 | LAION400M | 94.33 | 88.73 | 5.60 |
| ViT-L/14 | OpenAI | 96.12 | 93.19 | 2.93 |
| ViT-L/14 | DataComp1B* | 97.36 | 95.10 | 2.26 |
| ViT-L/14 | LAION2B | 93.24 | 89.76 | 3.48 |
| ViT-L/14 | DFN2B* | 96.76 | 94.53 | 2.23 |
| ViT-L/14-336 | OpenAI | 96.60 | 94.30 | 2.30 |
| ViT-H/14 | LAION2B | 95.26 | 91.72 | 3.55 |
| ViT-H/14 | DFN5B* | 97.03 | 94.51 | 2.52 |
| ViT-H/14-384 | DFN5B* | 97.02 | 95.45 | 1.57 |
| ViT-G/14 | LAION2B | 95.30 | 91.20 | 4.10 |
| ViT-bigG/14 | LAION2B | 95.31 | 93.01 | 2.29 |
| ConvNext-B | LAION400M | 81.67 | 69.90 | 11.77 |
| ConvNext-BW | LAION2B | 82.64 | 73.27 | 9.37 |

Table 15: The 1 vs. 1000 performance with other versions of CLIP checkpoints in OpenCLIP.

| backbone | pre-train dataset | checkpoint | easy | hard | drop |
|---|---|---|---|---|---|
| ViT-B/16 | LAION400M | E31 | 73.11 | 52.17 | 20.94 |
| ViT-B/16 | LAION400M | E32 | 73.59 | 52.53 | 21.06 |
| ViT-B/16 | DataComp1B | XL S13B B90K | 80.36 | 64.24 | 16.12 |
| ViT-B/16 | DataComp1B | L S1B B8K | 65.80 | 44.14 | 21.66 |
| ViT-B/32 | LAION400M | E31 | 67.13 | 36.95 | 30.18 |
| ViT-B/32 | LAION400M | E32 | 67.13 | 36.98 | 30.15 |
| ViT-B/32 | LAION2B | E16 | 71.32 | 47.21 | 24.11 |
| ViT-B/32 | LAION2B | S34B B79K | 72.94 | 48.74 | 24.20 |
| ViT-B/32 | DataComp1B | XL S13B B90K | 75.96 | 53.74 | 22.22 |
| ViT-B/32 | DataComp1B | M S128M B4K | 25.91 | 11.65 | 14.26 |
| ViT-B/32 | DataComp | S S13M B4K | 0.02 | 0.01 | 0.01 |
| ViT-L/14 | LAION400M | E31 | 80.90 | 63.31 | 17.59 |
| ViT-L/14 | LAION400M | E32 | 81.11 | 63.87 | 17.24 |
| ViT-G/14 | LAION2B | S12B B42K | 83.72 | 68.46 | 15.26 |
| ViT-G/14 | LAION2B | S34B B88K | 86.81 | 73.32 | 13.49 |

Table 16: The 1 vs. 1000 performance using prompts of OpenAI CLIP. The pre-train datasets with high-quality data are marked by $^*$.

| backbone | pre-train dataset | easy | hard | drop |
|---|---|---|---|---|
| RN-50 | OpenAI | 64.55 | 44.20 | 20.35 |
| RN-101 | OpenAI | 64.81 | 46.30 | 18.51 |
| RN-50×4 | OpenAI | 69.62 | 53.68 | 15.93 |
| RN-50×16 | OpenAI | 84.78 | 72.13 | 12.65 |
| RN-50×64 | OpenAI | 84.33 | 72.02 | 12.31 |
| ViT-B/16 | LAION400M | 76.20 | 58.17 | 18.18 |
| ViT-B/16 | OpenAI | 76.58 | 60.58 | 16.00 |
| ViT-B/16 | DataComp1B$^*$ | 82.85 | 69.74 | 13.11 |
| ViT-B/16 | LAION2B | 74.08 | 58.18 | 15.90 |
| ViT-B/16 | DFN2B$^*$ | 85.20 | 74.33 | 10.87 |
| ViT-B/32 | LAION400M | 66.68 | 43.22 | 23.46 |
| ViT-B/32 | OpenAI | 67.23 | 47.11 | 20.12 |
| ViT-B/32 | DataComp1B$^*$ | 76.00 | 59.23 | 16.77 |
| ViT-B/32 | LAION2B | 70.25 | 50.00 | 20.25 |
| ViT-B/32-256 | DataComp1B$^*$ | 79.77 | 64.20 | 15.57 |
| ViT-L/14 | LAION400M | 81.22 | 65.31 | 15.91 |
| ViT-L/14 | OpenAI | 85.76 | 73.23 | 12.53 |
| ViT-L/14 | DataComp1B$^*$ | 89.56 | 81.21 | 8.35 |
| ViT-L/14 | LAION2B | 83.43 | 69.44 | 13.99 |
| ViT-L/14 | DFN2B$^*$ | 90.45 | 82.28 | 8.17 |
| ViT-L/14-336 | OpenAI | 86.45 | 76.30 | 10.15 |
| ViT-H/14 | LAION2B | 86.11 | 75.30 | 10.81 |
| ViT-H/14 | DFN5B$^*$ | 91.33 | 85.20 | 6.13 |
| ViT-H/14-384 | DFN5B$^*$ | 92.20 | 88.01 | 4.19 |
| ViT-G/14 | LAION2B | 87.17 | 77.20 | 10.97 |
| ViT-bigG/14 | LAION2B | 87.57 | 76.96 | 10.61 |
| ConvNext-B | LAION400M | 60.20 | 44.15 | 16.05 |
| ConvNext-BW | LAION2B | 63.33 | 46.11 | 17.22 |

Table 17: The 1 vs. 1000 performance on `CounterAnimal` for CLIP models, evaluating based on the accuracy without balancing. The pre-train datasets with high-quality data are marked by $^*$.

| backbone | pre-train dataset | easy | hard | drop |
|---|---|---|---|---|
| RN-50 | OpenAI | 64.59 | 38.40 | 26.19 |
| RN-101 | OpenAI | 64.18 | 43.99 | 20.19 |
| RN50-×4 | OpenAI | 70.76 | 46.91 | 23.85 |
| RN50-×16 | OpenAI | 77.26 | 58.97 | 18.29 |
| RN50-×64 | OpenAI | 82.88 | 62.84 | 20.04 |
| ViT-B/16 | LAION400M | 75.58 | 48.46 | 27.12 |
| ViT-B/16 | OpenAI | 73.94 | 53.93 | 20.01 |
| ViT-B/16 | DataComp1B$^*$ | 81.83 | 61.47 | 20.36 |
| ViT-B/16 | LAION2B | 74.97 | 51.20 | 23.77 |
| ViT-B/16 | DFN2B$^*$ | 86.10 | 67.95 | 18.14 |
| ViT-B/32 | LAION400M | 69.02 | 33.94 | 35.08 |
| ViT-B/32 | OpenAI | 68.84 | 44.17 | 24.67 |
| ViT-B/32 | DataComp1B$^*$ | 78.16 | 51.50 | 26.66 |
| ViT-B/32 | LAION2B | 74.23 | 46.36 | 27.87 |
| ViT-B/32-256 | DataComp1B$^*$ | 82.38 | 58.56 | 23.82 |
| ViT-L/14 | LAION400M | 81.06 | 61.68 | 19.38 |
| ViT-L/14 | OpenAI | 85.29 | 69.25 | 16.04 |
| ViT-L/14 | DataComp1B$^*$ | 90.79 | 77.28 | 13.51 |
| ViT-L/14 | LAION2B | 83.47 | 62.33 | 21.14 |
| ViT-L/14 | DFN2B$^*$ | 91.81 | 78.10 | 13.71 |
| ViT-L/14-336 | OpenAI | 86.40 | 72.40 | 14.00 |
| ViT-H/14 | LAION2B | 87.10 | 69.84 | 17.26 |
| ViT-H/14 | DFN5B$^*$ | 90.36 | 76.19 | 14.17 |
| ViT-H/14-384 | DFN5B$^*$ | 92.29 | 80.95 | 11.34 |
| ViT-G/14 | LAION2B | 88.09 | 69.96 | 18.13 |
| ViT-bigG/14 | LAION2B | 88.47 | 73.45 | 15.02 |
| ConvNext-B | LAION400M | 60.16 | 34.27 | 25.89 |
| ConvNext-BW | LAION2B | 60.65 | 38.64 | 22.01 |

Table 18: The 1 vs. 1000 performance on `CounterAnimal` for ImageNet models, evaluating based on the accuracy without balancing.

| backbone | easy | hard | drop |
|---|---|---|---|
| AlexNet | 62.33 | 37.20 | 25.12 |
| VGG-11 | 75.92 | 53.35 | 22.57 |
| VGG-13 | 77.23 | 55.58 | 21.65 |
| VGG-19 | 79.40 | 58.93 | 20.47 |
| RN-18 | 76.46 | 52.79 | 23.67 |
| RN-34 | 80.38 | 57.80 | 22.58 |
| RN-50 | 83.52 | 62.97 | 20.54 |
| RN-101 | 83.58 | 64.74 | 18.84 |
| ViT-B/16 | 86.97 | 71.62 | 15.35 |
| ViT-B/32 | 82.03 | 61.71 | 20.32 |
| ViT-L/16 | 85.96 | 70.21 | 15.75 |
| ViT-L/32 | 82.89 | 64.64 | 18.25 |
| ConvNext-S | 89.88 | 76.61 | 13.27 |
| ConvNext-B | 90.27 | 77.51 | 12.76 |
| ConvNext-L | 90.67 | 78.34 | 12.33 |

Table 19: The 1 versus 20 performance on `CounterAnimal` for CLIP models. The pre-train datasets with high-quality data are marked by $^*$.

| backbone | pre-train dataset | easy | hard | drop |
|---|---|---|---|---|
| RN-50 | OpenAI | 67.41 | 43.63 | 23.78 |
| RN-101 | OpenAI | 66.92 | 47.23 | 19.69 |
| RN-50×4 | OpenAI | 71.82 | 50.50 | 21.32 |
| RN-50×16 | OpenAI | 78.60 | 60.63 | 17.97 |
| RN-50×64 | OpenAI | 82.33 | 69.05 | 13.28 |
| ViT-B/16 | LAION400M | 75.51 | 54.59 | 20.92 |
| ViT-B/16 | OpenAI | 75.89 | 58.74 | 17.15 |
| ViT-B/16 | DataComp1B$^*$ | 82.02 | 66.02 | 16.00 |
| ViT-B/16 | LAION2B | 75.85 | 55.48 | 20.37 |
| ViT-B/16 | DFN2B$^*$ | 86.04 | 72.13 | 13.91 |
| ViT-B/32 | LAION400M | 70.46 | 39.44 | 31.02 |
| ViT-B/32 | OpenAI | 72.17 | 49.25 | 22.92 |
| ViT-B/32 | DataComp1B$^*$ | 78.58 | 56.32 | 22.26 |
| ViT-B/32 | LAION2B | 75.68 | 51.86 | 23.82 |
| ViT-B/32-256 | DataComp1B$^*$ | 83.05 | 63.98 | 19.07 |
| ViT-L/14 | LAION400M | 82.27 | 64.89 | 17.38 |
| ViT-L/14 | OpenAI | 86.38 | 72.12 | 14.26 |
| ViT-L/14 | DataComp1B$^*$ | 90.13 | 80.46 | 9.67 |
| ViT-L/14 | LAION2B | 83.81 | 67.68 | 16.13 |
| ViT-L/14 | DFN2B$^*$ | 91.29 | 81.23 | 10.05 |
| ViT-L/14-336 | OpenAI | 87.56 | 75.16 | 12.40 |
| ViT-H/14 | LAION2B | 86.75 | 74.29 | 12.46 |
| ViT-H/14 | DFN5B$^*$ | 89.13 | 79.79 | 9.35 |
| ViT-H/14-384 | DFN5B$^*$ | 90.70 | 84.00 | 6.70 |
| ViT-G/14 | LAION2B | 87.74 | 74.11 | 13.63 |
| ViT-bigG/14 | LAION2B | 88.35 | 77.85 | 10.50 |
| ConvNext-B | LAION400M | 64.85 | 39.71 | 25.14 |
| ConvNext-BW | LAION2B | 65.61 | 44.21 | 21.40 |

Table 20: The 1 versus 20 performance on `CounterAnimal` for ImageNet models.

| backbone | easy | hard | drop |
|---|---|---|---|
| AlexNet | 67.71 | 46.43 | 21.29 |
| VGG-11 | 77.25 | 60.19 | 17.06 |
| VGG-13 | 79.07 | 62.02 | 17.04 |
| VGG-19 | 80.80 | 65.19 | 15.61 |
| RN-18 | 78.11 | 59.47 | 18.64 |
| RN-34 | 81.14 | 64.32 | 16.82 |
| RN-50 | 83.72 | 68.60 | 15.29 |
| RN-101 | 84.13 | 70.77 | 13.37 |
| ViT-B/16 | 86.57 | 76.88 | 9.69 |
| ViT-B/32 | 82.56 | 68.30 | 14.26 |
| ViT-L/16 | 85.71 | 74.94 | 10.77 |
| ViT-L/32 | 83.86 | 71.00 | 12.86 |
| ConvNext-S | 89.31 | 81.61 | 7.69 |
| ConvNext-B | 89.58 | 82.32 | 7.26 |
| ConvNext-L | 89.84 | 82.67 | 7.17 |

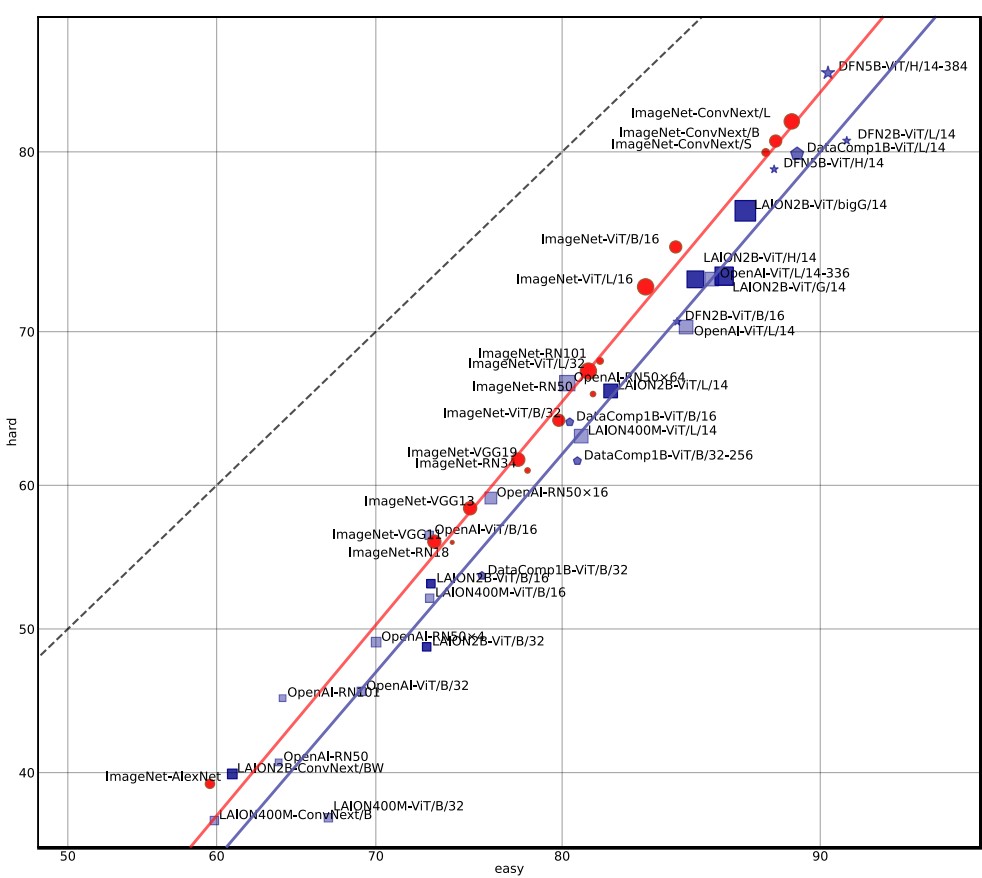

Figure 15: The `easy` versus `hard` performance (%) for CLIP and ImageNet models, following the 1 vs. 1000 setup. We also present the model setups for each `easy`-`hard` result pair.

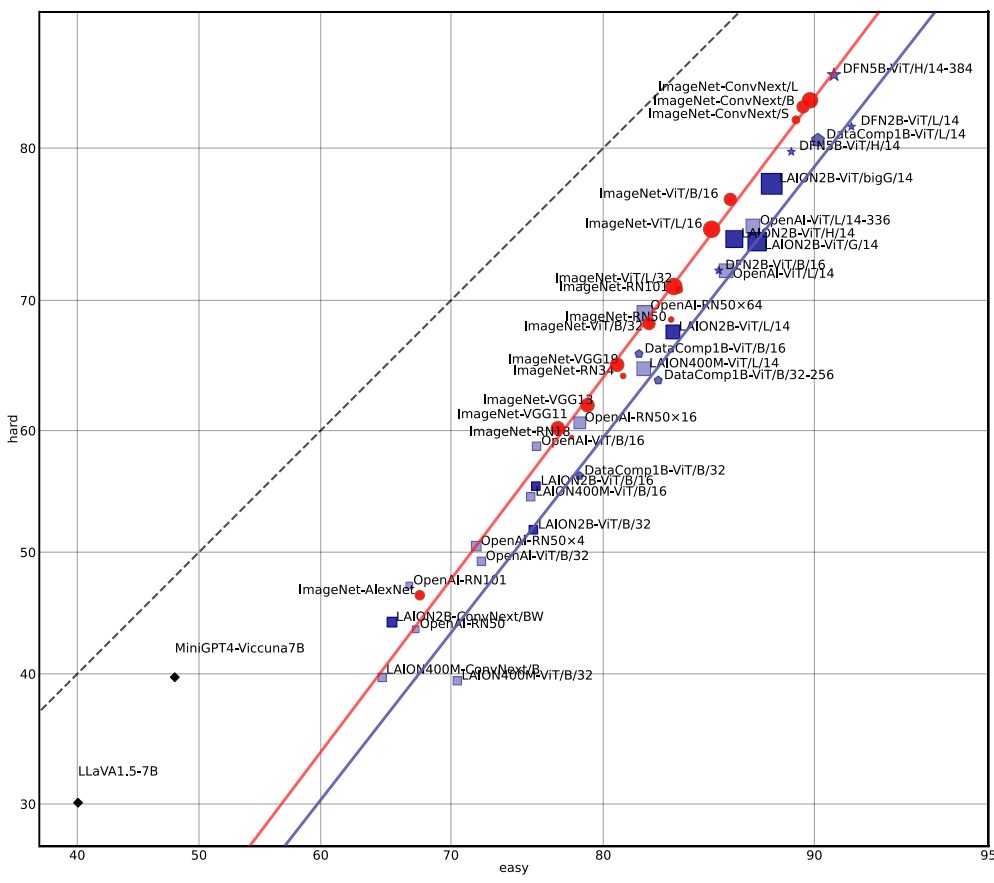

Figure 16: The `easy` versus `hard` performance (%) for CLIP, ImageNet models, and more advanced LVLMs, following the 1 vs. 20 setup. We also present the model setups for each `easy-hard` result pair.

Table 21: Class-wise 1 vs. 1000 performance on `CounterAnimal` for different backbones CLIP-trained on `LAION400M`.

| class ID | CLIP-LAION400M-ViT-B/16 | | | CLIP-LAION400M-ViT-B/32 | | | CLIP-LAION400M-ViT-L/14 | | |
|---|---|---|---|---|---|---|---|---|---|
| | easy | hard | drop | easy | hard | drop | easy | hard | drop |
| 1 | 71.36 | 64.60 | 6.76 | 79.61 | 57.52 | 22.09 | 93.20 | 91.15 | 2.05 |
| 2 | 87.18 | 69.37 | 17.81 | 78.63 | 49.55 | 29.08 | 94.02 | 75.68 | 18.34 |
| 3 | 18.85 | 8.65 | 10.20 | 28.69 | 14.59 | 14.09 | 14.75 | 7.57 | 7.19 |
| 4 | 90.00 | 70.99 | 19.01 | 81.15 | 48.15 | 33.01 | 94.23 | 90.12 | 4.11 |
| 5 | 76.19 | 67.35 | 8.84 | 87.76 | 41.84 | 45.92 | 97.96 | 82.65 | 15.31 |
| 6 | 88.32 | 67.72 | 20.60 | 73.36 | 48.10 | 25.26 | 83.94 | 68.35 | 15.59 |
| 7 | 78.64 | 43.96 | 34.68 | 73.64 | 18.68 | 54.96 | 81.36 | 69.23 | 12.13 |
| 8 | 69.23 | 44.00 | 25.23 | 73.85 | 49.00 | 24.85 | 87.69 | 74.00 | 13.69 |
| 9 | 74.00 | 37.50 | 36.50 | 54.00 | 30.83 | 23.17 | 54.00 | 39.17 | 14.83 |
| 10 | 79.92 | 26.00 | 53.92 | 60.64 | 4.00 | 56.64 | 69.48 | 13.00 | 56.48 |
| 11 | 62.43 | 28.87 | 33.55 | 74.26 | 28.87 | 45.39 | 60.95 | 42.96 | 17.99 |
| 12 | 83.52 | 51.19 | 32.33 | 72.53 | 35.71 | 36.81 | 89.01 | 72.62 | 16.39 |
| 13 | 64.04 | 26.83 | 37.21 | 22.17 | 2.44 | 19.73 | 17.24 | 7.32 | 9.92 |
| 14 | 63.60 | 53.06 | 10.54 | 32.46 | 22.45 | 10.01 | 64.04 | 44.90 | 19.14 |
| 15 | 61.54 | 22.09 | 39.45 | 67.95 | 19.68 | 48.27 | 85.90 | 18.47 | 67.42 |
| 16 | 82.12 | 13.50 | 68.62 | 68.87 | 1.23 | 67.65 | 88.08 | 50.92 | 37.16 |
| 17 | 56.09 | 52.00 | 4.09 | 48.50 | 20.00 | 28.50 | 77.25 | 52.80 | 24.45 |
| 18 | 68.35 | 54.92 | 13.43 | 29.11 | 4.17 | 24.95 | 87.34 | 69.32 | 18.02 |
| 19 | 83.98 | 74.05 | 9.93 | 81.82 | 43.67 | 38.15 | 91.34 | 70.89 | 20.46 |
| 20 | 67.21 | 59.62 | 7.60 | 55.74 | 20.19 | 35.55 | 75.41 | 70.19 | 5.22 |
| 21 | 67.31 | 37.12 | 30.19 | 73.08 | 43.94 | 29.14 | 71.15 | 56.82 | 14.34 |
| 22 | 87.72 | 57.01 | 30.71 | 96.49 | 67.29 | 29.20 | 100.00 | 80.37 | 19.63 |
| 23 | 85.33 | 50.57 | 34.75 | 59.85 | 17.24 | 42.60 | 83.78 | 41.38 | 42.40 |
| 24 | 98.77 | 78.00 | 20.77 | 88.34 | 63.00 | 25.34 | 98.77 | 95.00 | 3.77 |
| 25 | 98.04 | 88.68 | 9.36 | 93.63 | 68.87 | 24.76 | 99.02 | 86.79 | 12.23 |
| 26 | 5.60 | 1.81 | 3.79 | 20.00 | 4.07 | 15.93 | 43.20 | 8.60 | 34.60 |
| 27 | 86.42 | 62.42 | 24.00 | 77.78 | 14.77 | 63.01 | 85.19 | 78.52 | 6.66 |
| 28 | 65.48 | 27.72 | 37.76 | 79.70 | 55.45 | 24.25 | 91.37 | 82.18 | 9.19 |
| 29 | 92.20 | 67.92 | 24.27 | 80.49 | 39.62 | 40.87 | 95.12 | 83.02 | 12.10 |
| 30 | 96.98 | 82.83 | 14.15 | 86.21 | 71.72 | 14.49 | 99.14 | 93.94 | 5.20 |
| 31 | 93.10 | 78.30 | 14.80 | 82.76 | 42.45 | 40.31 | 94.83 | 94.34 | 0.49 |
| 32 | 95.71 | 84.72 | 10.99 | 85.24 | 63.89 | 21.35 | 98.57 | 97.22 | 1.35 |
| 33 | 83.24 | 80.00 | 3.24 | 92.20 | 82.00 | 10.20 | 86.42 | 80.00 | 6.42 |
| 34 | 65.03 | 61.90 | 3.13 | 69.23 | 59.05 | 10.18 | 76.92 | 71.43 | 5.49 |
| 35 | 76.42 | 36.13 | 40.29 | 67.48 | 26.05 | 41.43 | 88.62 | 61.34 | 27.27 |
| 36 | 16.92 | 5.75 | 11.17 | 33.85 | 13.72 | 20.13 | 83.08 | 67.70 | 15.38 |
| 37 | 79.47 | 62.61 | 16.86 | 74.90 | 45.95 | 28.96 | 93.16 | 82.43 | 10.72 |
| 38 | 96.70 | 77.36 | 19.34 | 80.66 | 55.66 | 25.00 | 98.11 | 83.02 | 15.09 |
| 39 | 99.21 | 79.09 | 20.12 | 97.62 | 70.91 | 26.71 | 100.00 | 90.91 | 9.09 |
| 40 | 49.23 | 23.40 | 25.83 | 56.92 | 17.02 | 39.90 | 58.46 | 14.89 | 43.57 |
| 41 | 86.90 | 61.36 | 25.53 | 68.97 | 48.86 | 20.10 | 80.69 | 56.82 | 23.87 |
| 42 | 75.73 | 85.00 | -9.27 | 84.47 | 67.00 | 17.47 | 90.29 | 93.00 | -2.71 |
| 43 | 67.37 | 66.67 | 0.70 | 37.89 | 22.92 | 14.98 | 64.21 | 60.42 | 3.79 |
| 44 | 22.08 | 3.92 | 18.16 | 18.18 | 0.00 | 18.18 | 72.73 | 24.51 | 48.22 |
| 45 | 72.52 | 51.43 | 21.09 | 72.52 | 40.95 | 31.57 | 80.92 | 53.33 | 27.58 |

Table 22: Class-wise 1 vs. 1000 performance on `CounterAnimal` for `ViT-B/32` CLIP-trained on different datasets.

| class ID | CLIP-LAION2B-ViT-B/32 | | | CLIP-LAION400M-ViT-B/32 | | | CLIP-OpenAI-ViT-B/32 | | |
|---|---|---|---|---|---|---|---|---|---|
| | easy | hard | drop | easy | hard | drop | easy | hard | drop |
| 1 | 86.41 | 79.65 | 6.76 | 79.61 | 57.52 | 22.09 | 81.55 | 66.37 | 15.18 |
| 2 | 86.32 | 72.97 | 13.35 | 78.63 | 49.55 | 29.08 | 85.47 | 58.56 | 26.91 |
| 3 | 10.66 | 9.73 | 0.93 | 28.69 | 14.59 | 14.09 | 18.85 | 12.43 | 6.42 |
| 4 | 91.54 | 74.69 | 16.85 | 81.15 | 48.15 | 33.01 | 77.69 | 38.27 | 39.42 |
| 5 | 75.51 | 55.10 | 20.41 | 87.76 | 41.84 | 45.92 | 61.22 | 38.78 | 22.45 |
| 6 | 83.58 | 64.56 | 19.02 | 73.36 | 48.10 | 25.26 | 77.74 | 65.82 | 11.91 |
| 7 | 72.27 | 29.67 | 42.60 | 73.64 | 18.68 | 54.96 | 88.64 | 67.03 | 21.60 |
| 8 | 92.31 | 77.50 | 14.81 | 73.85 | 49.00 | 24.85 | 87.69 | 72.50 | 15.19 |
| 9 | 44.00 | 25.00 | 19.00 | 54.00 | 30.83 | 23.17 | 70.00 | 35.83 | 34.17 |
| 10 | 87.55 | 43.00 | 44.55 | 60.64 | 4.00 | 56.64 | 67.87 | 15.00 | 52.87 |
| 11 | 68.64 | 45.07 | 23.57 | 74.26 | 28.87 | 45.39 | 53.85 | 11.27 | 42.58 |
| 12 | 82.42 | 41.67 | 40.75 | 72.53 | 35.71 | 36.81 | 78.02 | 61.90 | 16.12 |
| 13 | 31.53 | 11.38 | 20.14 | 22.17 | 2.44 | 19.73 | 38.92 | 16.26 | 22.66 |
| 14 | 60.09 | 47.96 | 12.13 | 32.46 | 22.45 | 10.01 | 17.98 | 18.37 | -0.38 |
| 15 | 71.79 | 22.89 | 48.90 | 67.95 | 19.68 | 48.27 | 75.64 | 31.33 | 44.32 |
| 16 | 71.52 | 15.95 | 55.57 | 68.87 | 1.23 | 67.65 | 72.85 | 8.59 | 64.26 |
| 17 | 61.08 | 36.00 | 25.08 | 48.50 | 20.00 | 28.50 | 54.29 | 32.80 | 21.49 |
| 18 | 67.09 | 39.77 | 27.41 | 29.11 | 4.17 | 24.95 | 74.68 | 25.76 | 48.93 |
| 19 | 68.40 | 60.76 | 7.64 | 81.82 | 43.67 | 38.15 | 77.49 | 51.27 | 26.22 |
| 20 | 73.77 | 54.81 | 18.96 | 55.74 | 20.19 | 35.55 | 70.49 | 34.62 | 35.88 |
| 21 | 69.23 | 31.82 | 37.41 | 73.08 | 43.94 | 29.14 | 67.31 | 27.27 | 40.03 |
| 22 | 92.98 | 62.62 | 30.37 | 96.49 | 67.29 | 29.20 | 89.47 | 53.27 | 36.20 |
| 23 | 64.86 | 37.93 | 26.93 | 59.85 | 17.24 | 42.60 | 60.62 | 32.18 | 28.43 |
| 24 | 95.09 | 71.00 | 24.09 | 88.34 | 63.00 | 25.34 | 95.09 | 85.00 | 10.09 |
| 25 | 91.67 | 57.55 | 34.12 | 93.63 | 68.87 | 24.76 | 96.57 | 83.96 | 12.61 |
| 26 | 13.60 | 0.45 | 13.15 | 20.00 | 4.07 | 15.93 | 15.20 | 0.45 | 14.75 |
| 27 | 66.67 | 48.32 | 18.34 | 77.78 | 14.77 | 63.01 | 77.78 | 69.13 | 8.65 |
| 28 | 68.53 | 49.50 | 19.02 | 79.70 | 55.45 | 24.25 | 63.45 | 16.83 | 46.62 |
| 29 | 85.37 | 53.77 | 31.59 | 80.49 | 39.62 | 40.87 | 78.54 | 46.23 | 32.31 |
| 30 | 93.10 | 61.62 | 31.49 | 86.21 | 71.72 | 14.49 | 82.76 | 30.30 | 52.46 |
| 31 | 86.21 | 63.21 | 23.00 | 82.76 | 42.45 | 40.31 | 91.38 | 68.87 | 22.51 |
| 32 | 95.71 | 84.72 | 10.99 | 85.24 | 63.89 | 21.35 | 88.57 | 72.22 | 16.35 |
| 33 | 85.84 | 80.00 | 5.84 | 92.20 | 82.00 | 10.20 | 76.59 | 80.00 | -3.41 |
| 34 | 49.65 | 36.19 | 13.46 | 69.23 | 59.05 | 10.18 | 69.93 | 56.19 | 13.74 |
| 35 | 78.05 | 21.85 | 56.20 | 67.48 | 26.05 | 41.43 | 73.17 | 48.74 | 24.43 |
| 36 | 61.54 | 42.92 | 18.62 | 33.85 | 13.72 | 20.13 | 58.46 | 46.46 | 12.00 |
| 37 | 83.27 | 51.80 | 31.47 | 74.90 | 45.95 | 28.96 | 85.55 | 62.22 | 19.34 |
| 38 | 92.92 | 68.87 | 24.06 | 80.66 | 55.66 | 25.00 | 78.77 | 47.17 | 31.60 |
| 39 | 96.03 | 76.36 | 19.67 | 97.62 | 70.91 | 26.71 | 100.00 | 93.64 | 16.36 |
| 40 | 64.62 | 34.04 | 30.57 | 56.92 | 17.02 | 39.90 | 56.92 | 25.53 | 31.39 |
| 41 | 86.21 | 54.55 | 31.66 | 68.97 | 48.86 | 20.10 | 86.21 | 84.09 | 2.12 |
| 42 | 72.82 | 69.00 | 3.82 | 84.47 | 67.00 | 17.47 | 71.84 | 71.00 | 0.84 |
| 43 | 67.37 | 59.38 | 7.99 | 37.89 | 22.92 | 14.98 | 69.47 | 62.50 | 6.97 |
| 44 | 63.64 | 19.61 | 44.03 | 18.18 | 0.00 | 18.18 | 10.39 | 2.94 | 7.45 |
| 45 | 70.99 | 48.57 | 22.42 | 72.52 | 40.95 | 31.57 | 35.88 | 30.48 | 5.40 |

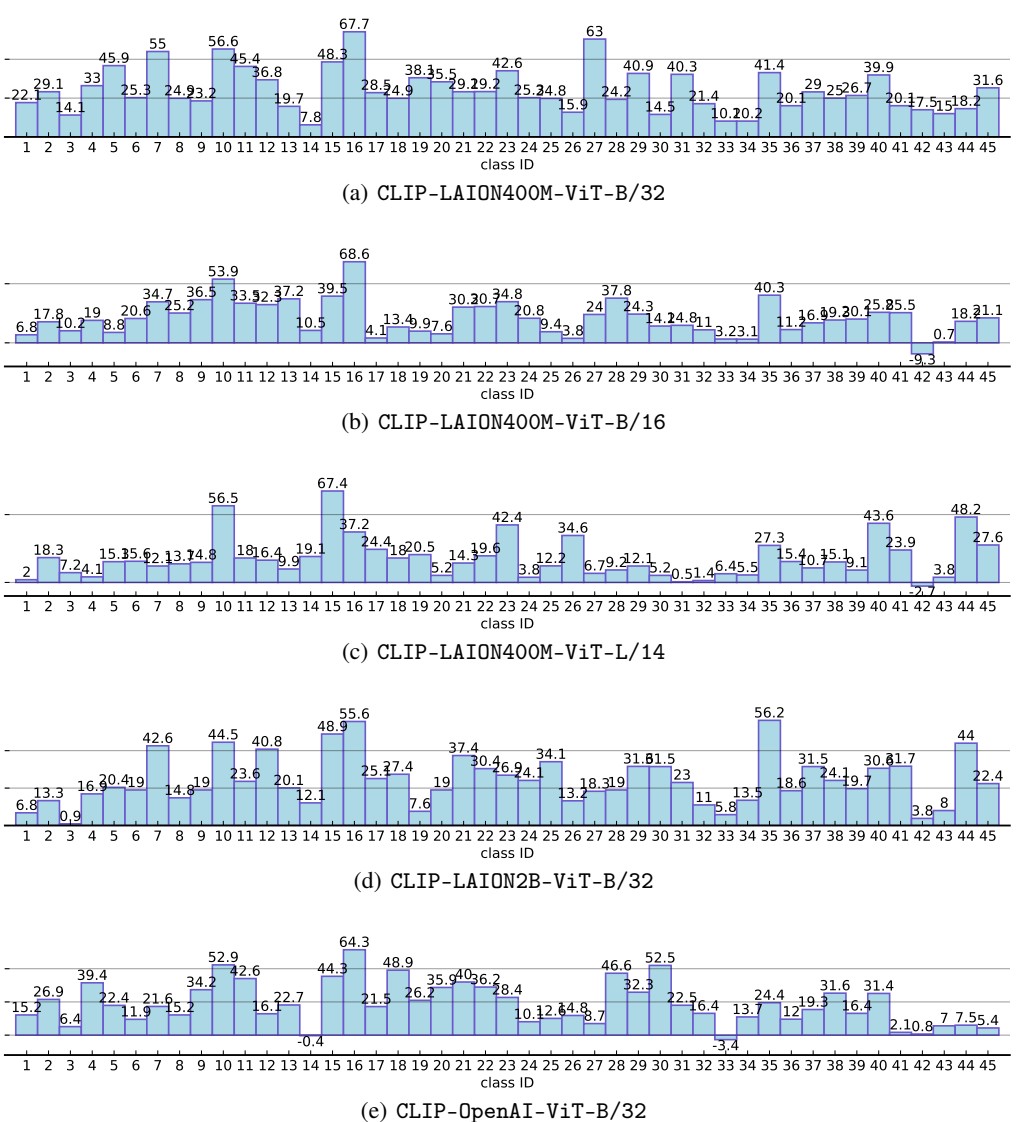

Figure 17: The performance drop (%) between easy to hard on varying CLIP setups. The horizontal axis denotes the class ids and the vertical axis denotes the class-wise accuracy drop.

