# OpenReview forum: "A Sober Look at the Robustness of CLIPs to Spurious Features"
_NeurIPS.cc/2024/Conference — NeurIPS 2024 poster_

### Official Review · Reviewer_BaJ3 · 2024-06-19

**Soundness:** 1
**Presentation:** 2
**Contribution:** 2
**Rating:** 6
**Confidence:** 5

**Summary:**

The authors aim to investigate spurious correlations learned by CLIP models. For this, they curate a novel dataset where animals are organized into common and uncommon backgrounds, e.g. a polar bear is more likely encountered in snow than on grass. The authors then perform experiments where they benchmark various CLIP and ImageNet models on the curated dataset. They observe that CLIP models suffer from spurious correlations which stem from changing the background.

**Strengths:**

I think the issue of spurious correlations is important and one needs to understand how and whether VLMs learn spurious features. The paper presents many experiments and shows that scale or backbone capacity do not improve the effective robustness on CounterAnimal which is interesting.

**Weaknesses:**

The paper has many issues, both in terms of writing and the methodology which need to be fixed.

### Major:
**The authors missed important previous works**: The paper “Recognition in Terra Incognita” is very related to this work and also proposes a “dataset designed to measure recognition generalization to novel environments” based on camera traps. The dataset is sorted according to difficult environments for different animals, which makes it very similar to CounterAnimal. I think the authors need to cite and discuss this paper. Currently, I do not understand the benefit of having a new dataset in addition to the already present one. The waterbirds dataset is also highly similar and should be discussed (https://arxiv.org/pdf/1911.08731). The authors cite that paper, but do not discuss it in the Related Work section, nor put it into context with CounterAnimal. The backgrounds challenge (https://github.com/MadryLab/backgrounds_challenge) is also highly related and should be discussed. In general, the related work section is very weak, given how extensively spurious correlations and worst-group-accuracy have been studied. Another important work to be discussed would be "Finding and Fixing Spurious Patterns with Explanations" (https://arxiv.org/abs/2106.02112).

**The naming of the common vs counter groups is misleading**:
Line 165: “Photos with the highest CLIP accuracy are assigned to the common group, and those with the lowest CLIP accuracy are assigned to the counter group.” I have a major understanding issue here. As far as I understood the paper before this line, the goal was to put images with common backgrounds into the common group and images with uncommon backgrounds into the counter group. This is also depicted in Fig. 1 or Table 1. The caption in Fig.1 says that “Most ice bears appear in a snow background (i.e., common), while it also is reasonable to find some ice bears in a grassy environment (i.e., counter)”. But here, the authors write that accuracy has actually been used to separate images into these groups? But then the frequency of the co-occurrence of certain backgrounds and classes has not been taken into account, or rather, it is a conjecture that those backgrounds where the CLIP model has higher accuracy on are more “common”?

**The terms "effective robustness" and "robustness" are used interchangeably which is wrong and confusing**:
I think the paper conflates the terms “robustness” and “effective robustness” which is confusing. When looking at effective robustness plots, such as in Fig. 2, we are interested in the residual difference between the measured value and the value predicted by the linear fit. As I can see, all plotted markers (CLIP and ImageNet) lie on their respective linear fits, and none of the interventions, such as CLIP-DC or CLIP-DFN offer any effective robustness benefits. It is though true that the **absolute** robustness numbers are overall higher for the CLIP-DFN models, for larger models or models trained on more data. I am however confused by the authors discussion of this observation. On the one hand, they write that larger CLIP models are more robust but increasing the dataset size does not yield improvements. First, I am confused whether they mean “effective robustness” or “robustness” here. Second, I do not see the effect the authors are describing: Both more data and larger backbones have higher absolute robustness but the same effective robustness as the other models. The statement “CLIP models trained on high-quality data are more robust” is also confusing, because it is not clear whether “robustness” or “effective robustness” is meant.

**Due to methodology issues, results on CLIP models cannot be compared to results on ImageNet models (or other advanced LVLMs):**
Line 60: “d) Spurious discovering: preserving classes and associated data based on the decrease in zero-shot performance (i.e., evaluating based on pre-trained CLIP models without fine-tuning) when shifting the backgrounds.” This step is really unclear. Do the authors curate the dataset based on the zero-shot accuracy of a CLIP model? From the introduction and the abstract, it sounds like the authors want to benchmark the robustness of CLIP vs ImageNet models on this custom dataset. But then, it is strange that CLIP models also seem to be used during the curation process. After reading the more detailed description in line 156, I think the statements made in line 85 are misleading. The authors write “ImageNet models are more robust to spurious correlations captured by CounterAnimal” and “Compared with CLIP models (colored in blue), surprisingly, we find that ImageNet models exhibit a stronger robustness to the spurious correlations in CounterAnimal.” Given that CounterAnimal has been curated based on the performance drop of a CLIP model, I find it very unsurprising that CLIP models perform worse on it compared to ImageNet models. I think that if CounterAnimal had been curated based on an ImageNet-trained ResNet50, the trend would have been reversed. I think all statements comparing CLIP and ImageNet trained models on CounterAnimal need to be relaxed and I think that this comparison is quite meaningless because of the described selection bias. I think that the whole Section 3.3. is misleading for this reason and statements such as the following cannot be made given the methodology issues: “Surprisingly, we find that ImageNet models are more robust to spurious features in the CounterAnimal dataset. This finding may contradict the common belief [Radford et al., 2021, Shi et al., 2023] that the CLIP models tend to be more robust to spurious correlations than single-modal supervised learning.” Similarly, the conjecture paragraph from line 265 onwards is wrong and cannot be made.

For the same reason, the comparison to advanced LVLMs in line 273 onwards cannot be made.

Figure 1: Are these examples cherry-picked or are they representative of the data points present in CounterAnimal? I am asking this, because of the Winoground dataset [A]. This dataset tests the compositionality of VLMs by forcing a model to match two captions to two respective images. Winoground has later been criticized because the two images in the choice process are not equally hard [B]. For example, the model needs to match “the glass is on the grass” and “the grass is in the glass” to the corresponding images. However, there is much more grass in the image matching to the first caption, and the model likely picks that image for both captions just because there is more grass and it makes the decision in a bag-of-words-manner. To summarize, Winoground did not control for object size, orientation and other confounders. In Fig.1, it appears that the main objects (the polar bears) are equal in size, so size could be excluded as a possible confounder? Did the authors consider this possibility, i.e. that the drop in performance could be explained by other differences in the images from the respective domains?
[A] https://arxiv.org/abs/2204.03162
[B] https://arxiv.org/abs/2211.00768

### General:
Line 25: please cite CLIP

Line 64: “The resulting dataset covers a total of 45 animal classes, ends up with 7,174 common photos and 5,926 counter photos, aligning with the standard size as an evaluation dataset [Recht et al., 2019, Hendrycks et al., 2021].” -> I do not understand this statement. Different test sets have different numbers of test images. ImageNet Val has 50k images for example. In what sense are the presented numbers standard?


Line 94: “Overall, larger CLIP backbones (i.e., larger markers) can improve the effective robustness, implying that scaling up backbones may enhance robustness against spurious features.” -> I do not see this in Fig. 2. The larger markers appear to be on the fitted line, same as the smaller markers. Effective robustness measures the difference with respect to the linear fit, and there is none for the larger CLIP backbones. Please clarify this point.


Line 146: “feature noise involves severe feature corruptions” -> Please be more specific here. What do you mean with feature noise? Do features refer to animals features such as missing ears or such? Or to the images themselves?

Line 147: “clarity issues arise when animal objects are not in major positions” -> unclear formulation: what is a major position? Do the authors mean that the animals are too small or not in the center of the image?

Line 153: “Note that the class space of backgrounds as above is not entirely orthogonal with each other due to the inherent ambiguity of the real-world situations. Nevertheless, we try our best to discern the assigned background labels within each animal class.” -> This is unclear. How many images would be ambiguous? I could imagine that many images would have two backgrounds, such as e.g. grass and sky or snow and water. For example, the last image in Fig. 1 on the left has both snow and water. It is not clear to me that only picking the snow background and ignoring the water is correct here. Further, at least for CLIP, the caption can contain several background keywords.

Further, I imagine animals occur in all kinds of environments, but there are only two backgrounds for each animal. Were the other images also discarded?

Line 214: “Therefore, we conclude that our CounterAnimal dataset possesses some realistic shifts that are generally contained in large-scale pre-training data, regardless of backbones.” This conclusion cannot be drawn from this experiment since the backbone has not been varied here.

### Section 4:
The proposed experiment is very similar to the well-known ShiftMNIST [D] or ColoredMNIST [E] datasets, which test the influence of spurious correlations. The findings here are not novel and should be brought into perspective with previous work. I do not understand how Fig. 11 relates to the text. What is “supervised”, “obj”, “objbkg”?
[D] https://arxiv.org/pdf/1811.00401
[E] https://arxiv.org/pdf/1907.02893

### Typos, grammar:
The quality of the text is poor on some occasions which makes reading and understanding the paper difficult. The manuscript would benefit from a round of proof-reading. Some statements and formulations should be made more precise.
Line 32: “The performance boosts over ImageNet models seem to suggest that CLIP resolves distribution shifts and thus spark a rich discussion about its rationale.” Strange formulation. How can “distribution shifts be resolved”? Please rephrase for clarity.

Line 112: “More specifically, [Yang et al., 2023] report that CLIP models may misaligned frequently co-occured objects with the corresponding texts.”

Line 115: “[Tong et al., 2024] find that CLIP misaligned samples will further cause the hallucination of LVLMs.” I do not understand this statement, grammar errors.

Line 132: “Meanwhile, many existing datasets, e.g., DomainBed and Wilds, do not have overlapped label space with ImageNet, making the comparison between ImageNet and CLIP models hard.” There is a version of DomainBed [C] where the dataset has been filtered to only include classes compatible with ImageNet, such that an evaluation of ImageNet models is possible out-of-the-box.
[C] https://openreview.net/pdf?id=LiC2vmzbpMO

Line 171: “Recalling that, when CLIP models resort to the shortcut of data, the model performance will heavily correlate with the backgrounds presented in the common group yet is compromised when coming to the counter group.” Grammar errors, I do not understand this sentence. What is “the shortcut of data”?

Line 208: “It suggests that the CounterAnimal dataset captures some general spurious shifts that at least commonly present in the pre-train dataset of LAION400M.” grammar

Line 213: “Here, the spurious features degenerate the zero-shot robustness of CLIP models trained on both LAION2B and by OpenAI.” Typo? “degenerate”?

Line 243: “In Figure 7, we consider two pre-train datasets, namely, LAION2B and the close-soured data from OpenAI” typo

Line 297: “Nevertheless, in the following theorem, we justify that CLIP remains learning to use spurious features, aligned with our experimental observations on the CounterAnimal dataset.” grammar

Strange space break between line 310 and 311.

# Summary of the review:
We could fix the naming convention from "common" and "counter" to something like "hard" and "easy" since accuracy has been used rather than frequency of certain backgrounds to classify backgrounds into certain groups. Based on my arguments below, I believe we cannot compare CLIP models to ImageNet models on the proposed dataset in any sensible way due to the introduced selection bias. I believe the very title of the paper is misleading since the posed question cannot be answered based on the methodology issues. But if we remove the claims about comparing ImageNet models and CLIP models, then, the main point of the paper is that there exist backgrounds which are harder for CLIP models, given certain classes, and other backgrounds which are easier. I don't think that this observation is particularly interesting on its own. The authors did not relate the hardness of the backgrounds to their frequency in the pretraining dataset or anything else. The observation that backgrounds matter is also not novel but quite well-known and the authors do not offer a solution. Further, the writing is quite poor and confusing on many occasions; I provided many examples of incorrect and confusing sentences below.

**Questions:**

I have written a very detailed review above. I expect clarifications with respect to the raised points, at the very least in the "Major" paragraph.

**Limitations:**

The limitations discuss the comparison in performance of the CLIP vs ImageNet models which I believe cannot be made due to the methodological issues in this work.

---

> ### Author Rebuttal · Authors · 2024-08-07
>
> Thank you for your constructive comments and suggestions! Please find our responses below.
>
> >Q1. The authors missed important previous works.
>
> A1. Many thanks for your suggestion. CounterAnimal utilizes high-quality data available on the internet, whereas Terralcognita relies on camera trapping data, which are not as widely spread. We believe CounterAnimal is preferred over Terralcognita as CounterAnimal much align with the training / deployment setups of CLIPs. Moreover, to compare with ImageNet models, we should ensure that we align the label space of ImageNet, satisfied by CounterAnimal but not by Terralcognita. WaterBird also has such a problem, and is limited to simple binary classification. For the Background Challenge, they study the robustness from another perspective, i.e., w/ or w/o the background. However, to find spurious features within CLIP datasets, we need to compare model performance across different backgrounds. At last, SPIRE masks a part of objects from the original image, which may introduce new factors to degenerate model performance.
>
> >Q2. The naming of the common vs counter groups is misleading.
>
> A2. Using accuracy instead of frequency makes the data collection procedure more effective in finding spurious features that make CLIP models fail. We will follow your suggestion and change the names of groups.
>
> >Q3. The terms "effective robustness" and "robustness" are used interchangeably which is wrong and confusing.
>
> A3. We agree that we need to clarify the term “robustness” in Section 3. We further depict the lines of effective robustness, where we found that the conclusions remain the same. However, the improvement from increasing model scales and improving data quality is much higher than simply scaling up the datasets. We will add the related discussion in our revision.
>
> >Q4. Due to methodology issues, results on CLIP models cannot be compared to results on ImageNet models (or other advanced LVLMs).
>
> A4. In general, we do not intend to claim that ImageNet models are generally more robust to spurious features. Instead, our goal is to emphasize that spurious features within CounterAnimal may not be as influential for ImageNet models, supporting that CounterAnimal captures spurious features within CLIPs, underscoring the significance of CounerAnimal.  For previous works, they assess CLIP robustness using OOD datasets primarily for ImageNet datasets. Such a tendency may not fully reflect the CLIP robustness, as spurious features within ImageNet datasets may not be learned by CLIP models. Our CounterAnimal fills this gap, offering a more comprehensive perspectives when studying CLIP robustness.
>
> >Q5. Did the authors consider this possibility, i.e. that the drop in performance could be explained by other differences in the images from the respective domains?
>
> A5. we have tried our best to do the quality control, and get rid of the influence of other factors. We also evaluate some potential confounders like gestures, as in the response to Reviewer iLSC, and find that it is relatively equally distributed across groups. Please kindly let us know if you feel there is another potential factor that may affect the performance.
>
> >Q6. Different test sets have different numbers of test images. ImageNet Val has 50k images for example. In what sense are the presented numbers standard?
>
> A6. We aim at clarifying that our CounterAnimal (about 13K) aligns with the standard size as an evaluation dataset, resembling the scales of other many popular OOD evaluation datasets such as [1-2]. We will make our discussion clearer in our revision.
>
> [1] The many faces of robustness: A critical analysis of out-of-distribution generalization.
>
> [2] Do ImageNet Classifier Generalize to ImageNet?
>
> >Q7. Unclear description in the dataset construction procedure in Section 2.1.
>
> A7. In data curation, feature noise refers to cases where some pixels are disrupted or missing. Clarity issues refer to cases where animal objects are largely occluded by backgrounds or other irrelevant objects. It also includes cases where animal objects do not occupy most of pictures. In background labeling, there are two cases where one image can be assigned more than one background label, i.e., some backgrounds can be confusing and other may possess more than one background. Due to space limit, we will further clarify our data construction procedure in our revision.
>
> >Q8. Further, I imagine animals occur in all kinds of environments, but there are only two backgrounds for each animal. Were the other images also discarded?
>
> A8. Yes, we only preserve two groups for each animal, aiming to better capture spurious features captured of CLIPs. In Sec 3.1, we further show that the identified spurious features are general across different CLIP setups, justifying that these spurious features will not be largely bias towards a particular CLIP checkpoint.
>
> >Q9.  “Therefore, we conclude that our CounterAnimal dataset possesses some realistic shifts that are generally contained in large-scale pre-training data, regardless of backbones.” This conclusion cannot be drawn from this experiment.
>
> A9. We have shown that using different backbones will always lead to poor performance on the counter group in Fig 7. We will make our discussion clearer in our revision. Thank you for your suggestion.
>
> >Q10. The proposed experiment is very similar to the well-known ShiftMNIST [D] or ColoredMNIST [E] datasets, which test the influence of spurious correlations.
>
> A10. The experiments are presented to echo our theoretical analysis that CLIP learns to align spurious features with object captions. For the legends in Fig. 11, “supervised” refers to the results of supervised trained models, while “obj” and “objbkg” refer to use different prompts to fine-tune CLIPs. We will add more discussion about the table legends in Fig. 11.
>
> >Q11. Typos and grammar issues.
>
> A11. Thanks for pointing out our typos. We will correct them in our revision.

---

> > ### Comment · Reviewer_BaJ3 · 2024-08-13
> > **Response to the rebuttal**
> >
> > I apologize for the late response. I have read the rebuttal, the other reviews and the comments.
> >
> > **Q3. The terms "effective robustness" and "robustness" are used interchangeably which is wrong and confusing."**
> > The authors write "*We further depict the lines of effective robustness, where we found that the conclusions remain the same. However, the improvement from increasing model scales and improving data quality is much higher than simply scaling up the datasets.*" When we look at effective robustness, we are interested in the residual difference between the measurement and the linear fit. I do not see any interventions in Fig.2 which go beyond the linear fit. Could you please clarify more directly what is meant here?
> >
> > **On the common vs unusual background issue**: It is good that the authors agreed to rename the groups to "easy" and "hard". Reading the other reviews, all reviewers "misunderstood" the naming convention. I find it very important to emphasize in the updated manuscript that the groups were created based on accuracy and not on occurrence frequency.
> >
> > **On analyzing the backgrounds frequency**: Since writing that the "hard" group is the "uncommon" one feels so natural, this should be analyzed. I would like to suggest to the authors to perform a caption based analysis to check whether "hard" examples actually do contain uncommon backgrounds in their captions. I understand that parsing the LAION dataset for the images themselves is unfeasible. I of course do not expect results on this until tomorrow; I merely think that performing this experiment would actually justify the authors in writing "common" and "uncommon" backgrounds and would also link the learned spurious correlations nicely to the training data. As it stands, the authors have not offered an explanation for why CLIP models perform worse on the "hard" backgrounds, which has been asked by Reviewer **2MXa**. Reviewer **2MXa** wrote: *I think the claim is somewhat "obvious": there exists a relatively strong correlation between the object captions and the parts of image backgrounds, CLIP will learn to align the backgrounds, i.e., spurious features. If the training dataset contains many examples of spurious correlations, then models will tend to be biased.* This is a natural question which can be analyzed.
> >
> > **On reframing the paper and the new title**: I agree with the other reviewers that reframing the paper to follow the spirit of ImageNet-A is a good idea. I agree with the other reviewers that a major revision might be necessary because the new title and the new framing effectively makes it a different paper. I think it might be helpful if the authors could post their reworked **abstract** here as it could be a good discussion base for the next reviewer-AC discussion stage.
> >
> > I greatly appreciate the new experiments and the authors' willingness to update their submission in such a major way. I am raising my score to 4, but would like to stress that I am very borderline on this. I am looking forward to the discussion with the other reviewers and AC on this submission. I felt in agreement with most of the other reviewers' comments and hope we can reach a collective decision together. I currently vote for a "weak reject" because I think that the paper lacks an analysis for why certain backgrounds are easy and other backgrounds are hard. I believe that the naming convention of "common" and "uncommon" aimed to (subconsciously) fill this gap by providing an untested hypothesis for the observed spurious correlations. I believe that merely presenting a dataset where CLIP trained models underperform is a bit weak and could be linked to the training data, as suggested by reviewer **2MXa**.

---

> > > ### Author Response · Authors · 2024-08-14
> > > **Background frequency results and further clarfications**
> > >
> > > Dear Reviewer BaJ3,
> > >
> > > Thank you for providing further feedback about our work and for raising the rating. We believe all your remaining concerns are addressable! **Although there is limited time before the end of the discussion, we have conducted an investigation of the background frequency to clarify your main concern about the observed spurious correlations.** Please find our responses below:
> > >
> > > > Residual differences in the effective robustness
> > >
> > > We kindly refer Reviewer BaJ3 to the uploaded figure, where we separately draw the linear fits in terms of each intervention:
> > > - In the uploaded Figure 1, we compared the effective robustness of CLIP models with `ViT/B/16` and `ViT/L/14`, shown as in the blue line and red line, respectively. It can be found that in the rightmost part of the x-axis, two red dots locate beyond the linear fit of the blue line, indicating an improvement of effective robustness of CLIP models based on larger backbone `ViT/L/14`.
> > > - In the uploaded Figure 2, we compared the effective robustness of CLIP models trained with `LAION2B` and `OpenAI` data, shown as in the blue line and red line, respectively. As both of them are considered web data with simple filtering, they have similar data quality that CLIP models trained on either `LAION2B` or `OpenAI` do not lead to improvements in terms of effective robustness.
> > > - In the uploaded Figure 3, we compared the effective robustness of CLIP models trained with high quality data (`HQ`) and relatively low quality data (`LQ`), shown as in the red line and blue line, respectively. It can be found that in the rightmost part of the x-axis, multiple red points locate beyond the linear fit of the blue line, indicating an improvement of effective robustness of CLIP models trained on high quality data (`HQ`).
> > >
> > > We have supplemented the aforementioned discussion in our revised manuscript.
> > >
> > > > Group naming issue
> > >
> > > We have revised our manuscript to avoid the confusion of the group naming.
> > >
> > > > Backgrounds frequency
> > >
> > > Thank you for your insightful question. We need to clarify that, **our theoretical analysis in Section 4 indeed explains  why CLIP models perform worse on the "hard" backgrounds, as we clarified [in the response to Reviewer 2MXa](https://openreview.net/forum?id=wWyumwEYV8&noteId=BNamSMaVj3)**.
> > >
> > > To provide more support of our theoretical explanation, we conduct an investigation in terms of the background frequency following your suggestion! Specifically, we adopt the searching tool of `Have I Been Trained` based on `clip-retrieval` to retrieve images from `LAION5B` closely matching to a given class name.
> > > - For each animal class, we obtain 500 images and count the frequencies for our considered backgrounds.
> > > - We control the sampled images aligned to the distribution of natural animal photos. For example, we filter out images with multiple distinct animal subjects or multiple distinct backgrounds. Due to time limit, we currently present three classes as follows and are extending our studies for revisions:
> > >
> > > | class name | easy bkg |     | hard bkg |    |
> > > |--------------|----------|-----|----------|----|
> > > | Ice Bear     | Ice      |  83 | Grass    |  7 |
> > > | Black Swan   | Water    | 101 | Earth    | 38 |
> > > | Flamingo     | Water    | 111 | Sky      | 16 |
> > >
> > > In general, it aligns with our conjecture that hard examples do contain uncommon backgrounds in the CLIP training data, e.g., `LAION5B`.
> > >
> > > We hope the supplemented results above could clarify the concern of Reviewer BaJ3 about our work.

---

> > > ### Author Response · Authors · 2024-08-14
> > > **The updated abstract**
> > >
> > > > Revision to the paper
> > >
> > > Thank you for acknowledging our revisions. Here we provide our reworked abstract for your reference:
> > >
> > > ```
> > > Large vision language models, such as CLIP, demonstrate impressive robustness to spurious features than single-modal models trained on ImageNet. However, existing test datasets are typically curated based on ImageNet-trained models, that aim to capture the spurious features inherited in ImageNet. Benchmarking CLIP models based on the ImageNet-oriented spurious features may not be sufficient to reflect the extent to which CLIP models are robust to spurious correlations inherited in CLIP training data, e.g., LAION. To this end, we craft a new challenging dataset named CounterAnimal that is designed to reveal the reliance of CLIP models on realistic spurious features.  Specifically, we split animal photos into groups according to the backgrounds, and then identify a pair of groups for each class where a CLIP model shows high-performance drops across the two groups. Our evaluations show that the spurious features captured by CounterAnimal are generically learned by CLIP models with different backbones and pretraining data, yet have limited influence for ImageNet models. We provide theoretical insights that the CLIP objective cannot offer additional robustness. Furthermore, we also re-evaluate strategies such as scaling up parameters and high-quality pre-trained data and find that they still help mitigate the spurious features, providing a promising path for future developments.
> > > ```

---

> > > > ### Comment · Reviewer_BaJ3 · 2024-08-14
> > > > **Response to the recent results**
> > > >
> > > > Thank you for the additional results and the revised abstract!
> > > >
> > > > **Additional results on the occurrence frequency**: I think these results are very nice. It would be great if you could extend this analysis for all backgrounds and groups for the final version. If the trend holds that the rarest animal-background combinations are the most challenging ones, I think it would also be ok to keep the naming of "common" and "uncommon" since it would have been tested then. And also, this maybe sounds a bit more intuitive. I also like the link between the training data and the test sets that we see.
> > > >
> > > > **On the revised abstract**: I think it reflects the message of the paper better than the previous one. I especially like that the backgrounds are not split according to their commonness, but according to the accuracy drop. I also like that the phrasing about the ImageNet models is a bit more cautious. Maybe you would want to also mention CounterAnimal-i and summarize the findings for this dataset, as well.
> > > >
> > > > I am raising my score to 6 now, especially because of the new analysis experiment, showing that the hardness of the backgrounds can indeed be tied to the co-occurrence frequency. I trust that the authors will conduct a larger scale analysis for the final version.
> > > >
> > > > Best,
> > > > Reviewer BaJ3

---

> > > > > ### Author Response · Authors · 2024-08-14
> > > > > **Thank you**
> > > > >
> > > > > Dear Reviewer BaJ3,
> > > > >
> > > > > We would like to thank you for your acknowledgment of our new results and the revised abstract! We are truly grateful for your insightful comments, which have helped improve our manuscript significantly!!
> > > > >
> > > > > Please feel assured we will follow your suggestions to extend the analysis and to incorporate all the results including those on `CounterAnimal-i` in future revisions. Thank you again for your valuable time and insights!

---

### Official Review · Reviewer_iLSC · 2024-07-10

**Soundness:** 2
**Presentation:** 3
**Contribution:** 2
**Rating:** 5
**Confidence:** 4

**Summary:**

This paper presents CounterAnimal, an evaluation dataset featuring two subsets: animals with common backgrounds and those with unusual backgrounds. The images were sourced from iNaturalist. Data with high CLIP accuracy are categorized as "Common",  while those with low CLIP accuracy are labeled as "Counter".  Results shows that CLIP models experience a greater accuracy drop compared to ImageNet models when tested on this dataset.

**Strengths:**

- This paper analyzes multiple factors affecting CLIP accuracy, including model size and training data quality.
- The paper combines both experimental results and theoretical analysis. The analysis in Section 5 is interesting and novel.
- The paper is well-written and easy to follow.

**Weaknesses:**

- The proposed dataset is not sufficiently robust to analyze the influence of spurious bias, as this is not the only difference between the common and counter datasets.
  - To analyze the accuracy drop caused by spurious features such as background, the background should be the only difference between common and counter image pairs. Prior work [4,5] has proposed such datasets focusing on background.
  - In the proposed dataset, other factors may influence the model accuracy gap besides background. For instance, as shown in Figure 1, the more varied gestures of ice bears on the right compared to the left could be a contributing factor to the accuracy drop.

- Current experiments cannot conclusively show that ImageNet models generalize better than CLIP.

   - As the common and counter groups are selected according to the CLIP accuracy (see line 165 in the paper),  they indicate easy and hard samples for CLIP.    Since ImageNet models have different training characteristics, it is natural that hard cases for these models may differ from those for CLIP, resulting in a smaller performance drop for ImageNet models. This result cannot support that ImageNet models are more robust than CLIP models.
   -  The accuracy drop from common to counter group can be greatly influenced by the model used to divide the common and counter dataset. Using the combined proposed common and counter dataset, a new Common' and Counter' dataset can be created based on the accuracy of ImageNet models. What is the impact of this dataset division on the accuracy drop for different models?


- Prior studies[1,2,3,4,5,6] have proposed datasets specifically to analyze the influence of background, which are not discussed in this work.  These datasets can be used for CLIP evaluation as they do not overlap with the CLIP training set. Additionally, creating datasets based on model accuracy in this work is similar to the approach in [6].

[1] Noise or Signal: The Role of Image Backgrounds in Object Recognition.

[2] Objectnet: A large-scale bias-controlled dataset for pushing the limits of object recognition models, NeurIPS 2019.

[3] Dataset Interfaces: Diagnosing Model Failures Using Controllable Counterfactual Generation.

[4] ImageNet-E: Benchmarking Neural Network Robustness via Attribute Editing, CVPR 2023.

[5] LANCE: Stress-testing Visual Models by Generating Language-guided Counterfactual Images, NeurIPS 2023.

[6] ImageNet-D: Benchmarking Neural Network Robustness on Diffusion Synthetic Object, CVPR2024.

**Questions:**

Please refer to the weaknesses.

**Limitations:**

This paper has discussed the limitations.

---

> ### Author Rebuttal · Authors · 2024-08-07
>
> Thank you for your constructive comments and suggestions! Please find our responses below.
>
> > Q1. The proposed dataset is not sufficiently robust to analyze the influence of spurious bias, as this is not the only difference between the common and counter datasets.
>
> A1. In our study, we primarily focus on the real-world spurious bias caused by backgrounds, while we also try our best to avoid the influence of other factors. As mentioned in Sec 2.1, during data curation, we took measures to control the influence of other covariates, e.g., ensuring that the main objects are in the major position. To further understand whether other factors such as gestures will introduce unintentional bias into the dataset, we count the proportion of different gestures within the class of Ice Bear. The results are listed as follows, which do not have an obvious difference in the distribution of gestures between common and counter scenarios. We will add the related discussion and explore other unintentional factors that may affect model predictions in our revision.
>
>
> | Ice Bear | Common | Counter |
> |----------|--------|---------|
> | Stand    | 89%    | 84%     |
> | Lying    | 13%    | 16%     |
>
> > Q2. Current experiments cannot conclusively show that ImageNet models generalize better than CLIP.
>
> A2. We apologize for any confusion in our description. In general, our conclusion that ImageNet models generalize better than CLIP models is limited to CounterAnimal. This is a preferred observation as it aligns with **our main goal of crafting a dataset that characterizes the spurious features that may widely exist in CLIP setups**. Prior to our work, there is no specialized benchmark curated for CLIPs to study the robustness of CLIPs to the real-world spurious correlations. Instead, previous works predominantly focus on the spurious features in ImageNets. Studying the OOD robustness of CLIPs on ImageNet specialized benchmarks is biased and may bring illusions about the robustness of CLIPs.
>
>
> > Q3. Prior studies [1,2,3,4,5,6] have proposed datasets specifically to analyze the influence of background, which are not discussed in this work. These datasets can be used for CLIP evaluation as they do not overlap with the CLIP training set. Additionally, creating datasets based on model accuracy in this work is similar to the approach in [6].
>
> A3. Many thanks for your suggested paper. Most of these suggested papers have shown that CLIP models are more robust to the associated spurious features than ImageNet models, indicating that they did not intend to capture spurious features within CLIP setups. The CounterAnimal remains the first dataset that can find some spurious features that are predominant for the CLIP setups while might not be so strong for the ImageNet benchmarks, which is the uniqueness of our paper.

---

> ### Comment · Reviewer_iLSC · 2024-08-09
> **Response**
>
> Thanks to the authors for providing the rebuttals, which partially addressed my concerns.
>
> * Regarding Q2,  I appreciate the adjustment to emphasize the spurious features found in CLIP setups. However, I did not find any discussion related to my original second point in Q2, which is quoted below:
>
>   > The accuracy drop from common to counter group can be greatly influenced by the model used to divide the common and counter dataset. Using the combined proposed common and counter dataset, a new Common' and Counter' dataset can be created based on the accuracy of ImageNet models. What is the impact of this dataset division on the accuracy drop for different models?
>
> * Regarding Q2, could you please clarify the following point? For example,  I would appreciate some examples of how prior work has focused on the spurious features in ImageNets.
>
>   >  Prior to our work, there is no specialized benchmark curated for CLIPs to study the robustness of CLIPs to the real-world spurious correlations. Instead, previous works predominantly focus on the spurious features in ImageNets.
>
> * Regarding Q3, the difference between this paper and prior studies remains unclear, which also affects the clarity of the paper's contribution. As also noted by reviewers KCA4 and BaJ3, the dataset creation makes it unsurprising that ImageNet models show a smaller gap between the two subsets. Therefore, it is confusing to claim that:
>
>   > The CounterAnimal remains the first dataset that can identify some spurious features predominant in CLIP setups **while these features might not be as strong for the ImageNet benchmarks**, which is the uniqueness of our paper.
>
> * Regarding Q3.  Prior studies have pointed out the spurious bias of the CLIP model about backgrounds, Therefore, a direct way to find CLIP-specific spurious bias is by splitting prior test set into two datasets—Counter and Common—based on CLIP accuracy. Could you further clarify how the proposed dataset differs from splitting prior test sets?

---

> > ### Author Response · Authors · 2024-08-10
> > **Further clarification on the remaining questions**
> >
> > Dear Reviewer iLSC,
> >
> > Thank you so much for engaging in the discussion! Now we provide more clarification on the remaining questions:
> >
> > > A new Common' and Counter' dataset can be created based on the accuracy of ImageNet models
> >
> > We apologize for not clearly responding to this question in the rebuttal. As also requested by Reviewer KCA4, we are currently collecting the new dataset based on the accuracy of Imagenet models and will update you once we have the preliminary results very soon!
> >
> > Nevertheless, we would like to clarify that, since previous ImageNet variant test sets are curated based on the accuracy of ImageNet models, the focus of this work is to provide a corresponding one for the CLIP models.
> >
> > > Examples of how prior work has focused on the spurious features in ImageNets.
> >
> > We briefly introduce some examples here and refer Reviewer iLSC to [a] for a detailed discussion of the examples:
> >
> > - ImageNetV2[b]: a reproduction of the ImageNet test set, where the authors introduce the distribution shifts by rerunning the Imagenet curation pipeline to collect a new test set;
> > - ObjectNet[c]: a test set of objects in a variety of scenes with 113 classes that overlap with ImageNet, where the authors introduce the distribution shifts by curating images with varying object poses, locations, etc.;
> > - ImageNet-Sketch[d]: the authors curate images from Google Image with queries "sketch of __", where __ is the standard class name;
> > - ImageNet-R[e]:  the authors curate images containing various renditions (e.g., paintings, embroidery, etc.) of ImageNet object classes;
> > - ImageNet-A[f]: the authors curate a hard version of the ImageNet test set with adversarial filtration based on the correctness of a ImageNet-trained model;
> >
> > All the aforementioned works introduce distribution shifts against the original ImageNet test sets in a way that they find that ImageNet models perform badly on the newly curated test sets. Based on those ImageNet variant test sets, [a] conducted extensive experiments and found that the performance gains of ImageNet models on the original ImageNet test sets can hardly be generalized to the ImageNet variant test sets. Therefore, it is suspected that the ImageNet models learn some spurious features from the original ImageNet train set, that do not hold on the ImageNet variant test sets.
> >
> > > The differences between this paper and prior studies
> >
> > We apologize for any confusion. The uniqueness of this work compared to those referred by Reviewer iLSC is that:
> > - [1,2] are mainly designed for ImageNet models. Especially, in ObjectNet[2], CLIP models have been shown to be more robust than ImageNet models;
> > - [3,4,5,6] mainly consider **synthetic distribution shifts instead of natural distribution shifts**. We acknowledge the value of the synthetic distribution shifts in debugging neural networks, and have already revised our manuscript to discuss them. Natural distribution shifts may better reflect the robustness against real-world spurious features [e].
> >
> > >  How this paper differs from splitting prior test set into two datasets—Counter and Common—based on CLIP accuracy
> >
> > Thank you for this insightful question!
> > - First, merely splitting test data according to the accuracy without considering the group information (e.g., backgrounds), will also involve additional factors, since the low-accuracy group will contain samples that are also affected by label noise or misspecification together [f]. Consequently, **the accuracy difference between the two groups may not uniquely reflect the influence of spurious features**. Note that our goal is to study the influence of spurious features. Therefore, we first labeled the background and then measured the differences between background groups, in order to measure the influence of spurious features in the backgrounds;
> > - As for the previous datasets specially designed for Imagenet models such as [1,2] referred by Reviewer iLSC, splitting them according to CLIP accuracy can not get rid of influence by other factors such as label noises, and may not reflect the real-world spurious features **uniquely in the CLIP training data**, as ImageNet models and CLIP models will both perform badly on the "counter/hard" splits.
> > - As for the synthetic datasets [3,4,5,6], they may not reflect the **real-world spurious features** in the CLIP training data.
> >
> >
> > **References**
> >
> > [a] Measuring Robustness to Natural Distribution Shifts in Image Classification, NeurIPS'20.
> >
> > [b] Do ImageNet Classifiers Generalize to ImageNet? ICML'19.
> >
> > [c] ObjectNet: A large-scale bias-controlled dataset for pushing the limits of object recognition models, NeurIPS'19.
> >
> > [d] Learning Robust Global Representations by Penalizing Local Predictive Power, NeurIPS'19.
> >
> > [e] The many faces of robustness: A critical analysis of out-of-distribution generalization, ICCV'21.
> >
> > [f] Natural adversarial examples, ECCV'21.
> >
> > [g] ZIN: When and How to Learn Invariance Without Environment Partition? NeurIPS'22.

---

> > > ### Author Response · Authors · 2024-08-10
> > > **Follow-up experiments on splitting prior test set into two datasets based on CLIP accuracy**
> > >
> > > Dear Reviewer iLSC,
> > >
> > > Following your suggestion of the last question, we conduct additional ablation study that uses ImageNet-a[f] to create common and counter splits based on the `CLIP-ViT/B/32-LAION400M`. Meanwhile, as the original ImageNet-a is curated based on the `ResNet-50` trained on ImageNet, therefore the accuracy of `ResNet-50-ImageNet` is 0.
> > >
> > > In the table below, we present the accuracies of CLIP models and ImageNet models across the common and counter groups. It can be found that
> > > - CLIP models demonstrate a larger performance drop from common to counter groups, which is as expected;
> > > - However, **none of the ImageNet models have a better generalization performances than CLIP models, despite a smaller performance drop**. In contrast, in CounterAnimal, ImageNet models can achieve compeitive performances in the Common groups while suffer less performance drops, or better effective robustness[a].
> > > - The results demonstrate the new splits are more challenging to generalize for both CLIP and ImageNet models. It is hard to dissect the influence of spurious features from the other factors such as label noises or model misspecification[g].
> > >
> > >
> > > |     **CLIP**     |           | common | counter |  drop |  **ImageNet**  | common | counter | drop   |
> > > |:------------:|-----------|:------:|:-------:|:-----:|:----------:|:------:|:-------:|--------|
> > > |     RN50     |   OPENAI  |  0.502 |  0.144  | 0.358 |   AlexNet  |  0.04  |  0.012  |  0.028 |
> > > |     RN101    |   OPENAI  |  0.606 |  0.199  | 0.407 |    VGG11   |  0.032 |  0.011  |  0.021 |
> > > |    RN50-4    |   OPENAI  |  0.672 |  0.299  | 0.373 |    VGG13   |  0.048 |  0.012  |  0.036 |
> > > |    RN50-16   |   OPENAI  |  0.748 |  0.449  | 0.299 |    VGG19   |  0.046 |  0.015  |  0.031 |
> > > |    RN50-64   |   OPENAI  |  0.807 |  0.584  | 0.223 |  ResNet18  |  0.022 |  0.008  |  0.014 |
> > > |   ViT/B/16   | LAION400M |  0.749 |  0.216  | 0.533 |  ResNet34  |  0.039 |  0.013  |  0.026 |
> > > |   ViT/B/16   |   OPENAI  |  0.778 |  0.384  | 0.394 |  ResNet50  |    0   |    0    |    0   |
> > > |   ViT/B/16   |  DATACOMP |  0.812 |   0.38  | 0.432 |  ResNet101 |  0.09  |  0.037  |  0.053 |
> > > |   ViT/B/16   |  LAION2B  |  0.734 |  0.256  | 0.478 |  ViT/B/16  | 0.3767 |  0.1699 | 0.2068 |
> > > |   ViT/B/16   |   DFN2B   |  0.832 |  0.386  | 0.446 |  ViT/B/32  | 0.1937 |   0.07  | 0.1237 |
> > > |   ViT/B/32   | LAION400M |    1   |    0    |   1   |  ViT/L/16  |  0.286 |  0.138  |  0.148 |
> > > |   ViT/B/32   |   OPENAI  |  0.665 |  0.205  |  0.46 |  ViT/L/32  |  0.232 |  0.093  |  0.139 |
> > > |   ViT/B/32   |  DATACOMP |  0.689 |  0.192  | 0.497 | CONVNEXT-S |  0.473 |  0.264  |  0.209 |
> > > |   ViT/B/32   |  LAION2B  |  0.704 |   0.15  | 0.554 | CONVNEXT-B |  0.527 |  0.297  |  0.23  |
> > > |   ViT/L/14   | LAION400M |  0.858 |  0.354  | 0.504 | CONVNEXT-L |  0.538 |  0.343  |  0.195 |
> > > |   ViT/L/14   |   OPENAI  |  0.872 |  0.608  | 0.264 |            |        |         |        |
> > > |   ViT/L/14   |  DATACOMP |  0.902 |   0.62  | 0.282 |            |        |         |        |
> > > |   ViT/L/14   |  LAION2B  |  0.856 |  0.441  | 0.415 |            |        |         |        |
> > > |   ViT/L/14   |   DFN2B   |  0.914 |  0.593  | 0.321 |            |        |         |        |
> > > | ViT/L/14-336 |   OPENAI  |  0.893 |  0.691  | 0.202 |            |        |         |        |
> > > |   ViT/H/14   |  LAION2B  |  0.869 |  0.477  | 0.392 |            |        |         |        |
> > > |   ViT/H/14   |   DFN5B   |  0.926 |  0.626  |  0.3  |            |        |         |        |
> > > | ViT/H/14-384 |   DFN5B   |  0.961 |  0.745  | 0.216 |            |        |         |        |
> > > |   ViT/G/14   |  LAION2B  |  0.877 |  0.494  | 0.383 |            |        |         |        |
> > > |  ViT-bigG/14 |  LAION2b  |  0.908 |  0.597  | 0.311 |            |        |         |        |
> > > |  CONVNEXT-B  | LAION400M |  0.509 |   0.13  | 0.379 |            |        |         |        |
> > > |  CONVNEXT-BW |  LAION2B  |  0.546 |  0.189  | 0.357 |            |        |         |        |

---

> > ### Author Response · Authors · 2024-08-12
> > **Preliminary results of suggested experiments are released**
> >
> > Dear Reviewer iLSC,
> >
> > Following your suggestion, we curate a new test set `CounterAnimal-i` according to the accuracy of ImageNet models. The detailed results and discussions about `CounterAnimal-i`  are present in [the latest general response](https://openreview.net/forum?id=wWyumwEYV8&noteId=4IR8MTyJIm), as the experiments are also suggested by Reviewer KCA4.
> >
> > The results of the experiments on `CounterAnimal-i` show that **curating the OOD test data according to different models will reveal different spurious features** and one needs to be cautious when selecting the proper OOD test data to evaluate the robustness against spurious features. Since most of the previous OOD test sets are designed for ImageNet models, it again highlights the necessity and significance of a test benchmark like CounterAnimal specifically for CLIP models.
> >
> > Please kindly let us know whether the aforementioned responses address your concerns. We would sincerely appreciate it if you could jointly consider our responses above when making the final evaluation of our work. Thank you again for your time and insightful suggestions about our work!

---

> ### Comment · Reviewer_iLSC · 2024-08-13
> **Final rating update**
>
> Thanks a lot to the authors for the detailed response and new experiments.  I appreciate the changes to the title and the additional experiments, which have improved this work a lot in the rebuttal. I also appreciate the contribution of creating a real-world dataset about backgrounds.  Considering all the comments and discussions,  I am raising my rating to 5.
>
> I share Reviewer 2MXa's concern that the paper needs significant revisions due to the shift in title and focus. I want to emphasize the modifications in the experimental results under the new shift. The original paper takes test accuracy as main experimental results to support the old title. However, test accuracy is not solid enough to be the main results for the new title as they are heavily influenced by the model used for dataset splitting. More analysis and results on spurious features are needed.
>
> Thanks again to the authors for their efforts during the rebuttal and discussion.

---

> > ### Author Response · Authors · 2024-08-13
> > **We used **effective robustness** instead of test accuracies as the main supporting experimental results**
> >
> > Dear Reviewer iLSC,
> >
> > Thank you for acknowledging our responses and updating the rating lean to an acceptance.
> >
> > We feel necessary to clarify that **we do not use the test accuracy as the supporting experiments in our original manuscript**.
> > - **We use effective robustness[1] in the main figures to support our claims about the robustness of CLIP models on `CounterAnimal`**. The test accuracy results are adopted to provide more details.
> > - In response to Reviewer BaJ3, we also provide **more figures of effective robustness** (as attached in [the general response](https://openreview.net/forum?id=wWyumwEYV8&noteId=yKMULf9RNp)) to support our claims that scaling up the model parameters and increasing the quality of data help with improving robustness on `CounterAnimal`.
> >
> > Since our focus is to study the robustness of CLIP models under spurious features instead of comparing whether ImageNet models are more robust than CLIP models, the results and analysis in our original paper sufficiently help with addressing our main research question `Is there a benchmark that reflects the exact reliance on spurious features of CLIP?` in line 41, including
> > - the curation procedure of `CounterAnimal`;
> > - the experimental results of effective robustness of CLIP models on `CounterAnimal`;
> > - the influence of factors such as parameters of models, and the quality of data to the effective robustness of CLIP models on `CounterAnimal`;
> > - the theoretical analysis of why CLIP models still learn spurious features;
> >
> > **Therefore, it does not need significant revisions to accommodate with our new title, as all the original contents sufficiently address our main research question**. The revisions we need to make (most of which has already been done) are to:
> > - Adjust the abstract and introduction following [2], to avoid potential misunderstanding and precisely present the motivation of our work: We do not focus on the comparison of CLIP and ImageNet models but on curating a test set specifically for CLIP models to study the robustness of CLIP models against spurious features;
> > - Supplement the ablation studies with `CounterAnimal-i` to strengthen our motivation;
> >
> > Please kindly let us know if our aforementioned revisions could address your concerns especially about the focus and the revisions of our work. We again thank you for your time and constructive comments!
> >
> > **References**
> >
> > [1] Measuring Robustness to Natural Distribution Shifts in Image Classification, NeurIPS'20.
> >
> > [2] Natural adversarial examples, ECCV'21.

---

### Official Review · Reviewer_KCA4 · 2024-07-12

**Soundness:** 2
**Presentation:** 3
**Contribution:** 2
**Rating:** 6
**Confidence:** 4

**Summary:**

In this work, the authors create an evaluation dataset comprising two groups, one with animals in usual backgrounds (common group) and another with unusual backgrounds (counter group). They then evaluate a suite of models of different backbones, model sizes, and datasets. They find that CLIP models do poorly than ImageNet-trained models, and generally high quality data or bigger model size improves counter group accuracy.

**Strengths:**

1. The CounterAnimal dataset is a nice contribution that can be of value to the community.
2. The authors have evaluated a number of models on the dataset and that too could be of value to the community.

**Weaknesses:**

Please see questions for more information.

**Questions:**

1. **Biased dataset:** The dataset is split into to common and counter group using a CLIP model. Therefore by construction, the CLIP models will perform poorly and it is no surprise that the ImageNet-trained models do a bit better. One could construct a split of this dataset where ImageNet models do better than the CLIP models. If the dataset was collected in a model agnostic way then the conclusions could potentially be more interesting.

2. **On the premise:** As such the premise or the primary question seems a bit vacuous. Models arguably learn different features and there will exist some type of evaluation where one does better than the other. But are there useful tasks/evaluations where ImageNet models are preferred over CLIP models? That is an interesting open question. This work doesn't necessarily start from there and create a benchmark that is supposed to represent a task. The authors rather create a biased dataset that by design make CLIP models perform poorly. Therefore the primary premise of the work seems erroneous. There is some value in the other evaluations so maybe the paper could be rewritten by positioning things differently.

3. **Lack of novelty:** Keeping the primary result aside, other conclusions like that better datasets or model sizes improve robustness are not new. Please see [1, 2, 3, 4].

[1] [Geirhos 2021] https://proceedings.neurips.cc/paper/2021/hash/c8877cff22082a16395a57e97232bb6f-Abstract.html

[2] [Idrissi 2022] https://arxiv.org/abs/2211.01866

[3] [Fang 2022] https://arxiv.org/abs/2205.01397

[4] [Nguyen 2022]  https://arxiv.org/abs/2208.05516

**Limitations:**

The authors have addressed some limitations. For the rest, please see my questions block.

---

> ### Author Rebuttal · Authors · 2024-08-07
>
> Thank you for your constructive comments and suggestions! Please find our responses below.
>
> > Q1. Biased dataset: The dataset is split into common and counter groups using a CLIP model. Therefore, by construction, the CLIP models will perform poorly, and it is no surprise that the ImageNet-trained models do a bit better. One could construct a split of this dataset where ImageNet models do better than the CLIP models. If the dataset was collected in a model agnostic way then the conclusions could potentially be more interesting.
>
> A1. We need to clarify that, **one of the major objectives of this work is to construct a benchmark specifically to reflect the spurious correlations learned by CLIPs**. To characterize the spurious features captured by CLIP, it is reasonable to use a CLIP model to curate the data. Our experiments in Section 3.1 justify that spurious features captured by CounterAnimal are general across different CLIP setups, and our experiments in Section 3.3 verify that the spurious features within CounterAnimal may not be so influential for ImageNet benchmarks. These results justify that our crafted CounterAnimal satisfies our original goal.
>
> Furthermore, we would like to note that, **previous comparisons between CLIPs and ImageNet models on ImageNet variant test sets are also biased**. However, biases are not entirely bad. The past few years witnessed a lot of developments built upon the ImageNet variant test sets. Therefore, **our benchmark share the same goal to provide a testbed for developing more advanced CLIP and vision-language models**.
>
> > Q2. On the premise: Are there useful tasks/evaluations where ImageNet models are preferred over CLIP models? That is an interesting open question. Therefore the primary premise of the work seems erroneous. There is some value in the other evaluations so maybe the paper could be rewritten by positioning things differently.
>
> A2. As explained in A1, our benchmark indeed captures spurious features learned by CLIP models. Future works can be developed upon our benchmark to mitigate the spurious correlations learned by CLIP models, similar to those built upon ImageNet variant testsets.
>
> Moreover, we need to clarify that the comparison between ImageNet and CLIP models is not to argue which model is the best universally. Rather, **we would like to highlight the biases existing in previous evaluations of CLIPs’ OOD robustness using ImageNet variant testsets**. Benchmarking models with improper test sets would cause illusions that CLIP models seem to resolve the spurious correlations, especially compared with ImageNet models. However, the experiments with CounterAnimals provide a sober look of the vulnerability of CLIP models to spurious correlations, and provide a platform for future developments of more advanced and robust CLIP and vision-language models.
>
> Finally, we understand that our original title may bring unnecessary misunderstandings about our work. We thus propose to revise it to `A Sober Look at the Robustness of CLIPs to Spurious Features` to more precisely reflect the contents of this work. Please kindly let us know if you have better suggestions to resolve the misunderstandings!
>
>
> > Q3. Lack of novelty: Keeping the primary result aside, other conclusions like that better datasets or model sizes improve robustness are not new.
>
> A3. As clarified in A1 and A2, CounterAnimal is more suitable than ImageNet variant test sets to benchmark the OOD robustness of CLIP models. As the main comparison results between CLIPs and ImageNet models already differs from the previous studies, extending the benchmarking of CLIP models to more variants is necessary to verify the conclusions of previous works. **We do not claim we are the first to discover those findings, rather, we are verifying those findings in order to provide insights of developing more robust CLIP and vision-language models**.

---

> ### Comment · Reviewer_KCA4 · 2024-08-09
>
> Thank you for your response and the new title—it’s definitely an improvement. I believe this paper warrants acceptance, but additional revisions are necessary. The paper would benefit from aligning more closely with the direction of Hendrycks et al. (2019) in terms of the motivation. Specifically, the ImageNet/ResNet comparison needs to be reframed, as its current presentation is potentially misleading.
>
> One potential experiment that could add value is identifying a common/counter split of the same ~13K dataset for ImageNet-trained models. Since you have the classes and backgrounds marked, it could be interesting to compare the differences between CLIP-based splits and ResNet-based splits. This comparison might reveals insights into the nature of spurious correlations in these two models. Incorporating these experiments and suggested changes could significantly enhance the manuscript. Hoping that the authors could make these changes, I am increasing my score.
>
> Hendrycks et al. (2019) Natural Adversarial Examples

---

> > ### Author Response · Authors · 2024-08-10
> > **Thank you and we are working on the required experiments**
> >
> > Dear Reviewer KCA4,
> >
> > Thank you for acknowledging our revisions to the title and the paper an improvement that warrants acceptance!
> >
> > In addition to the existing revisions, we promise that we will revise our manuscript more aligned with [1] following your suggestion.
> >
> > Meanwhile, we are also working on creating a similar common/counter split based on the ImageNet-trained models. We will share more details during the discussion period very soon once we have some preliminary results!
> >
> >
> > **References**
> >
> > [1] Natural adversarial examples, ECCV'21.

---

> > > ### Author Response · Authors · 2024-08-12
> > > **Preliminary results of suggested experiments are released**
> > >
> > > Dear Reviewer KCA4,
> > >
> > > We would like to thank you again for your time and constructive comments about our work. Following your suggestion, we curate a new test set `CounterAnimal-i` according to the accuracy of ImageNet models. The detailed results and discussions about `CounterAnimal-i`  are present in [the latest general response](https://openreview.net/forum?id=wWyumwEYV8&noteId=4IR8MTyJIm), as the experiments are also suggested by Reviewer iLSC.
> > >
> > > The results of the experiments on `CounterAnimal-i` show that **curating the OOD test data according to different models will reveal different spurious features** and one needs to be cautious when selecting the proper OOD test data to evaluate the robustness against spurious features. Since most of the previous OOD test sets are designed for ImageNet models, it again highlights the necessity and significance of a test benchmark like CounterAnimal specifically for CLIP models.
> > >
> > > Please kindly let us know if our additional experiments could address your concerns. Thank you again for your valuable suggestions about our work!

---

### Official Review · Reviewer_2MXa · 2024-07-13

**Soundness:** 3
**Presentation:** 3
**Contribution:** 3
**Rating:** 5
**Confidence:** 5

**Summary:**

This work asks one interesting question: "Do CLIP models always generalize better than ImageNet models?" Driven by this question, this work proposes a new benchmark dataset named CounterAnimal. This dataset consists of a) the common group: comprising animals in common backgrounds, and b) the counter group: including animals in plausible yet unusual backgrounds. The main idea is that the performance drops from the common to counter groups quantify the reliance on spurious background features for animal predictions. The main observation is that CLIP models exhibit notable performance drops when tested on the counter group. In comparison, ImageNet models can be more robust than CLIP models.

**Strengths:**

- It is always good to see a new and novel dataset proposed for evaluating CLIP and ImageNet-trained models. The proposed dataset CounterAnimal is complementary to existing datasets that cannot reflect the robustness of CLIP models to spurious correlations.

- The dataset construction is well-presented. The statistics, curation, background labeling, and spurious discovery are well introduced in Section 2

- The analysis around spurious correlation is good. This work tries to give insights from several aspects, such as pre-trained datasets, scaling up, and learning paradigms. The observations are sound to me.

**Weaknesses:**

-  I found the analysis of why CLIPs rely on spurious features interesting. However, I think the claim is somewhat "obvious": there exists a relatively strong correlation between the object captions and the parts of image backgrounds, CLIP will learn to align the backgrounds, i.e., spurious features. If the training dataset contains many examples of spurious correlations, then models will tend to be biased.

- I am curious about why ImageNet models may not be so influenced by the spurious bias in CounterAnimal. Is this because the ImageNet training set does not have too many spurious correlation examples? Or ImageNet has a spurious bias but such bias is different from the one in CounterAnimal? Please provide a discussion or share some insights on this question.

- This paper adopts absolute performance drop in Section 3.3. Such a metric may not be so robust. For example, model A drops from 40 to 39, and model B drops from 90 to 89. They drop the same but the I would say model B is better. Please comment on this, and discuss the metric of absolute performance drop.

**Questions:**

- ImageNet models are not so biased toward spurious correlations compared with CLIP models. Why? Is this because the ImageNet training set does not have too many examples that exhibit spurious correlations?

- While I appreciate this work includes results on ColoredCOO and ColoredMINIST, some other spurious correlation benchmarks (e.g., WaterBirds) would be greater if they were also included.

**Limitations:**

The dataset is proposed, so a discussion on the potential bias/ privacy is needed. I appreciate this work highlights the future improvement of expanding semantic scope, data source, and ImageNet testbed.

---

> ### Author Rebuttal · Authors · 2024-08-07
>
> Thank you for your constructive comments and suggestions! Please find our responses below.
>
> > Q1. I think the claim is somewhat "obvious": there exists a relatively strong correlation between the object captions and the parts of image backgrounds, CLIP will learn to align the backgrounds, i.e., spurious features. If the training dataset contains many examples of spurious correlations, then models will tend to be biased.
>
> A1. We need to clarify that, our theoretical analysis has unique and significant values for its implications to both theory and practice:
>
> - From the theoretical perspective, it complements the literature of theoretical studies about CLIPs. Prior to this work, previous works such as [1,2] mainly focus on developing theories and experiments justifying the superior OOD generalization capabilities of CLIPs in ImageNet variant tests. To the best of our knowledge, **our theory is the first to provably demonstrate the drawbacks of CLIPs in OOD generalization**, providing the foundation for future developments tackling the issue.
>
> - As a direct implication, the theory shows that CLIP training objectives can not offer additional robustness against spurious correlations. Consequently, the biases in large-scale multimodal training data such as LAION-5B is the major source for the spurious correlations learned by CLIPs. **However, prior to our work, there is no proper benchmark capturing the spurious correlations for large-scale multimodal training data**.
>
>
> [1] Identifiability Results for Multimodal Contrastive Learning, ICLR’23.
>
> [2] Does CLIP’s generalization performance mainly stem from high train-test similarity? ICLR’24.
>
>
>
> > Q2. I am curious about why ImageNet models may not be so influenced by the spurious bias in CounterAnimal. Is this because the ImageNet training set does not have too many spurious correlation examples? Or ImageNet has a spurious bias but such bias is different from the one in CounterAnimal? Please provide a discussion or share some insights on this question.
>
> A2. Thank you for the insightful question. First, we need to clarify that, ImageNet models still learn the spurious correlations as evidenced by the performance gaps in CounterAnimal benchmark. One reason for the lower performance gaps of ImageNet models is that, **CounterAnimal is specifically designed to reflect the biases of the CLIP pretrained datasets**, similar to ImageNet variant test sets which are designed for ImageNets[3,4]. A key takeaway for the phenomenon is that, we need to carefully choose suitable benchmarks to evaluate the OOD robustness of different models.
>
> [3] Do ImageNet Classifiers Generalize to ImageNet? ICML’19.
>
> [4] Natural Adversarial Examples, CVPR’21.
>
>
> > Q3. This paper adopts absolute performance drop in Section 3.3. Such a metric may not be so robust. For example, model A drops from 40 to 39, and model B drops from 90 to 89. They drop the same but the I would say model B is better. Please comment on this and discuss the metric of absolute performance drop.
>
> A3. Thank you for the suggestion. We further depict the lines of effective robustness, where we found that the conclusions remain the same. However, the improvement from increasing model scales and improving data quality is much higher than simply scaling up the datasets.
>
> > Q4. While I appreciate this work includes results on ColoredCOO and ColoredMINIST, some other spurious correlation benchmarks (e.g., WaterBirds) would be greater if they were also included.
>
> A4. We would like to note that the WaterBirds benchmark is similar to ColoredCOO, that synthesizes unnatural images by composing birds and different image backgrounds. Essentially it can be considered as a binary classification variant of ColoredCOCO and ColoredMNIST, hence the results of ColoredCOO and ColoredMNIST also generalizes to WaterBirds.
> In addition, we need to clarify that the main focus of this work is to identify natural spurious correlations learned by CLIPs instead of the synthetic ones. CounterAnimal essentially captures the desired real-world spurious features, differing from previous synthetic benchmarks like WaterBirds.
>
> > Q5. The dataset is proposed, so a discussion on the potential bias/ privacy is needed.
>
> A5. As discussed in question 12 of the checklist (see Appendix), we use the publicly available data from the internet, following the CC BY-NC license, which permits scientific use. We have revised the Broader Impacts in our manuscript to include a discussion regarding the issue.

---

> ### Comment · Reviewer_2MXa · 2024-08-12
>
> Dear Authors,
>
> Thank you for providing the response. I share the concern about using a CLIP model to curate the data. The newly added CounterAnimal-i dataset is helpful, but it could introduce differences in some of the observations made in the submission. For example, it raises questions about whether the title, “Do CLIP Models Always Generalize Better than ImageNet Models?” can still be fully addressed.
>
> The authors now emphasize that “the focus of this work is to construct a test set for studying the robustness of CLIP models against real-world spurious features.” While this is a valid focus, it necessitates significant revisions to reflect this shift. For instance, the abstract and introduction currently do not align with this focus.
>
> Given these points, I am actually in the boderline case. I would like to point out that the motivation behind the work is always good—whether it is to answer the title’s question or to study the real-world spurious features of CLIP models.
>
> However, I would like to ***draw the Area Chairs’ attention*** to the fact that this submission needs significant modification, and some observations should be carefully adjusted based on the newly curated dataset. The presentation also needs to be revised to align with the goal of studying real-world spurious features of CLIP models.
>
> Best regards,
>
> Reviewer 2MXa

---

> ### Author Response · Authors · 2024-08-12
> **The original maunscript does not need significant modifications**
>
> Dear Reviewer 2MXa,
>
> Thank you for engaging in the discussion and acknowledging our motivation. We need to clarify that **as our focus is to study the robustness of CLIP models, the additional experiments do not necessarily introduce significant modifications**.
>
> Meanwhile, to avoid any potential misunderstandings, we have changed the title to `A Sober Look at the Robustness of CLIPs to Spurious Features` to **more precisely reflect the research problem that our original abstract and introduction are addressing**.
>
> To accommodate the concern of adopting CLIP models to curate the splits, we have revised our abstract and the introduction to more align with the corresponding work for ImageNet [1], which curates an OOD test set `ImageNet-a` according to the accuracy of ImageNet models. **We have revised our manuscript to clarify that it is a common practice in the literature to curate the OOD test set to evaluate the robustness of neural networks. The revisions are also acknowledged and suggested by Reviewer KCA4.**
>
> Please kindly let us know if you still feel any additional revisions are needed. We would sincerely appreciate it if you could jointly take our clarifications above when making the final evaluation of this work.

---

### Author Rebuttal · Authors · 2024-08-07

We sincerely appreciate all the reviewers for their careful reviews and constructive feedback. We are also grateful for reviewers’ recognitions of our efforts on dataset constructions, the empirical findings, as well as the theoretical analysis. In response, we would like to emphasize the contributions of this paper and clarify the potentially misleading content.

**Dataset Contribution**. We would like to highlight the biases existing in previous evaluations of CLIPs’ OOD robustness using ImageNet variant test sets. **Benchmarking models with improper test sets would cause illusions that CLIP models seem to resolve the spurious correlations, especially compared with ImageNet models**. It motivates us to craft the CounterAnimal dataset, specifically capturing the spurious correlations within CLIP setups. The experiments with CounterAnimals provide a sober look of the vulnerability of CLIP models to spurious correlations and provide a platform for future developments of more advanced and robust CLIP and vision-language models.

**Comparative Experiments**. In general, **It is not our intention to claim that ImageNet models are inherently more robust to spurious features than CLIP models**. The comparative experiments between CLIP and ImageNet models in our study demonstrate that the features captured by CounterAnimal are generalizable across different CLIP setups. Furthermore, our experiments detailed in Section 3.3 suggest that the spurious features identified by CounterAnimal may have limited influence on ImageNet benchmarks.

Moreover, following the suggestions from the reviewers, we have applied measures of effective robustness to further support our conclusions in Section 3.3. These results are detailed in the appendix. Overall, our previous conclusions are upheld, with additional findings indicating that **improvements from increasing model scales and enhancing data quality significantly outweigh the benefits of merely scaling up datasets**.

Finally, we understand that our original title may bring unnecessary misunderstandings about our work. We thus propose to revise it to `A Sober Look at the Robustness of CLIPs to Spurious Features` to more precisely reflect the contents of this work. Please kindly let us know if you have better suggestions to resolve the misunderstandings!

---

### Comment · Area_Chair_BKNm · 2024-08-08
**Follow-Up on Author Responses to Your Review Comments Before Aug 13th**

Dear Reviewers,

Thank you for your time and effort in reviewing the manuscript. Please kindly review and acknowledge the authors' rebuttal, and feel free to request any additional clarification if needed.

Your AC

---

### Comment · Area_Chair_BKNm · 2024-08-12

Dear Reviewers,

This is a friendly reminder that the authors have submitted a rebuttal and the end of the discussion period is August 13th AoE. Due to the tight deadline please take a look at it as soon as possible and check if your questions/comments have been properly addressed. At the very least, please acknowledge that you have read the authors' response to your review.

Thank you for your time and effort!

AC

---

### Author Response · Authors · 2024-08-12
**Preliminary results of model performances on splits created according to the accuracy of ImageNet models**

Dear Area Chair BKNm and Reviewers,

We would like to thank you again for your valuable time and constructive comments about our work. We follow the suggestions of Reviewer KCA4 and iLSC to further curate a new `CounterAnimal-i` dataset, for which we follow the same curation procedure while using ImageNet models (i.e., `ResNet50-ImageNet`) to construct Common and Counter splits.

Here, again we would like to clarify that the objective of this work is not for comparing which model is universally more robust than the other. Rather, **the focus of this work is to construct a test set for studying the robustness of CLIP models against real-world spurious features**:
- The construction of CounterAnimal follows the common practice in the literature[1].
- Prior to CounterAnimal, most of the test sets are constructed for ImageNet models. Comparing CLIP models with ImageNet models on OOD test sets specifically designed for ImageNet models is insufficient to reflect the robustness of CLIPs against real-world spurious features.

Now we present the results of CLIP and ImageNet models on `CounterAnimal-i`. We first present the summary of key observations below and then give the detailed performances in the next comment due to the character limit.

- CLIPs demonstrate better robustness against the spurious features discovered with ImageNet models. **It aligns with our expectation since different training data (e.g., LAION for CLIPs, and ImageNet for ImageNet models) follow different distributions and naturally contain different spurious features**. It also demonstrates the generality of our data curation method to reveal the spurious features of different models.
- More concretely, we list the common and counter split names for some of the classes, where we bold the background names differently across `CounterAnimal` and` CounterAnimal-i`. From the results, we can find that,
    - Using different models to split data captures the different spurious features.
    - Both CLIP and ImageNet models are good at generalizing under certain spurious features while performing worse than the other under some other spurious features. **It again highlights the necessity to curate an OOD test set for CLIP models, as CLIP models learn different spurious features than ImageNet models**.


|               | CounterAnimal |           | CounterAnimal-i |           |
|---------------|---------------|-----------|-----------------|-----------|
|               | common        | counter   | common          | counter   |
| Ostrich       | ground        | **water** | ground          | **rock**  |
| Brambling     | grass         | **sky**   | grass           | **water** |
| Bulbul        | sky           | **tree**  | sky             | **grass** |
| Vluture       | sky           | tree      | sky             | tree      |
| Box turtle    | grass         | **earth** | grass           | **water** |
| Ccommon iguana | earth         | shrub     | earth           | shrub     |
| Whiptail      | **earth**     | **human** | **water**       | **shurb** |
| Agama         | rock          | **tree**  | rock            | **grass** |
| Crocodile     | earth         | **grass** | earth           | **tree**  |



- Furthermore, we give the averaged performance drops of CLIP and ImageNet models under `CounterAnimal` and `CounterAnimal-i`, respectively. From the table below, two interesting observations can be found that:
    - CLIP models show similar averaged performance drops across `CounterAnimal` and `CounterAnimal-i`, which demonstrates the robustness of our curation method in revealing the spurious features of CLIP models;
    - When considering the averaged performance drops, it can be found that both models demonstrate similar performance drops, which again shows the necessity of using proper benchmarks to evaluate the robustness of different models.

|                 | CounterAnimal | CounterAnimal-i | AVG.  |
|-----------------|---------------|-----------------|-------|
| CLIP models     | 16.96         | 15.71           | 16.34 |
| ImageNet models | 13.95         | 21.9            | 17.93 |

**References**

[1] Natural adversarial examples, ECCV'21.

---

> ### Author Response · Authors · 2024-08-12
> **Detailed performances on CounterAnimal-i**
>
> Following the last response, we present the detailed results of CLIP and ImageNet models on `CounterAnimal-i` in this comment due to the character limits.
>
> |   **CLIP**   |           | common | counter |  drop | **ImageNet** | common | counter |  drop |
> |:------------:|-----------|:------:|:-------:|:-----:|:------------:|:------:|:-------:|:-----:|
> |     RN50     |   OPENAI  |  60.9  |  42.56  | 18.34 |    AlexNet   |  59.27 |  31.87  |  27.4 |
> |     RN101    |   OPENAI  |  61.22 |  40.25  | 20.97 |     VGG11    |  73.82 |  46.66  | 27.15 |
> |    RN50-4    |   OPENAI  |  64.4  |  47.85  | 16.55 |     VGG13    |  74.32 |  48.08  | 26.24 |
> |    RN50-16   |   OPENAI  |   72   |  57.65  | 14.35 |     VGG19    |  77.07 |  52.77  |  24.3 |
> |    RN50-64   |   OPENAI  |  81.41 |  68.36  | 13.05 |   ResNet18   |  73.08 |  49.19  | 23.89 |
> |   ViT/B/16   | LAION400M |  73.71 |  53.22  | 20.49 |   ResNet34   |  77.52 |  52.74  | 24.78 |
> |   ViT/B/16   |   OPENAI  |  73.46 |  56.36  |  17.1 |   ResNet50   |  80.71 |  52.97  | 27.74 |
> |   ViT/B/16   |  DATACOMP |  79.33 |   63.1  | 16.23 |   ResNet101  |  82.35 |  59.46  |  22.9 |
> |   ViT/B/16   |  LAION2B  |  68.66 |  52.13  | 16.53 |   ViT/B/16   |  85.06 |  66.64  | 18.42 |
> |   ViT/B/16   |   DFN2B   |  83.39 |  68.75  | 14.64 |   ViT/B/32   |  78.27 |  55.42  | 22.85 |
> |   ViT/B/32   | LAION400M |  57.32 |  37.61  | 19.71 |   ViT/L/16   |  83.65 |  64.03  | 19.63 |
> |   ViT/B/32   |   OPENAI  |  66.95 |  47.12  | 19.84 |   ViT/L/32   |  80.28 |  59.17  | 21.11 |
> |   ViT/B/32   |  DATACOMP |  73.59 |  53.99  |  19.6 |  CONVNEXT-S  |  88.87 |  73.86  | 15.01 |
> |   ViT/B/32   |  LAION2B  |  67.37 |  47.64  | 19.73 |  CONVNEXT-B  |  88.92 |  74.48  | 14.44 |
> | ViT/B/32-256 |  DATACOMP |  78.18 |   60.8  | 17.39 |  CONVNEXT-L  |  89.88 |  77.21  | 12.67 |
> |   ViT/L/14   | LAION400M |  77.96 |  60.85  | 17.11 |              |        |         |       |
> |   ViT/L/14   |   OPENAI  |  81.67 |  67.55  | 14.12 |              |        |         |       |
> |   ViT/L/14   |  DATACOMP |  88.87 |  77.06  | 11.82 |              |        |         |       |
> |   ViT/L/14   |  LAION2B  |  78.89 |  63.14  | 15.75 |              |        |         |       |
> |   ViT/L/14   |   DFN2B   |  88.72 |  77.51  | 11.21 |              |        |         |       |
> | ViT/L/14-336 |   OPENAI  |  84.09 |  71.62  | 12.47 |              |        |         |       |
> |   ViT/H/14   |  LAION2B  |  83.77 |  71.04  | 12.72 |              |        |         |       |
> |   ViT/H/14   |   DFN5B   |  89.32 |  79.65  |  9.68 |              |        |         |       |
> | ViT/H/14-384 |   DFN5B   |  92.55 |  83.19  |  9.36 |              |        |         |       |
> |   ViT/G/14   |  LAION2B  |  84.46 |  68.16  | 16.31 |              |        |         |       |
> |  ViT-bigG/14 |  LAION2b  |  86.39 |  74.03  | 12.36 |              |        |         |       |
> |  CONVNEXT-B  | LAION400M |  52.06 |  38.85  | 14.22 |              |        |         |       |
> |  CONVNEXT-BW |  LAION2B  |  57.19 |  38.74  | 18.45 |              |        |         |       |

---

### Decision · Program_Chairs · 2024-09-25

**Decision:**

Accept (poster)

**Comment:**

**Summary**
CLIP is believed to be robust to distribution shifts due to its performance on ImageNet shifts. This paper presents a new evaluation dataset named CounterAnimal that contains a common group of animals in common backgrounds and a counter group including animals in plausible yet unusual backgrounds. When evaluated with this dataset, CLIP exhibits performance drops challenging common assumptions on the robustness of this model.

**Recommendation**
Accept. All the reviewers tend towards acceptance and the authors did a good job with the rebuttal. I also believe this work is interesting as it challenges a common assumption about CLIP in the research community.